# Whole-brain annotation and multi-connectome cell typing of *Drosophila*

Philipp Schlegel[1,2], Yijie Yin[2], Alexander S. Bates[1,3,4], Sven Dorkenwald[5,6], Katharina Eichler[2], Paul Brooks[2], Daniel S. Han[1,7], Marina Gkantia[2], Marcia dos Santos[2], Eva J. Munnelly[2], Griffin Badalamente[2], Laia Serratosa Capdevila[2], Varun A. Sane[2], Alexandra M. C. Fragniere[2], Ladann Kiassat[2], Markus W. Pleijzier[1], Tomke Stürner[1,2], Imaan F. M. Tamimi[2], Christopher R. Dunne[2], Irene Salgarella[2], Alexandre Javier[2], Siqi Fang[2], Eric Perlman[8], Tom Kazimiers[9], Sridhar R. Jagannathan[2], Arie Matsliah[6], Amy R. Sterling[6,10], Szi-chieh Yu[6], Claire E. McKellar[6], FlyWire Consortium*,**, Marta Costa[2], H. Sebastian Seung[5,6], Mala Murthy[6], Volker Hartenstein[11], Davi D. Bock[12✉] & Gregory S. X. E. Jefferis[1,2✉]

The fruit fly *Drosophila melanogaster* has emerged as a key model organism in neuroscience, in large part due to the concentration of collaboratively generated molecular, genetic and digital resources available for it. Here we complement the approximately 140,000 neuron FlyWire whole-brain connectome[1] with a systematic and hierarchical annotation of neuronal classes, cell types and developmental units (hemilineages). Of 8,453 annotated cell types, 3,643 were previously proposed in the partial hemibrain connectome[2], and 4,581 are new types, mostly from brain regions outside the hemibrain subvolume. Although nearly all hemibrain neurons could be matched morphologically in FlyWire, about one-third of cell types proposed for the hemibrain could not be reliably reidentified. We therefore propose a new definition of cell type as groups of cells that are each quantitatively more similar to cells in a different brain than to any other cell in the same brain, and we validate this definition through joint analysis of FlyWire and hemibrain connectomes. Further analysis defined simple heuristics for the reliability of connections between brains, revealed broad stereotypy and occasional variability in neuron count and connectivity, and provided evidence for functional homeostasis in the mushroom body through adjustments of the absolute amount of excitatory input while maintaining the excitation/inhibition ratio. Our work defines a consensus cell type atlas for the fly brain and provides both an intellectual framework and open-source toolchain for brain-scale comparative connectomics.

The adult fruit fly represents the current frontier for whole-brain connectomics. With 139,255 neurons, the newly completed full adult female brain (FAFB) connectome is intermediate in log scale between the first connectome of *Caenorhabditis elegans* (302 neurons[3,4]) and the mouse ($10^8$ neurons), a desirable but currently intractable target[5]. The availability of a complete adult fly brain connectome now allows brain-spanning circuits to be mapped and linked to circuit dynamics and behaviour as has long been possible for the nematode and more recently the *Drosophila* larva (3,016 neurons)[6]. However, the adult fly has richer behaviour, including complex motor control while walking or in flight[7], courtship behaviour[8], involved decision making[9], flexible associative memory[10,11], spatial learning[12] and complex[13,14] multisensory[15,16] navigation.

The FlyWire brain connectome reported in our companion paper[1] is by some margin the largest and most complex yet obtained. The full connectome, derived from the approximately 100 teravoxel FAFB whole-brain electron microscopy (EM) volume[17], can be represented as a graph with 139,255 nodes and around 15.1 million weighted edges. Here we formulate and answer key questions that are essential to interpreting connectomes at this scale regarding (1) how we know which edges are important; (2) how we can simplify the connectome graph to aid automated or human analysis; and (3) the extent to which this connectome is a snapshot of a single brain or representative of this species as a whole (or have we collected a 'snowflake'?). These questions are inextricably linked with connectome annotation and cell type identification[18,19] within and across datasets.

At the most basic level, navigating this connectome would be extremely challenging without a comprehensive system of annotations, which we now provide. Our annotations represent an indexed

[1]Neurobiology Division, MRC Laboratory of Molecular Biology, Cambridge, UK. [2]Drosophila Connectomics Group, Department of Zoology, University of Cambridge, Cambridge, UK. [3]Department of Neurobiology and Howard Hughes Medical Institute, Harvard Medical School, Boston, MA, USA. [4]Centre for Neural Circuits and Behaviour, University of Oxford, Oxford, UK. [5]Computer Science Department, Princeton University, Princeton, NJ, USA. [6]Princeton Neuroscience Institute, Princeton University, Princeton, NJ, USA. [7]School of Mathematics and Statistics, University of New South Wales, Sydney, New South Wales, Australia. [8]Yikes, Baltimore, MD, USA. [9]kazmos, Dresden, Germany. [10]Eyewire, Boston, MA, USA. [11]Molecular, Cell and Developmental Biology, University of California Los Angeles, Los Angeles, CA, USA. [12]Department of Neurological Sciences, Larner College of Medicine, University of Vermont, Burlington, VT, USA. *A list of authors and their affiliations appears at the end of the paper. **A full list of members and their affiliations appears in the Supplementary Information. ✉e-mail: dbock@uvm.edu; jefferis@mrc-lmb.cam.ac.uk

and hierarchical human-readable parts list[18,20], enabling biologists to explore their systems and neurons of interest. Connectome annotation is also crucial to ensuring data quality as it inevitably reveals segmentation errors that must be corrected. Furthermore, there is a rich history in *Drosophila* of probing the circuit basis of a wide range of innate and learned behaviours as well as their developmental genetic origins; realizing the full potential of this dataset is only possible by cross-identifying cell types within the connectome with those characterized in the published and in-progress literature. This paper reports this key component of the connectome together with the open source tools (Table 1) and resources that we have generated. As the annotation and proofreading of the connectome are inextricably linked, the companion paper[1] and this paper will preferably be co-cited as they jointly describe the FlyWire resource.

Comparison with cell types proposed using the partial hemibrain connectome[2] confirmed that the majority of fly cell types is highly stereotyped, and defined simple rules for which connections within a connectome are reliable and therefore more likely to be functional. However, this comparison also revealed unexpected variability in some cell types and demonstrated that many cell types originally reported in the hemibrain could not be reliably reidentified. This discovery necessitated the development and application of a new robust approach for defining cell types jointly across connectomics datasets. Overall, this effort lays the foundation both for deep interrogation of current and anticipated fly connectomes from normal individuals, but also future studies of sexual dimorphism, experience-dependent plasticity, development and disease at the whole-brain scale.

## Hierarchical annotation of a connectome

Annotations defining different kinds of neurons are key to exploring and interpreting any connectome; but, with the FlyWire connectome—which we report jointly with the companion paper[1]—now exceeding the 100,000 neuron mark, they are also both of increased significance and more challenging to generate. We defined a comprehensive, systematic and hierarchical set of annotations based on the anatomical organization of the brain (Fig. 1 and Supplementary Videos 1 and 2), as well as the developmental origin and coarse morphology of neurons (Fig. 2). Building on these as well as validating cell types identified from pre-existing datasets, we then defined a set of consensus terminal cell types intended to capture the finest level of organization that is reproducible across brains (Fig. 3).

We first collected and curated basic metadata for every neuron in the dataset including soma position and side, and entry or exit nerve for afferent and efferent neurons, respectively (Fig. 1). Our group also predicted neurotransmitter identity for all neurons as reported elsewhere[21]. We then defined a hierarchy of four levels: flow > superclass > class > cell type, which provide salient labels at different granularities (Fig. 1a, Supplementary Table 1 and Extended Data Fig. 2).

The first two levels, flow and superclass, were densely annotated: every neuron is either afferent, efferent or intrinsic to the brain (flow) and falls into one of the nine superclasses: sensory (periphery to brain), motor (brain to periphery), endocrine (brain to corpora allata/cardiaca), ascending (ventral nerve cord (VNC) to brain), descending (brain to VNC), visual projection (optic lobes to central brain), visual centrifugal (central brain to optic lobes), or intrinsic to the optic lobes or the central brain (Fig. 1b and Supplementary Table 2). Mapping to the https://virtualflybrain.org/ (ref. 22) database enables cross-referencing of neurons and types with other publications (Methods). Note that due to an inversion of the left–right axis during the original acquisition of the FAFB dataset[17], identified during preparation of this work (Extended Data Fig. 1; see the 'FAFB laterality' section of the Methods), frontal figures in this work and the FlyWire connectome[1] have the fly's left on the viewer's left, and the fly's right on the viewer's right, that

## Table 1 | Software tools used

| Name | Github repository | Description |
|---|---|---|
| navis | navis-org/navis | Analysis (for example, NBLAST) and visualization of neuron morphologies in Python. |
| navis-flybrains | navis-org/navis-flybrains | Transform data between brain templates (including hemibrain and FAFB) in Python. |
| fafbseg-py | flyconnectome/fafbseg-py | Query and work with auto-segmented FAFB data (including FlyWire) in Python. |
| cocoa | flyconnectome/cocoa | Analysis suite for comparative connectomics in Python. |
| neuprint-python | connectome-neuprint/neuprint-python | Query data from neuPrint, developed by Stuart Berg (Janelia Research Campus). |
| fafbseg | natverse/fafbseg | Support for working with FlyWire segmentation, meshes and annotations in R. |
| neuprintr | natverse/neuprintr | Support for working with neuPrint databases including the hemibrain connectome in R. |
| coconat coconatfly | natverse/coconat natverse/coconatfly | Analysis suite for comparative connectomics in R. |
| Pyroglancer | SridharJagannathan/pyroglancer | Pythonic interface to neuroglancer for displaying neuronal data. |

is, the opposite of the usual convention. However, all side labels are biologically correct.

The class field contains pre-existing neurobiological groupings from the literature (for example, for central complex neurons; Supplementary Table 3) and is sparsely annotated (43%) for the central brain, in large part because past research has favoured some brain areas over others. In the optic lobes, 99% of neurons have a generic class based on their neuropil innervation. Finally, 98% of all central brain neurons were given a terminal cell type, a majority of which could be linked to at least one report in the literature (Fig. 1c). Our annotations for the optic lobes include cell types for 92% of neurons in both left and right optic lobes. A separate report[23] will describe comprehensive typing of all neurons intrinsic to the optic lobes. In total, we collected over 870,000 annotations for all 139,255 neurons; all are available for download and through neuroglancer scenes (Methods and Extended Data Fig. 11). A total of 32,388 (23%) neurons are intrinsic to the central brain and 77,536 (54%) neurons are intrinsic to the optic lobes. The optic lobes and the central brain are connected through 8,053 visual projection and 524 visual centrifugal neurons. The central brain receives afferent input through 5,512 sensory and 2,362 ascending neurons. Efferent output is realized through 1,303 descending, 80 endocrine and 106 motor neurons.

We find marked stereotypy in the number of central brain intrinsic neurons—for example, between the left and the right hemisphere, they differ by only 27 (0.1%) neurons. For superclasses with less consistency in left versus right counts, such as the ascending neurons (140, 11%), the discrepancies are typically due to ambiguity in the sidedness (Fig. 1d and Methods).

Combining the dense superclass annotation for all neurons with the connectome[1] gives a birds-eye view of the input/output connectivity of the central brain (Fig. 1f): 55% of the central brain's synaptic input comes from the optic system; 25% from the VNC through ascending neurons; and only 18% from peripheral sensory neurons. This is surprising as sensory neurons are almost as numerous as visual projection neurons (Fig. 1d,e); individual visual projection neurons therefore provide about 2.5 times more synapses, underscoring the value of this information stream. Input neurons make about two synapses onto central brain neurons for every one synapse onto output

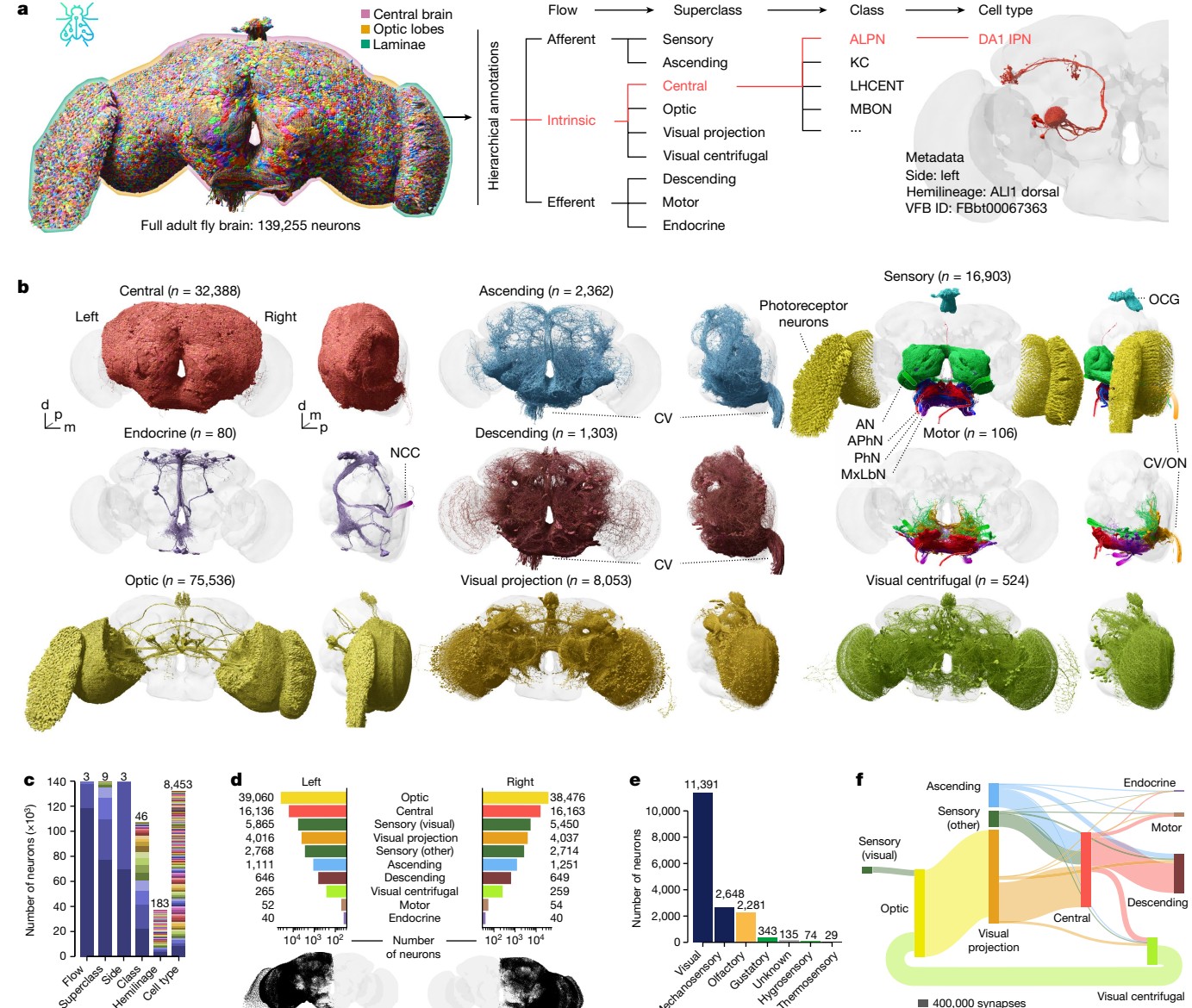

**Fig. 1 | Hierarchical annotation schema for a whole-brain connectome.**
**a**, Hierarchical annotation schema for the FlyWire dataset (see the companion paper[1]). Annotations for example cell type DA1 lPN (right) are highlighted in red. **b**, Renderings of neurons for each superclass. AN, antennal nerve; APhN, accessory pharyngeal nerve; CV, cervical connective; d, dorsal; m, medial; MxLbN, maxillary-labial nerve; NCC, corpora cardiaca nerves; OCG, ocellar ganglion; ON, occipital nerves; PhN, pharyngeal nerve; p, posterior.

**c**, Annotation counts per field. Each colour within a bar represents discrete values; the numbers above bars count the discrete values. **d**, Left versus right neuron counts per superclass. Bottom, the left and right soma locations, respectively. **e**, Breakdown of sensory neuron counts into modalities. **f**, Flow chart of superclass-level, feed-forward (afferent to intrinsic to efferent) connectivity.

neurons. Most output synapses target the VNC through descending neurons (75%); the rest provide centrifugal feedback onto the optic system (15%), motor neuron output (9%) and endocrine output to the periphery (1%).

## A full atlas of neuronal lineages

Our top-level annotations (flow, superclass, class) provide a systematic but relatively coarse grouping of neurons compared with >5,000 terminal cell types expected from previous work on the hemibrain[2]. We therefore developed an intermediate level of annotation based on hemilineages—this provides a powerful bridge between the developmental origin and molecular specification of neurons and their place within circuits in the connectome (Fig. 2a).

Central brain neurons and a minority of visual projection neurons are generated by around 120 identified neuroblasts per hemisphere. Each of these stem cells is defined by a unique transcriptional code and generates a stereotyped lineage in a precise birth order by asymmetric division[24–27] (Fig. 2b). Each neuroblast typically produces two hemilineages[28,29] that differ markedly in neuronal morphology and can express different neurotransmitters from one another, but neurons in each hemilineage usually express a single fast-acting transmitter[21,30]. Hemilineages therefore represent a natural functional as well as developmental grouping by which to study the nervous system. Within a hemilineage, neurons form processes that extend together in one cohesive bundle (the hemilineage tract) that enters, traverses and interconnects neuropil compartments in a stereotypical pattern (Fig. 2c). Comparing these features between EM and previous light-level

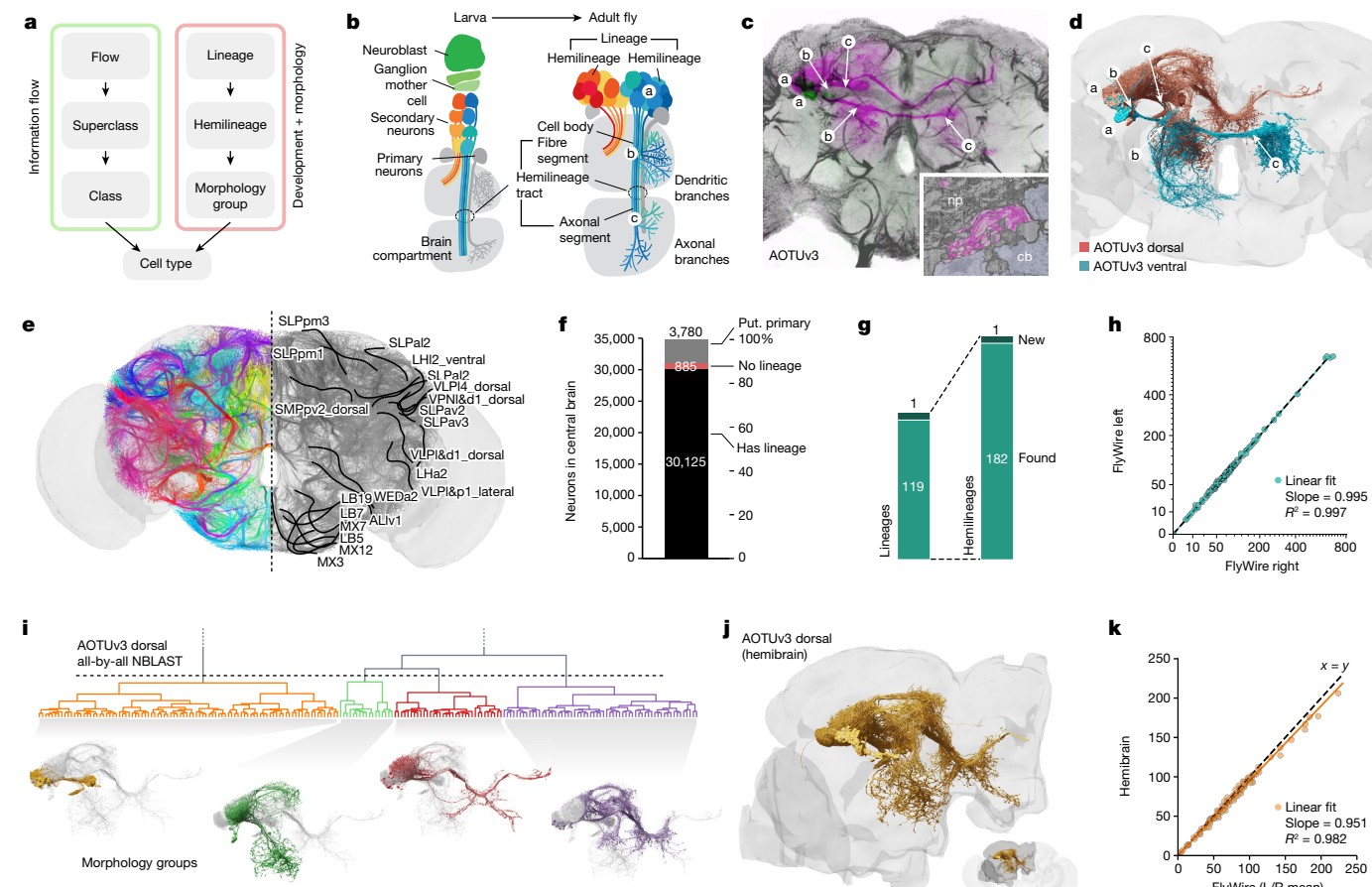

**Fig. 2 | Annotation of developmental units. a**, Illustration of the two complementary sets of annotations. **b**, Developmental organization of neuroblast hemilineages. **c**, Light-level image of an example AOTUv3 lineage clone; the lower case letters link canonical features of each hemilineage to the cartoon in **b**. Inset: cell body fibre tract in the EM. cb, cell body; np, neuropil. **d**, AOTUv3 neurons in FlyWire split into its two hemilineages. **e**, Cell body fibre bundles from all identified hemilineages, partially annotated on the right. **f**, The number of central brain neurons with an identified lineage; annotation of (putative (put.)) primary neurons is based on literature or expert assessment of morphology. **g**, The number of identified unique (hemi)lineages. **h**, Left versus right number of neurons contained in each hemilineage. **i**, Example morphological clustering of the AOTUv3 dorsal hemilineage reveals four distinct subgroups. **j**, Neurons belonging to the AOTUv3 dorsal hemilineage identified in the hemibrain connectome. **k**, FlyWire versus hemibrain number of neurons for cross-identified hemilineages.

data[31–34] enabled us to compile the first definitive atlas of all hemilineages in the central brain (Fig. 2c–e and Methods).

In total, we successfully identified 120 neuroblast lineages in FlyWire comprising 183 hemilineages for 88% (30,233 total) of central brain neurons (Fig. 2e,f and Extended Data Fig. 3). The unassigned neurons are likely primary neurons born during embryonic development, which account for 10% of neurons in the adult brain[35,36]. We tentatively designated 3,779 (11%) as primary neurons either based on specific identification in the literature[27] or expert assessment of diagnostic morphological features such as larger cell bodies and broader projections. A further 797 neurons (2%) did not co-fasciculate with any hemilineage tracts, even though their morphology suggested that they are later-born secondary neurons[37]. This developmental atlas is comprehensive as, after reviewing discrepancies between previous studies (Methods), we identified all 119 expected lineages plus one new lineage.

The number of neurons per hemilineage can vary widely (Fig. 2h)—for example, counting both hemispheres, FLAa1 contains just 30 neurons whereas MBp4 (which makes the numerous Kenyon cells that are required for memory storage) has 1,335. However, in general, the number of neurons per hemilineage is between 60 and 282 (10th to 90th percentile, respectively). Nevertheless, the numbers of neurons within each hemilineage were highly reliable, differing only by 3% (±4%) between the left and right hemispheres (Fig. 2h). This is consistent with the near-equality of neurons per hemisphere noted in Fig. 1, and indicates great precision in the developmental programs controlling neuron number. We also identified neurons belonging to 125 hemilineages in the hemibrain dataset (Fig. 2j), a connectome comprising approximately half of a female fly brain[2] (Fig. 3a). The number of neurons per hemilineage strongly correlates across brains ($R^2 = 0.98$), with FlyWire hemilineages containing on average around 5% more neurons (Fig. 2k).

Although hemilineages typically contain functionally and morphologically related neurons, subgroups can be observed[37]. We further divided each hemilineage into distinct morphology groups, each innervating similar brain regions and taking similar internal tracts, using NBLAST morphological clustering[38] (Fig. 2i, Methods, Extended Data Fig. 3, Supplementary Files 3 and 4 and Supplementary Video 3). This generated a total of 528 groupings that are consistent across hemispheres and provide an additional layer of annotations between the hemilineage and cell type levels.

## Validating cell types across brains

We next sought to compare FlyWire against the hemibrain connectome[2]; this contains most of one central brain hemisphere and parts

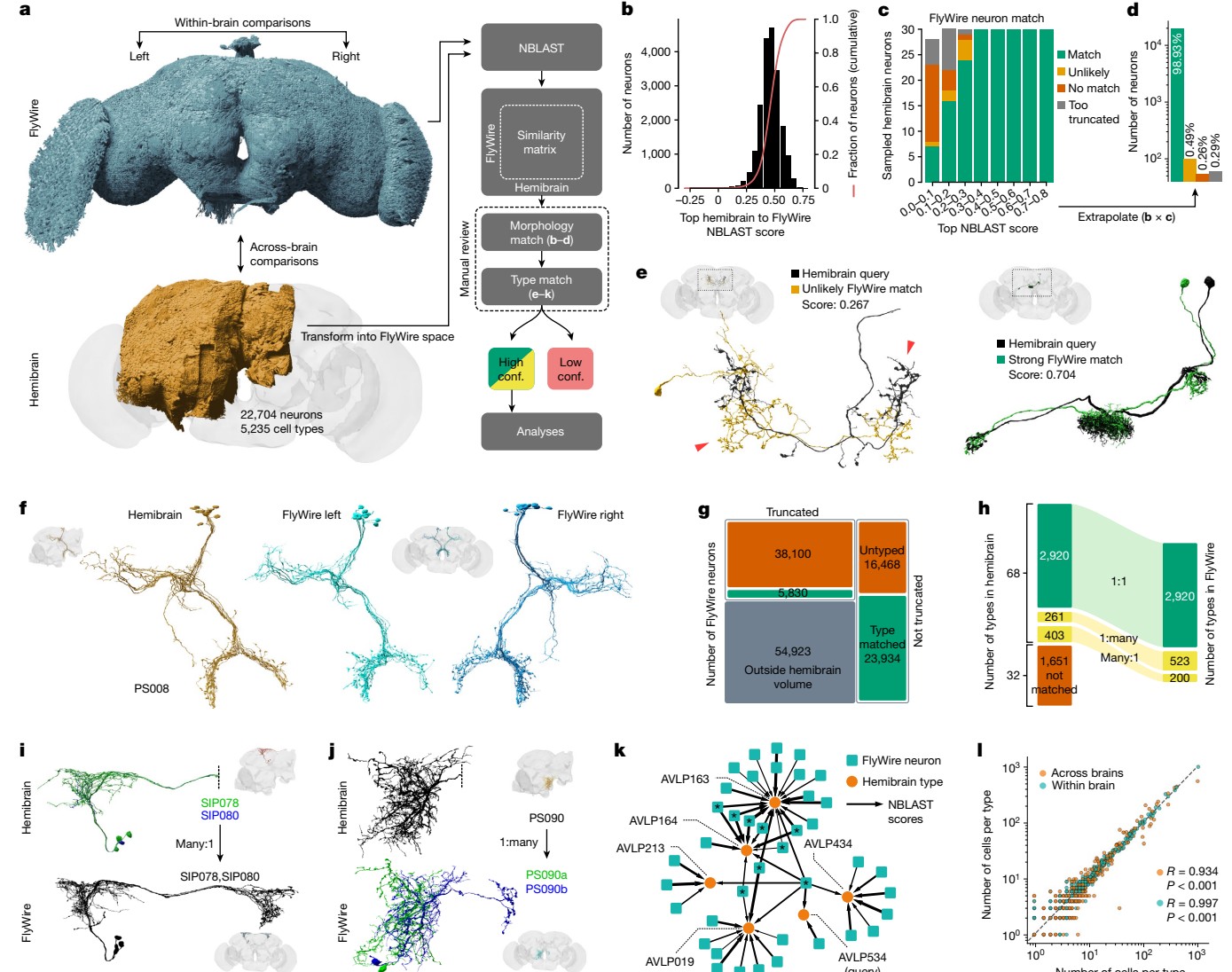

**Fig. 3 | Across-brain stereotypy. a**, Schematic of the pipeline for matching neurons between FlyWire and the Janelia hemibrain connectomes. Conf., confidence. **b**, The distribution of top hemibrain to FlyWire NBLAST scores. **c**, Manual review for a sample of top NBLAST hits. **d**, The extrapolated number of hemibrain neurons with matches in FlyWire. **e**, Example for unlikely (left) and strong (right) morphology match. **f**, Example of a high-confidence cell type (PS008) that is unambiguously identifiable across all three hemispheres. **g**, Counts of FlyWire neurons that were assigned a hemibrain type. **h**, The number of hemibrain cell types that were successfully identified and the resulting number of FlyWire cell types. **i,j**, Examples for many:1 (**i**) and 1:many (**j**) hemibrain type matches. The dotted vertical lines indicate truncation of the hemibrain neurons. **k**, Graph representation of top NBLAST hits between FlyWire neurons and hemibrain types. This subgraph contains nodes within a radius of three edges from the query cell type (AVLP534). Neurons matching multiple cell types (asterisks) must be manually resolved, which is not always possible. **l**, The number of cells per cross-matched cell type within a brain (FlyWire left versus right) and across brains (FlyWire versus hemibrain).

of the optic lobe. The hemibrain was previously densely cell typed by a combination of two automated procedures followed by extensive manual review[2,39–41]: NBLAST morphology clustering initially yielded 5,235 morphology types; multiple rounds of CBLAST connectivity clustering split some types, generating 640 connectivity types for a final total of 5,620 types. We have reidentified just 14% of connectivity types and therefore use the 5,235 morphology types as a baseline for comparison. Although 389 (7%) of the hemibrain cell types were previously established in the literature and recorded in the https://virtualflybrain.org/ database[22], principally through analysis of genetic driver lines[19], the great majority (90%) were newly proposed using the hemibrain, that is, derived from a single hemisphere of a single animal. This was reasonable given the pioneering nature of the hemibrain reconstruction, but the availability of the FlyWire connectome now allows for a more stringent re-examination.

We approach this by considering each cell type in the hemibrain as a prediction: if we can reidentify a distinct group of cells with the same properties in both hemispheres of the FlyWire dataset, then we conclude that a proposed hemibrain cell type has been tested and validated. To perform this validation, we first used non-rigid three-dimensional (3D) registration to map meshes and skeletons of all hemibrain neurons into FlyWire space, enabling direct co-visualization of both datasets and a range of automated analyses. We then used NBLAST[38] to calculate morphological similarity scores between all hemibrain neurons and the approximately 84,000 FlyWire neurons with arbours at least partially contained within the hemibrain volume (Fig. 3a,b and Extended Data Fig. 4a–c). We manually reviewed the top five NBLAST hits for a random sample of individual neuron-to-neuron matches and found that high NBLAST scores typically indicate a good morphological match (Fig. 3c). Extrapolating from this sample, we expect 99% of hemibrain

neurons to have a morphologically very similar neuron in FlyWire (Fig. 3d).

We next attempted to map hemibrain cell types onto FlyWire neurons. Candidate type matches were manually reviewed by co-visualization and only those with high confidence were accepted (Fig. 3f–h and Methods). Crucially, this initial morphological matching process generated a large corpus of shared cell type labels between datasets; with these in place, we developed an across-dataset connectivity clustering method that enabled us to investigate and resolve difficult cases (see the 'hemibrain cell type matching with connectivity' section of the Methods).

The majority of hemibrain cell types (56%; 2,920 out of 5,235 types) were unambiguously found in the FlyWire dataset (Fig. 3f). A further 664 (13%) hemibrain types were mapped but had to be either merged (many:1) or further split (1:many) (Fig. 3h). In total, 7% of proposed hemibrain types were combined to define new 'composite' types (for example, SIP078,SIP080) because the hemibrain split could not be recapitulated when examining neurons from both FlyWire and the hemibrain (Fig. 3i and Extended Data Fig. 4e–g). This is not too surprising as the hemibrain philosophy was explicitly to err on the side of splitting in cases of uncertainty[2]. We found that 5% of proposed hemibrain types needed to be split, for example, because truncation of neurons in the hemibrain removed a key defining feature (Fig. 3j). Together these revisions mean that the 3,584 reidentified hemibrain cell types map onto 3,643 consensus cell types (Fig. 3h). All revisions were confirmed by across-dataset connectivity clustering.

Notably, 1,651 (32%) hemibrain cell types could not be reidentified in FlyWire. Ambiguities due to hemibrain truncation can partially explain this: we were much more successful at matching neurons that were not truncated in the hemibrain (Fig. 3g). However, this appears not to be the main explanation. Especially in cases of multiple, very similar, 'adjacent' hemibrain types, we often encountered 'chains' of ambiguity that made assigning types difficult (Fig. 3k). Further investigation (Fig. 6) suggests that the majority of these unmatched hemibrain types are not exactly replicable across animals. Instead, we show that multi-connectome analysis can generate validated cell types that are robust to interindividual variation.

In conclusion, we validated 3,643 high-confidence consensus cell type labels for 43,737 neurons from three different hemispheres and two different brains (Fig. 3g). Collectively these cross-matched neurons cover 46.5% of central brain edges (comprising 49% of synapses) in the FlyWire graph. This body of high-confidence cross-identified neurons enables both within-brain (FlyWire left versus right hemisphere) and across-brain (FlyWire versus hemibrain) comparisons.

## Cell types are highly stereotyped

Using the consensus cell type labels, we found that the numbers of cells per type across the three hemispheres are closely correlated (Fig. 3l). About one in six cell types shows a difference in numbers between the left and right hemisphere and one in three across brains (FlyWire versus hemibrain). The mean difference in the number of cells per type is small though: 0.3 (±1.8) within brains and 0.8 (±10) across brains. Importantly, cell types with fewer neurons per type are less variable (Extended Data Fig. 4i,j). At the extreme, 'singleton' cell types account for 59% of all types in our sample; they often appear to be embryonic-born, or early secondary neurons, and only very rarely comprise more than one neuron—only 3% of neurons that are singletons in both FlyWire hemispheres have more than 1 member in the hemibrain. By contrast, more numerous cell types are also more likely to vary in number both within but even more so across brains (Extended Data Fig. 4i,j).

Synapse counts were also largely consistent within cell types, both within and across brains. To enable a fair comparison, the FlyWire synapse cloud was restricted to the smaller hemibrain volume. Although this does not correct for other potential confounds such as differences in the synaptic completion rates or synapse detection, pre- and post-synapse counts per cell type were highly correlated, both within brains (Pearson $R = 0.99$; $P < 0.001$) and across brains (Pearson $R = 0.92$ and 0.76 for pre- and post-synapses, respectively; $P < 0.001$; Fig. 4a,b and Extended Data Fig. 4k,l). This is an important quality control and pre-requisite for subsequent connectivity comparisons.

The fly brain is mostly left–right symmetric, but inspection of the FlyWire dataset revealed a small number of asymmetries. For example, LC6 and LC9 visual projection neurons form a large axon bundle that follows the normal path in the right hemisphere[42] but, in the left hemisphere, it loops over (that is, medial) the mushroom body peduncle; nevertheless, the axons still find their correct targets as previously reported[43]. We annotated other examples of this ranging from small additional/missing branches to misguided neurite bundles and found that only 0.4% of central brain neurons exhibit such biological oddities (Extended Data Fig. 5).

## Interpreting connectomes

Brain wiring develops through a complex and probabilistic developmental process[44,45]. To interpret the connectome, it is vital to obtain a basic understanding of how variable that biological process is. This is complicated by the fact that the connectome we observe is shaped not just by biological variability but also by technical noise, for example, from segmentation issues, synapse detection errors and synaptic completion rates (the fraction of synapses attached to proofread neurons) (Fig. 4a). Here we use the consensus cell types to assess which connections are reliably observed across three hemispheres of connectome data. We use the term 'edge' to describe the set of connections between two cell types, and its 'weight' as the number of unitary synapses (no threshold, that is, ≥1 synapses) forming that connection.

Weights of individual edges are highly correlated within (Pearson $R = 0.97$, $P < 0.001$) and across (Pearson $R = 0.8$, $P < 0.001$) brains (Fig. 4c and Extended Data Fig. 6a). Consistent with this, cell types exhibit highly similar connectivity within as well as across brains (Fig. 4d and Extended Data Fig. 6b,c). While the connectivity (cosine) similarity across brains is lower than within brains ($P < 0.001$), the effect size is small (0.045 ± 0.096) and is at least in part due to the aforementioned truncation in the hemibrain.

We next examined, for a given edge between two cell types in one hemisphere, the odds of finding the same connection in another hemisphere or brain. Examination of 572,980 edges present in at least one of the three brain hemispheres showed that 53% of the edges observed in the hemibrain were also found in FlyWire. This fraction is slightly higher when comparing between the two FlyWire hemispheres: left to right: 61%; right to left: 59% (Fig. 4e). Weaker edges were less likely to be consistent: an edge consisting of a single synapse in the hemibrain has a 42% chance to be also present in a single FlyWire hemisphere, and only a 16% chance to be seen in both hemispheres of FlyWire (Fig. 4f). By contrast, any edge of more than ten synapses in any hemisphere can be reproducibly (>90% of the time, rounded) found in the other two hemispheres. Although only 16% of all edges meet this threshold, they comprise around 79% of all synapses (Fig. 4g and Extended Data Fig. 6e). We also analysed normalized edge weights expressed as the fraction of the input onto each downstream neuron; this accounts for the small difference in synaptic completion rate between FlyWire and the hemibrain. With this treatment, the distributions are almost identical for within and across brain comparisons (Fig. 4g (compare the left and right panels)); edges constituting ≥0.9% of the target cell type's total inputs have a greater than 90% chance of persisting (Fig. 4g (right)). Around 7% of edges, collectively containing over half (54%) of all synapses, meet this threshold.

We observed that the fraction of edges persisting across datasets plateaued as the edge weight increased. Using a level of 99% edge persistence, we can define a second principled heuristic: edges greater than

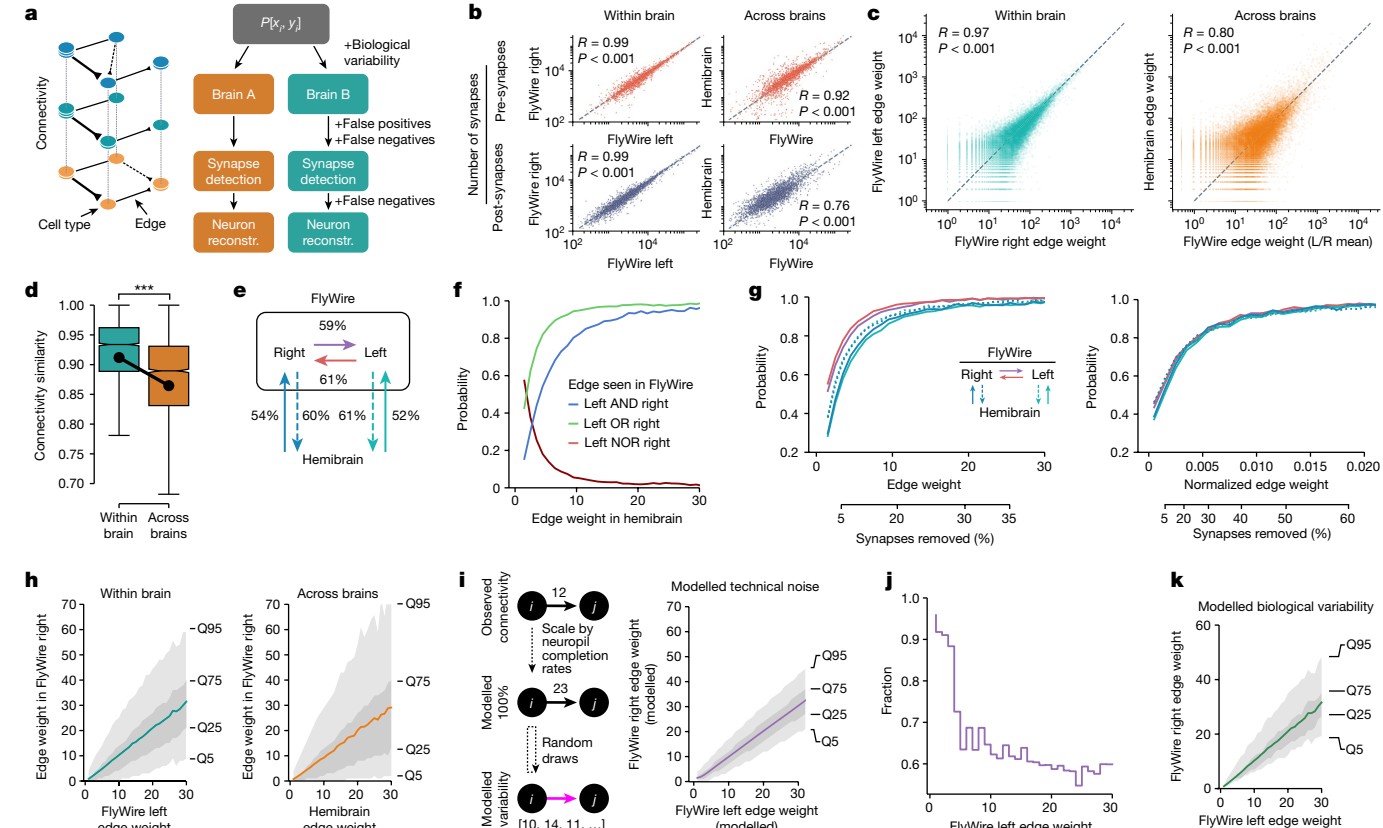

**Fig. 4 | Connectivity stereotypy. a**, Connectivity comparisons and potential sources of variability. Reconstr., reconstruction. **b**, The number of pre- and post-synapses per cross-matched cell type. **c**,**d**, Edge weights (**c**) and cosine connectivity similarity (**d**) between cross-matched cell types. The whiskers represent 1.5× the interquartile range. **e**, The percentage of edges in one hemisphere that can be found in another hemisphere. **f**, The probability that an edge present in the hemibrain is found in one, both or neither of the hemispheres in FlyWire. A plot with normalized edge weights is shown in Extended Data Fig. 6d. **g**, The probability that an edge is found within and across brains as a function of total (left) and normalized (right) edge weight. The second x axis shows the percentage of synapses below a given weight. **h**, Correlation of across-edge (left) and within-edge (right) edge weights.

The envelopes represent quantiles. **i**, Model for the impact of technical noise (synaptic completion rate, synapse detection) on synaptic weight from cell types *i* to *j*. The raw weight from the connectome for each individual edge is scaled up by the computed completion rate for all neurons within the relevant neuropil; random draws of the same fraction of those edges then allow an estimate of technical noise. **j**, Observed variability explainable by technical noise as fraction of FlyWire left–right edge pairs that fall within the 5–95% quantiles for the modelled technical noise. **k**, Modelled biological variability (observed variability − technical noise). *R* (**b** and **c**) is the Pearson correlation coefficient. For **d**, statistical analysis was performed using unpaired *t*-tests; ***P < 0.001.

2.6% edge weight (or 31 synapses) can be considered to be strong. Note that these statistics defined across the whole connectome can have exceptions in individual neurons. For example, descending neuron DNp42 receives 34 synapses from PLP146 in FlyWire right, but none on the left or hemibrain; this may well be an example of developmental noise (that is, bona fide biological variability, rather than technical noise).

So far, we have examined only the binary question of whether an edge exists or not. However, the conservation of edge weight is also highly relevant for interpreting connectomes. We next considered, given that an edge is present in two or more hemispheres, the odds that it will have a similar weight. Edge weights within and across brains are highly correlated (Fig. 4c), a 30-synapse edge in the hemibrain, for example, will on average consist of 29 synapses in FlyWire, despite differences in synaptic detection and completion rates for these two datasets imaged with different EM modalities[1]. The variance of edge weights is considerable though: 25% of all 30-synapse hemibrain edges will consist of fewer than 13 synapses in FlyWire, and 5% will consist of only 1–2 synapses. Consistency is greater when looking within FlyWire: a 30-synapse edge on the left will, on average, consist of 31 synapses on the right. Still, 25% of all 30-synapse edges on the left will consist of 21 synapses or less on the right, and 5% of only 1–8 synapses (Fig. 4h).

To assess how much of this edge weight variability is biological and how much is technical, we modelled the impact of technical noise on a fictive ground truth connectome (Fig. 4i and Methods). This model was randomly subsampled according to postsynaptic completion rate (in the mushroom body calyx, for example, there is a 6% difference between the left and right hemisphere of FlyWire; Extended Data Fig. 6f), and synapses were randomly added and deleted according to the false-positive and false-negative rates reported for the synapse detection[46]. Repeated application of this procedure generated a distribution of edge weights between each cell type pair expected due to technical noise. On average, 65% of the observed variability of edge weight between hemispheres fell within the range expected due to technical noise; this fraction approached 100% for weaker synapses (Fig. 4j). For example, cell type LHCENT3 targets LHAV3g2 with 30 synapses on the left but only 23 on the right of FlyWire, which is within the 5–95% quantiles expected due to technical noise alone. Overall, this analysis shows that observed variability (Fig. 4h (left)) is greater than can be accounted for by technical noise, establishing a lower bound for likely biological variability (Fig. 4k), and suggests another simple heuristic: differences in edge weights of 30% or less may be entirely due to technical noise and should not be overinterpreted.

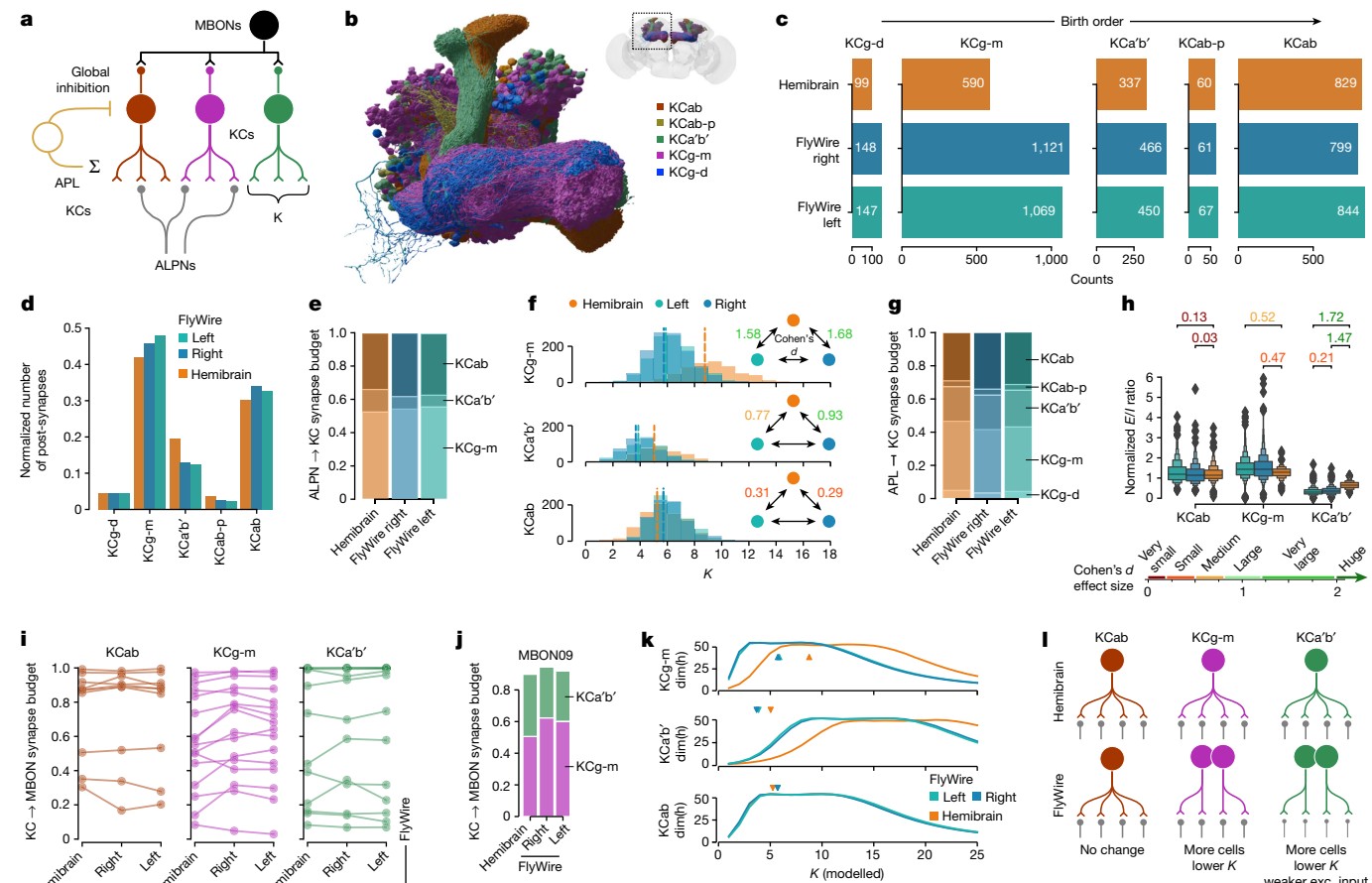

**Fig. 5 | Variability in the mushroom body. a**, Schematic of mushroom body circuits. *K* refers to the number of ALPN types that a KC samples from. Neuron types not shown are as follows: DANs, DPM and OANs. **b**, Rendering of KC types. **c**, Per-type KC counts across the three hemispheres. **d**, KC post-synapse counts, normalized to total KC post-synapses in each dataset. **e**, The fraction of ALPN to KC budget spent on individual KC types. **f**, The number of ALPN types a KC receives input from *K*. The dotted vertical lines represent the mean. **g**, The fraction of APL to KC budget spent on individual KC types. **h**, The normalized excitation/inhibition ratio for KCs. An explanation of enhanced box plots is provided in the Methods. **i**, The fraction of MBON input budget coming from KCs. Each line represents an MBON type. **j**, MBON09 as an example for KC to MBON connectivity. All MBONs are shown in Extended Data Fig. 7. **k**, Dimensionality (dim(h)) as function of a modelled *K*. The arrowheads mark observed mean *K* values. **l**, Summarizing schematic. Exc., excitatory. For **f** and **h**, Cohen's *d* effect size values are shown for pairwise comparisons where *P* ≤ 0.01; Welch's tests (**f**) and Kolmogorov–Smirnov tests (**h**) were applied.

## Variability in the mushroom body

The comprehensive annotation of cell types in the FlyWire dataset revealed that the number of Kenyon cells (KCs), the intrinsic neurons of the mushroom body, is 30% larger per hemisphere than in the hemibrain (2,597 KCs in FlyWire right; 2,580 in FlyWire left; and 1,917 in hemibrain), well above the average variation in cell counts (5 ± 12%). While these KC counts are within the previously reported range[47], the difference presents an opportunity to investigate how connectomes accommodate perturbations in cell count. The mushroom body contains five principal cell classes: KCs, mushroom body output neurons (MBONs), modulatory neurons (dopaminergic neurons (DANs) and octopaminergic neurons (OANs)), the dorsal paired medial (DPM) and anterior paired lateral neuron (APL) giant interneurons[48] (Fig. 5a). KCs further divide into five main cell types on the basis of which parts of the mushroom body they innervate: KCab, KCab-p, KCg-m, KCa′b′ and KCg-d (Fig. 5b). Of those, KCab, KCa′b′ and KCg-m are the primary recipients of largely random[39,49] (but see ref. 50) olfactory input through around 130 antennal lobe projection neurons (ALPNs) comprising 58 canonical types[39,40]. Global activity in the mushroom body is regulated through an inhibitory feedback loop mediated by APL, a single large GABAergic neuron[51]. Analogous to the mammalian cerebellum, KCs transform the dense overlapping odour responses of the early olfactory system into sparse non-overlapping representations that enable the animal to discriminate between individual odours during associative learning[52,53]. The difference in cell counts is not evenly distributed across all KC types: KCg-m (and to a lesser extent KCg-d and KCa′b′) are almost twice as numerous in FlyWire versus hemibrain while KCab and KCab-p are present in similar numbers (Fig. 5c). Protein starvation during the larval stage can induce specific increases in KCg-m number[54], suggesting that environmental variations in food resources may have contributed to this difference.

To examine how this affects the mushroom body circuitry, we opted to compare the fraction of the input or output synaptic budget across different KCs, as this is well matched to our question and naturally handles a range of technical noise issues that seemed particularly prominent in the mushroom body completion rate (Methods and Extended Data Fig. 7a). We found that, despite the large difference in KCg-m cell counts between FlyWire and hemibrain, this cell type consistently makes and receives 32% and 45% of all KC pre-synapses and post-synapses, respectively (Fig. 5d and Extended Data Fig. 7e). This suggested that individual FlyWire KCg-m neurons receive fewer inputs and make fewer outputs than their hemibrain counterparts. The share of ALPN outputs allocated to KCg-m is around 55% across all hemispheres (Fig. 5e), and the average ALPN to KCg-m connection is comparable in strength across hemispheres (Extended Data Fig. 7f); however, each

KCg-m neuron receives input from a much smaller number of ALPN types in FlyWire than in the hemibrain (5.74, 5.89 and 8.76 for FlyWire left, right and hemibrain, respectively; Fig. 5f). FlyWire KCg-m neurons therefore receive inputs with the same strength but from fewer ALPNs.

This pattern holds for other KCg-m synaptic partners as well. Similar to the excitatory ALPNs, the share of APL outputs allocated to KCg-m neurons is essentially constant across hemispheres (Fig. 5g). Thus, each individual KCg-m neuron receives proportionally less inhibition from the APL, as well as less excitation, maintaining a similar excitation/inhibition ratio (Fig. 5h). Furthermore, as a population, KCg-m neurons contribute similar amounts of input to MBONs (Fig. 5i,j and Extended Data Fig. 7h).

Past theoretical work has shown that the number ($K$) of discrete odour channels (that is, ALPN types) providing input to each KC has an optimal value for maximizing dimensionality of KC activity and, therefore, discriminability of olfactory input[52,53]. The smaller value for $K$ observed for KCg-m neurons in the FlyWire connectome (Fig. 5g) raises the question of how dimensionality varies with $K$ for each of the KC types. Using the neural network rate model described previously[52], we calculated dimensionality as a function of $K$ for each of the KC types, using the observed KC counts, ALPN to KC connectivity and global inhibition from the APL. This analysis revealed that optimal values for $K$ are lower for KCg-m neurons in FlyWire than in the hemibrain (Fig. 5k), consistent with the observed values.

Taken together, these results demonstrate that, for KCg-m neurons, the brain compensates for a developmental perturbation by changing a single parameter: the number of odour channels each KC samples from. By contrast, KCa′b′ cells, which are also more numerous in FlyWire than in the hemibrain, appear to use a hybrid strategy of reduced $K$ combined with a reduction in ALPN to KCa′b′ connection strength (Extended Data Fig. 7f). These findings contradict earlier studies in which a global increase in KC numbers through genetic manipulation triggered an increase in ALPN axon boutons (indicating an compensatory increase in excitatory drive to KCs) and a modest increase in KC claws (suggesting an increase rather than decrease in $K$)[55,56]. This may be due to the differences in the nature and timing of the perturbation in KC cell number, and the KC types affected.

## Toward multiconnectome cell typing

As the first dense, large-scale connectome of a fly brain, the hemibrain dataset proposed over 5,000 previously unknown cell types in addition to confirming around 400 previously reported types recorded in the http://virtualflybrain.org/ database[22]. As this defines a de facto standard cell typing for large parts of the fly brain, our initial work plan was simply to reidentify hemibrain cell types in FlyWire, providing a critical resource for the fly neuroscience community. While this was successful for 68% of hemibrain cell types (Fig. 3), 32% could not be validated. Given the great stereotypy generally exhibited by the fly nervous system, this result is both surprising and interesting.

We can imagine two basic categories of explanation. First, that through ever closer inspection, we may successfully reidentify these missing cell types. Second, that these definitions, mostly based on a single brain hemisphere, might not be robust to variation across individuals. Distinguishing between these two explanations is not at all straightforward. We began by applying across-dataset connectivity clustering to large groups of unmatched hemibrain and FlyWire neurons. We observed that most remaining hemibrain types showed complex clustering patterns, which both separated neurons from the same proposed cell type and recombined neurons of different proposed hemibrain types.

While it is always more difficult to prove a negative result, these observations strongly suggest that the majority of the remaining 1,696 hemibrain types are not robust to interindividual variation. We

therefore developed a definition of cell type that uses interanimal variability: a cell type is a group of neurons that are each more similar to a group of neurons in another brain than to any other neuron in the same brain. This definition can be used with different similarity metrics but, for connectomics data, a similarity measure incorporating morphology and/or connectivity is most useful. Our algorithmic implementation of this definition operates on the co-clustering dendrogram by finding the smallest possible clusters that satisfy two criteria (Fig. 6a): (1) each cluster must contain neurons from all three hemispheres (hemibrain, FlyWire right and FlyWire left); (2) within each cluster, the number of neurons from each hemisphere must be approximately equal.

Determining how to cut a dendrogram generated by data clustering is a widespread challenge in data science for which there is no single satisfactory solution. A key advantage of the cell type definition that we propose is that it provides very strong guidance about how to assign neurons to clusters. This follows naturally from the fact that connectome data provide us with all neurons in each dataset, rather than a random subsample. This advantage of completeness is familiar from analogous problems such as the ability to identify orthologous genes when whole genomes are available[57].

Analysis of the hemibrain cell type AOTU063 provides a relatively straightforward example of our approach (Fig. 6b and Extended Data Fig. 10). Morphology-based clustering generates a single group, comprising all six AOTU063 neurons from each of the three hemispheres. However, clustering based on connectivity reveals two discrete groups, with equal numbers of neurons from each hemisphere, suggesting that this type should be split further. Here, algorithmic analysis across multiple connectomes reveals consistent connectivity differences between subsets of AOTU063 neurons.

To test whether this approach is applicable to more challenging sets of neurons, we set aside the hemibrain types and performed a complete retyping of neurons in the central complex (Fig. 6c), a centre for navigation in the insect brain that has been subject to detailed connectome analysis[41]. We selected two large groups of neurons innervating the fan-shaped body (FB) that show a key difference in organization. The first group, FC1–3 (357 neurons in total), consists of columnar cell types that tile the FB innervating adjacent non-overlapping columns. The second group, FB1–9 (897 neurons in total), contains tangential neurons where neurons of the same cell type are precisely co-located in space[41] (Fig. 6d). Standard NBLAST similarity assumes that neurons of the same cell type overlap closely in space; although this is true for most central brain types, it does not hold for repeated columnar neurons such as those in the optic lobe or these FC neurons of the FB. We therefore used a connectivity-only distance metric co-clustering across the three hemispheres. This resulted in seven FC clusters satisfying the above criteria (Fig. 6e,f). Five of these cross-brain types have a one-to-one correspondence with hemibrain types, while two are merges of multiple hemibrain types; only a small number of neurons are recombined across types (Fig. 6g). For the second group, FB1–9, a combined morphology and connectivity embedding was used. Co-clustering across the three hemispheres generated 114 cell types compared to 146 cell types in the hemibrain (Fig. 6h and Extended Data Fig. 8). In total, 44% of these types correspond one-to-one to a hemibrain cell type; 11% are splits (1:many), 12% are merges (many:1) and 33% are recombinations (many:many) of hemibrain cell types. The 67% (44 + 11 + 12) success rate of this de novo approach in identifying hemibrain cell types is slightly higher than the 61% achieved in our directed work in Fig. 3; it is consistent with the notion that further effort could still identify some unmatched hemibrain types, but that the majority will probably require retyping.

All of the preceding efforts have focused on cell typing neurons contained within both FlyWire and the hemibrain. We next examined the extensive regions of the brain covered only by FlyWire and not by hemibrain. Based on the lessons learned from the joint analysis of

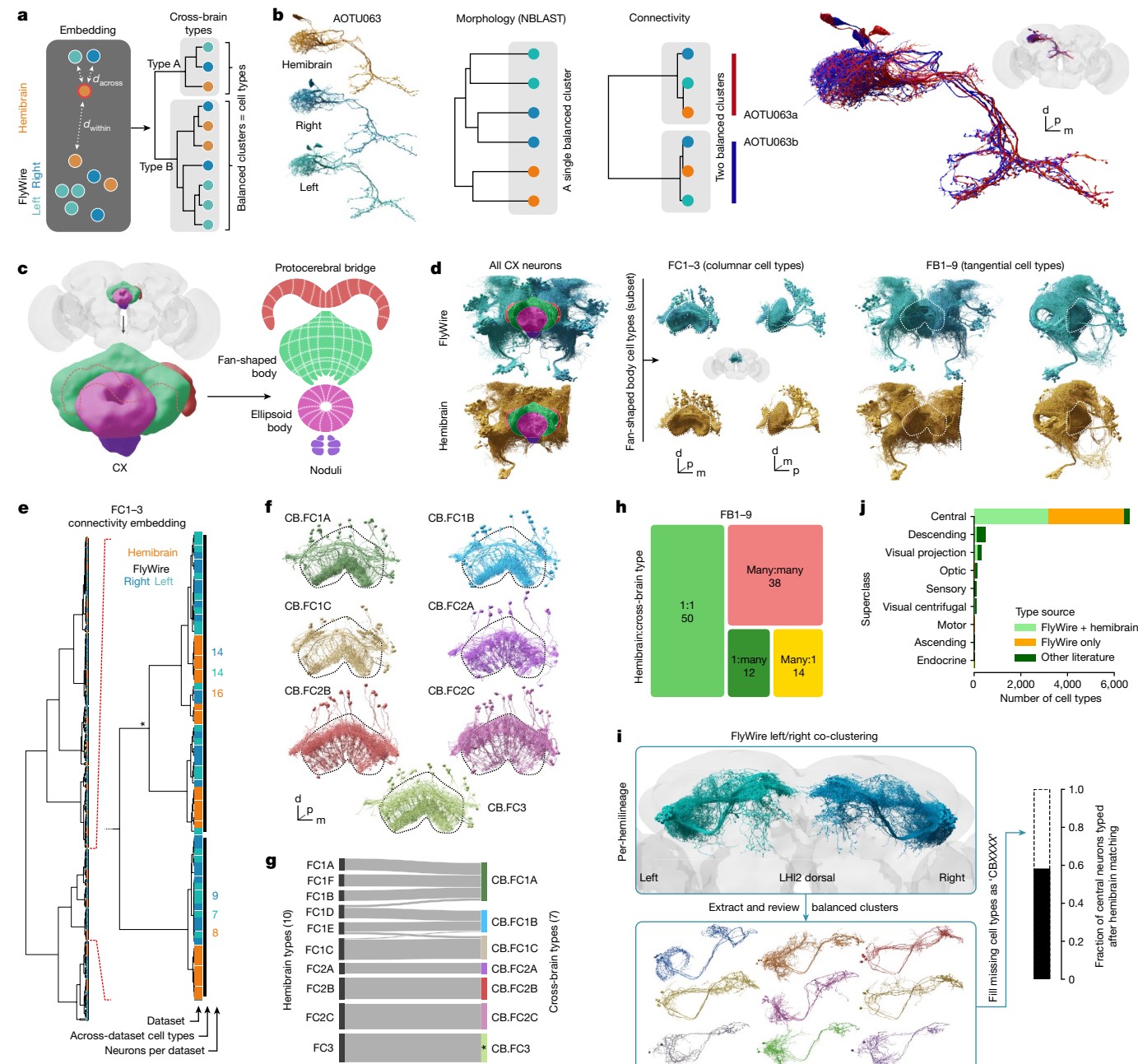

**Fig. 6 | Across-brain cell typing. a**, Cell type is defined as a group of neurons that are each more similar to a group in another brain than to any neurons in the same brain. We expect cell type clusters to be balanced, that is, contain neurons from all three hemispheres in approximately even numbers. **b**, Example of a hemibrain cell type (AOTU063) that is morphologically homogeneous but has two cross-brain consistent connectivity types and can therefore be split. **c**, Main neuropils making up the central complex (CX). **d**, Overview of all CX cells (left) and two subsets of fan-shaped body (FB, dotted outlines) cell types: FC1–3 and FB1–9 (right). **e**, Hierarchical clustering from connectivity embedding for FC1–3 cells. A magnification of cross-brain cell type clusters is

shown. The asterisk marks a cluster that was manually adjusted. **f**, Renderings of FC1–3 across-brain types; the FB is outlined. The tiling of FC1–3 neurons can be discerned. **g**, Comparison of FC1–3 hemibrain and cross-brain cell types. The colours correspond to those in **f**. **h**, Mappings between hemibrain and cross-brain cell types for FB1–9. A detailed flow chart is provided in Extended Data Fig. 8. **i**, The pipeline for generating types for neurons without a hemibrain cell type. Hemilineage LHl2 dorsal is shown as an example. The box plot shows the fraction of FlyWire neurons with a hemibrain-derived cell type. **j**, Cell type source broken down by super class.

hemibrain and FlyWire, we ran a co-clustering of neurons from the two hemispheres of FlyWire to fill in missing cell types (Fig. 6i,j and Extended Data Fig. 9). This combined both morphology and connectivity measures, was carried out separately for each hemilineage and produced 3,200 new central brain cell types for a total of 8,453 including the optic lobes. We further compared double-hemisphere (FlyWire left/right) and triple-hemisphere analysis (FlyWire + hemibrain) for 25 cross-identified lineages that are not truncated in the hemibrain. This

comparison found that 70% of these new types survive addition of a third hemisphere with minor edits (1:many, many:1). That percentage increases to 84% if we exclude cases in which just one neuron changes clusters (Extended Data Fig. 9).

In summary, cell typing based on joint analysis of multiple connectomes proved capable of recapitulating many cell types identified in the hemibrain dataset, while also defining new candidate cell types that are consistent both within and across datasets. Further validation

of the new types proposed by this approach will depend on additional *Drosophila* connectomes, which are forthcoming. We predict that cell types defined in this manner will be substantially more robust than cell types defined from a single connectome alone.

## Discussion

Here we generated human-readable annotations for all neurons in the fly brain at various levels of granularity: superclass, cell class, hemilineage, morphology group and cell type. These annotations provide salient groupings that have already been proven to be useful not only in our own analyses, but also in many of those in our companion paper[1] as well as other publications in the FlyWire paper package introduced there, and to researchers now using the online platforms Codex (https://codex.flywire.ai) and FAFB-FlyWire CATMAID spaces (https://fafb-flywire.catmaid.org). Hemilineage annotations also provide a key starting point to link the molecular basis of the development of the central brain to the wiring revealed by the connectome; such work has already begun in the more repetitive circuits of the optic lobe[58].

The cell type atlas that we provide of 8,453 cell types, covering 96.4% of all neurons in the brain, is to our knowledge the largest ever proposed (the hemibrain had 5,235) and, crucially, by some margin the largest ever validated collection of cell types[19]. In *C. elegans*, the 118 cell types inferred from the original connectome have been clearly supported by analysis of subsequent connectomes and molecular data[3,59,60]. In a few cases in mammals, it has been possible to produce catalogues of order 100 cell types that have been validated by multimodal data, for example, in the retina or motor cortex[20,61]. Although large scale molecular atlases in the mouse produce highly informative hierarchies of up to 5,000 clusters[62–64], they do not yet try to define terminal cell types—the finest unit that is robust across individuals—with precision. Here we tested over 5,000 predicted cell types, resulting in 3,884 validated cell types using three hemispheres of connectome data. Informed by this, we use the FlyWire dataset to propose an additional 3,685 cell types.

### Lessons for cell typing

Our experience of cell typing the FlyWire dataset together with our earlier participation in the hemibrain cell typing effort leads us to draw a number of lessons. First, we think that it is helpful to frame cell types generated in one dataset as predictions or hypotheses that can be tested either through additional connectome data or data from other modalities. Related to this, although the two hemispheres of the same brain can be treated as two largely independent datasets, we do see evidence that variability can be correlated across hemispheres (Fig. 4). We therefore recommend the use of three or more hemispheres to define and validate new cell types both because of increased statistical confidence and because across-brain comparisons are a strong test of cell type robustness. Third, there is no free lunch in the classic lumping versus splitting debate. The hemibrain cell typing effort preferred to split rather than lump cell types, reasoning that over-splitting could easily be remedied by merging cell types at a later date[2]. Although this approach seemed reasonable at the time, it appears to have led to cell types being recombined: when using a single dataset, even domain experts may find it very hard to distinguish conserved differences between cell types from interindividual noise. Moreover, although some recent studies have argued that cell types are better defined by connectivity than morphology, we find that there is a place for both. For de novo cell typing of future connectomes, we recommend an initial morphology-only matching to assign obvious matches; these shared cell type labels can then be used to define connection similarity across datasets. This then allows extraction of balanced clusters from combined morphology and connectivity co-clustering that can be used to assign or refine existing cell types.

Related to this, we find that across-dataset connection similarity is an extremely powerful way to identify cell types. However, connectivity-based typing is typically used iteratively and especially when used within a single dataset this may lead to selection of idiosyncratic features. Moreover, neurons can connect similarly but come from a different developmental lineage, or express a different neurotransmitter, precluding them from sharing a cell type. Combining these two points, we would summarize that matching by morphology appears to be both more robust and sometimes less precise, whereas connectivity matching is a powerful tool that must be wielded with care.

In conclusion, connectome data are particularly suitable for cell typing: they are inherently multimodal (by providing morphology and connectivity), while the ability to see all cells within a brain (completeness) is uniquely powerful. Our multiconnectome typing approach (Fig. 6) provides a robust and efficient way to use such data; cell types that have passed the rigorous test of across-connectome consistency are very unlikely to be revised (permanence). We suspect that connectome data will become the gold standard for cell typing. Linking molecular and connectomics cell types will therefore be key. One promising new approach is exemplified by the prediction of neurotransmitter identity directly from EM images[21] but many others will be necessary.

Finally, we address the three questions introduced in the introduction.

### Can we simplify the connectome graph?

Cell typing reduces the complexity of the connectome graph. This has important implications for analysis, modelling, experimental work and developmental biology. For example, we can reduce the 131,811 typed nodes in the raw connectome graph into a cell type graph with 8,453 nodes; the number of edges is similarly reduced. This should significantly aid human reasoning about the connectome. It will also make numerous network analyses possible as well as substantially reduce the degrees of freedom in brain scale modelling[65,66]. It is important to note that, while collapsing multiple cells for a given cell type into a single node is often desirable, other use cases such as modelling studies may still need to retain each individual cell. However, if key parameters are determined on a per cell type basis, then the complexity of the resultant model can be much reduced. A recent study[65] optimized and analysed a highly successful model of large parts of the fly visual system with just 734 free parameters by using connectomic cell types.

For *Drosophila* experimentalists using the connectome, cell typing identifies groups of cells that probably form functional units. Most of these are linked though http://virtualflybrain.org/ to the published literature and in many cases to molecular reagents. Others will be more easily identified for targeted labelling and manipulation after typing. Finally, cell typing effectively compresses the connectome, reducing the bits required to store and specify the graph. For a fly-sized connectome, this is no longer that important for computational analysis, but it may be important for brain development. Some[67] have argued that evolution has selected highly structured brain connectivity enabling animals to learn very rapidly, but that these wiring diagrams are far too complex to be specified explicitly in the genome; rather, they must be compressed through a 'genomic bottleneck', which may itself have been a crucial part of evolving robust and efficient nervous systems. If we accept this argument, lossy compression based on aggregating nodes with similar cell type labels, approximately specifying strong edges and largely ignoring weak edges would reduce the storage requirements by orders of magnitude and could be a specific implementation of this bottleneck.

### Which edges are important?

The question of which of the 15.1 million edges in the connectome to pay attention to is critical for its interpretation. Intuitively, we assume that the more synapses that connect two neurons, the more important that connection must be. There is some very limited evidence in support of this assumption correlating anatomical and functional connectivity[68,69] (compare in mammals[70]). In lieu of physiological data, we postulate that edges that are critical to brain function should be

consistently found across brains. By comparing connections between cell types identified in three hemispheres, we find that edges stronger than ten synapses or ≥0.9% of the target's inputs have a greater than 90% chance to be preserved (Fig. 4f). This provides a simple heuristic for determining which edges are likely to be functionally relevant. It is also highly consistent with findings from the larval connectome, in which left–right asymmetries in connectivity vanish after removing edges weaker than <1.25% (ref. 71). However, note that edges falling below the threshold might still significantly contribute to the brain's function.

We further address an issue that has received little attention (but see ref. 72): the impact of technical factors (such as segmentation, proofreading, synapse detection) and biological variability on the final connectome and how to compensate for it. In our hands, a model of technical noise could explain up to 30% difference in edge weights. While this model was made specifically for the two hemispheres of FlyWire, it highlights the general point that a firm understanding of all sources of variability will be vital for the young field of comparative connectomics to distinguish real and artificial differences.

## Have we collected a snowflake?

The field of connectomics has long been criticized for unavoidably low $n$[73,74], raising the question of whether the brain of a single specimen is representative for all. For insects, there is a large body of evidence for morphological and functional stereotypy, although this information is available for only a minority of neurons and much less is known about stereotyped connectivity[19,75,76]. For vertebrate brains, the situation is less clear again; it is generally assumed that subcortical regions will be more stereotyped, but cortex also has conserved canonical microcircuits[77] and recent evidence has shown that some cortical elements can be genetically and functionally stereotyped[78]. Given how critical stereotypy is for connectomics, it is important to check whether that premise actually still holds true at the synaptic resolution.

For the fly connectome, the answer to our question is actually both more nuanced and more interesting than we initially imagined. Based on conservation of edges between FlyWire and hemibrain hemispheres, over 50% of the connectome graph is a snowflake. Of course, these non-reproducible edges are mostly weak. Our criterion for strong (highly reliable) edges applies to between 7–16% of edges but 50–70% of synapses.

We previously showed that the early olfactory system of the fly is highly stereotyped in both neuronal number and connectivity[40]. That study used the same EM datasets—FAFB and the hemibrain—but was limited in scope as only manual reconstruction in FAFB was then available. We now analyse brain-wide data from two brains (FlyWire and the hemibrain) and three hemispheres to address this question and find a high degree of stereotypy at every level: neuron counts are highly consistent between brains, as are connections above a certain weight. However, when examining so many neurons in a brain, we can see that cell counts are very different for some neurons; furthermore, neurons occasionally do something unexpected (take a different route or make an extra branch on one side of the brain). In fact, we hypothesize that such stochastic differences are unnoticed variability present in most brains; this is reminiscent of the observation that most humans carry multiple significant genetic mutations. We did observe one example of a substantial biological difference that was consistent across hemispheres but not brains: the number of the KCg-m neurons in the mushroom bodies is almost twice as numerous in FlyWire than in the hemibrain. Notably, we found evidence that the brain compensates for this perturbation by modifying connectivity (Fig. 5).

In conclusion, we have not collected a snowflake. The core FlyWire connectome is highly conserved and the accompanying annotations will be broadly useful across all studies of *D. melanogaster*. However, our analyses show the importance of calibrating our understanding of

biological (and technical) variability—as has recently been done across animals in *C. elegans*[60] and across hemispheres in larval *Drosophila*[71,79]. This will be crucial when using future connectomes to identify true biological differences, for example, in sexually dimorphic circuits or changes due to learning.

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

**FlyWire Consortium**

Krzysztof Kruk[10], Doug Bland[6], Zairene Lenizo[13], Austin T. Burke[6], Kyle Patrick Willie[6], Alexander S. Bates[1,3,4], Nikitas Serafetinidis[10], Nashra Hadjerol[13], Ryan Willie[6], Katharina Eichler[2], Ben Silverman[6], John Anthony Ocho[13], Joshua Bañez[13], Yijie Yin[2], Rey Adrian Candilada[13], Sven Dorkenwald[5,6], Jay Gager[6], Anne Kristiansen[10], Nelsie Panes[13], Arti Yadav[15], Remer Tancontian[13], Shirleyjoy Serona[13], Jet Ivan Dolorosa[13], Kendrick Joules Vinson[13], Dustin Garner[16], Regine Salem[13], Ariel Dagohoy[13], Philipp Schlegel[1,2], Jaime Skelton[10], Mendell Lopez[13], Laia Serratosa Capdevila[2], Griffin Badalamente[2], Thomas Stocks[10], Anjali Pandey[15], Darrel Jay Akiatan[13], James Hebditch[6], Celia David[6], Dharini Sapkal[15], Shaina Mae Monungolh[13], Varun Sane[2], Mark Lloyd Pielago[13], Miguel Albero[13], Jacquilyn Laude[13], Márcia dos Santos[2], David Deutsch[6,17], Zeba Vohra[15], Kaiyu Wang[18], Allien Mae Gogo[13], Emil Kind[19], Alvin Josh Mandahay[13], Chereb Martinez[13], John David Asis[13], Chitra Nair[15], Dhwani Patel[15], Marchan Manaytay[13], Imaan F. M. Tamimi[2], Clyde Angelo Lim[13], Philip Lenard Ampo[13], Michelle Darapan Pantujan[13], Alexandre Javier[13], Daril Bautista[13], Rashmita Rana[15], Jansen Seguido[13], Bhargavi Parmar[15], John Clyde Saguimpa[13], Merlin Moore[15], Markus W. Pleijzier[1], Mark Larson[20], Joseph Hsu[14], Itisha Joshi[15], Dhara Kakadiya[15], Amalia Braun[21], Cathy Pilapil[13], Marina Gkantia[2], Kaushik Parmar[15], Quinn Vanderbeck[14], Claire E. McKellar[6], Irene Salgarella[2], Christopher Dunne[2], Eva Munnelly[2], Chan Hyuk Kang[22], Lena Lörsch[23], Jinmook Lee[22], Lucia Kmecova[24], Gizem Sancer[25], Christa Baker[6], Szi-chieh Yu[6], Jenna Joroff[14], Steven Calle[24], Yashvi Patel[15], Olivia Sato[20], Siqi Fang[2], Janice Salocot[13], Farzaan Salman[26], Sebastian Molina-Obando[23], Paul Brooks[2], Mai Bui[27], Matthew Lichtenberger[10], Edmark Tamboboy[13], Katie Molloy[20], Alexis E. Santana-Cruz[24], Anthony Hernandez[13], Seongbong Yu[2], Marissa Sorek[6,10], Arzoo Diwan[15], Monika Patel[15], Travis R. Aiken[10], Sarah Morejohn[6], Sanna Koskela[18], Tansy Yang[18], Daniel Lehmann[10], Jonas Chojetzki[23], Sangeeta Sisodiya[15], Selden Koolman[6], Philip K. Shiu[28], Sky Cho[27], Annika Bast[23], Brian Reicher[20], Marlon Blanquart[2], Lucy Houghton[16], Hyungjun Choi[22], Maria Ioannidou[23], Matt Collie[6], Joanna Eckhardt[6], Benjamin Gorko[16], Li Guo[16], Zhihao Zheng[6], Alisa Poh[29], Marina Lin[27], István Taisz[1], Wes Murfin[30], Álvaro Sanz Díez[31], Nils Reinhard[32], Peter Gibb[14], Nidhi Patel[15], Sandeep Kumar[6], Minsik Yun[33], Megan Wang[6], Devon Jones[6], Lucas Encarnacion-Rivera[34], Annalena Oswald[23], Akanksha Jadia[15], Mert Erginkaya[35], Nik Drummond[2], Leonie Walter[19], Ibrahim Tastekin[35], Xin Zhong[19], Yuta Mabuchi[36], Fernando J. Figueroa Santiago[24], Urja Verma[15], Nick Byrne[20], Edda Kunze[19], Thomas Crahan[16], Ryan Margossian[10], Haein Kim[36], Iliyan Georgiev[10], Fabianna Szorenyi[24], Atsuko Adachi[31], Benjamin Bargeron[37], Tomke Stürner[1,2], Damian Demarest[38], Burak Gür[23], Andrea N. Becker[10], Robert Turnbull[2], Ashley Morren[10], Andrea Sandoval[28], Anthony Moreno-Sanchez[39], Diego A. Pacheco[14], Eleni Samara[21], Haley Croke[39], Alexander Thomson[18], Connor Laughland[18], Suchetana B. Dutta[19], Paula Guiomar Alarcón de Antón[19], Binglin Huang[16], Patricia Pujols[24], Isabel Haber[8], Amanda González-Segarra[28], Albert Lin[6,40], Daniel T. Choe[41], Veronika Lukyanova[42], Marta Costa[2], Nino Mancini[37], Zequan Liu[43], Tatsuo Okubo[14], Miriam A. Flynn[18], Gianna Vitelli[37], Meghan Laturney[28], Feng Li[8], Shuo Cao[44], Carolina Manyari-Diaz[37], Hyunsoo Yim[41], Anh Duc Le[39], Kate Maier[37], Seungyun Yu[22], Yeonju Nam[22], Daniel Bąba[10], Amanda Abusaif[28], Audrey Francis[45], Jesse Gayk[15], Sommer S. Huntress[46], Raquel Barajas[35], Mindy Kim[20], Xinyue Cui[36], Amy R. Sterling[6,10], Gabriella R. Sterne[28], Anna Li[14], Keehyun Park[22], Georgia Dempsey[2], Alan Mathew[13], Jinseong Kim[22], Taewan Kim[22], Guan-ting Wu[47], Serene Dhawan[48], Margarida Brotas[35], Cheng-hao Zhang[47], Shanice Bailey[2], Alexander Del Toro[28], Arie Matsliah[6], Kisuk Lee[6,49], Thomas Macrina[5,6], Casey Schneider-Mizell[50], Sergiy Popovych[5,6], Oluwaseun Ogedengbe[6], Runzhe Yang[6], Akhilesh Halageri[6], Will Silversmith[6], Stephan Gerhard[51], Andrew Champion[1,2], Nils Eckstein[18], Dodam Ih[6], Nico Kemnitz[6], Manuel Castro[6], Zhen Jia[6], Jingpeng Wu[6], Eric Mitchell[6], Barak Nehoran[5,6], Shang Mu[6], J. Alexander Bae[6,52], Ran Lu[6], Eric Perlman[8], Ryan Morey[6], Kai Kuehner[6], Derrick Brittain[50], Chris S. Jordan[6], David J. Anderson[44], Rudy Behnia[31], Salil S. Bidaye[37], Davi D. Bock[12], Alexander Borst[21], Eugenia Chiappe[35], Forrest Collman[50], Kenneth J. Colodner[48], Andrew Dacks[26], Barry Dickson[18], Jan Funke[18], Denise Garcia[39], Stefanie Hampel[24], Volker Hartenstein[11],

Bassem Hassan[19], Charlotte Helfrich-Forster[32], Wolf Huetteroth[53], Gregory S. X. E. Jefferis[1,2], Jinseop Kim[22], Sung Soo Kim[16], Young-Joon Kim[33], Jae Young Kwon[22], Wei-Chung Lee[14], Gerit A. Linneweber[19], Gaby Maimon[45], Richard Mann[31], Mala Murthy[6], Stéphane Noselli[54], Michael Pankratz[38], Lucia Prieto-Godino[48], Jenny Read[42], Michael Reiser[18], Katie von Reyn[39], Carlos Ribeiro[35], Kristin Scott[28], Andrew M. Seeds[24], Mareike Selcho[53], H. Sebastian Seung[5,6], Marion Silies[23], Julie Simpson[16], Scott Waddell[55], Mathias F. Wernet[19], Rachel I. Wilson[14], Fred W. Wolf[56], Zepeng Yao[57], Nilay Yapici[36] & Meet Zandawala[32]

[13]SixEleven, Davao City, Philippines. [14]Harvard Medical School, Boston, MA, USA. [15]ariadne.ai, Buchrain, Switzerland. [16]University of California, Santa Barbara, Santa Barbara, CA, USA. [17]Department of Neurobiology, University of Haifa, Haifa, Israel. [18]Janelia Research Campus, Howard Hughes Medical Institute, Ashburn, VA, USA. [19]Freie Universität Berlin, Berlin, Germany. [20]Harvard, Boston, MA, USA. [21]Department Circuits-Computation-Models, Max Planck Institute for Biological Intelligence, Planegg, Germany. [22]Sungkyunkwan University, Seoul, South Korea. [23]Johannes-Gutenberg University Mainz, Mainz, Germany. [24]Institute of Neurobiology, University of Puerto Rico Medical Sciences Campus, San Juan, Puerto Rico. [25]Department of Neuroscience, Yale University, New Haven, CT, USA. [26]Department of Biology, West Virginia University, Morgantown, WV, USA. [27]Program in Neuroscience and Behavior, Mount Holyoke College, South Hadley, MA, USA. [28]University of California, Berkeley, Berkeley, CA, USA. [29]University of Queensland, Brisbane, Queensland, Australia. [30]Independent consultant, Fort Collins, CO, USA. [31]Zuckerman Institute, Columbia University, New York, NY, USA. [32]Julius-Maximilians-Universität Würzburg, Würzburg, Germany. [33]Gwangju Institute of Science and Technology, Gwangju, South Korea. [34]Stanford University School of Medicine, Stanford, CA, USA. [35]Champalimaud Foundation, Lisbon, Portugal. [36]Cornell University, Ithaca, NY, USA. [37]Max Planck Florida Institute for Neuroscience, Jupiter, FL, USA. [38]University of Bonn, Bonn, Germany. [39]Drexel, Philadelphia, PA, USA. [40]Center for the Physics of Biological Function, Princeton University, Princeton, NJ, USA. [41]Seoul National University, Seoul, South Korea. [42]Newcastle University, Newcastle, UK. [43]RWTH Aachen University, Aachen, Germany. [44]Caltech, Pasadena, CA, USA. [45]Rockefeller University, New York, NY, USA. [46]Mount Holyoke College, South Hadley, MA, USA. [47]National Hualien Senior High School, Hualien, Taiwan. [48]The Francis Crick Institute, London, UK. [49]Brain & Cognitive Sciences Department, Massachusetts Institute of Technology, Cambridge, MA, USA. [50]Allen Institute for Brain Science, Seattle, WA, USA. [51]Aware, Zurich, Switzerland. [52]Electrical and Computer Engineering Department, Princeton University, Princeton, NJ, USA. [53]Institute of Biology, Leipzig University, Leipzig, Germany. [54]Université Côte d'Azur, CNRS, Inserm, iBV, Nice, France. [55]University of Oxford, Oxford, UK. [56]University of California, Merced, Merced, CA, USA. [57]University of Florida, Gainesville, FL, USA.

## Methods

### Annotations

**Base annotations.** At the time of writing, the general FlyWire annotation system operates in a read-only mode in which users can add additional annotations for a neuron but cannot edit or delete existing annotations. Furthermore, the annotations consist of a single free-form text field bound to a spatial location. This enabled many FlyWire users (including our own group) to contribute a wide range of community annotations, which are reported in our companion paper[1] but are not considered in this study. As it became apparent that a complete connectome could be obtained, we found that this approach was not a good fit for our goal of obtaining a structured, systematic and canonical set of annotations for each neuron with extensive manual curation. We therefore set up a web database (seatable; https://seatable. io/) that allowed records for each neuron to be edited and corrected over time; columns with specific acceptable values were added as necessary.

Each neuron was defined by a single point location (also known as a root point) and its associated PyChunkedGraph supervoxel. Root IDs were updated every 30 min by a Python script based on the fafbseg package (Table 1) to account for any edits. The canonical point for the neuron was either a location on a large-calibre neurite within the main arbour of the neuron, a location on the cell body fibre close to where it entered the neuropil or a position within the nucleus as defined by the nucleus segmentation table[80]. The former was preferred as segmentation errors in the cell body fibre tracts regularly resulted in the wrong soma being attached to a given neuronal arbour. These soma swap errors persisted late into proofreading and, when fixed, resulted in annotation information being attached to the wrong neuron until this in turn was fixed.

We also note that our annotations include a number of non-neuronal cells/objects such as glia cells, trachea and extracellular matrix that others might find useful (superclass not_a_neuron; listed in Supplementary Data 2).

**Soma position and side.** Besides the canonical root point, the soma position was recorded for all neurons with a cell body. This was either based on curating entries in the nucleus segmentation table (removing duplicates or positions outside the nucleus) or on selecting a location, especially when the cell body fibre was truncated and no soma could be identified in the dataset. These soma locations were critical for a number of analyses and also allowed a consistent side to be defined for each neuron. This was initialized by mapping all soma positions to the symmetric JRC2018F template and then using a cutting plane at the midline perpendicular to the mediolateral ($x$) axis to define left and right. However, all soma positions within 20 μm of the midline plane were then manually reviewed. The goal was to define a consistent logical soma side based on examination of the cell body fibre tracts entering the brain; this ultimately ensured that cell types present, for example, in one copy per brain hemisphere, were always annotated so that one neuron was identified as the left and the other the right. In a small number of cases, for example, for the bilaterally symmetric octopaminergic ventral unpaired medial neurons, we assigned side as 'central'.

For sensory neurons, side refers to whether they enter the brain through the left or the right nerve. In a small number of cases we could not unambiguously identify the nerve entry side and assigned side as 'na'.

**Biological outliers and sample artefacts.** Throughout our proofreading, matching and cell typing efforts, we recorded cases of neurons that we considered to be biological outliers or showed signs of sample preparation and/or imaging artefacts.

Biological outliers range from small additional/missing branches to entire misguided neurite tracks, and were typically assessed within the context of a given cell type and best possible contralateral matches within FlyWire and/or the hemibrain. When biological outliers were suspected, careful proofreading was undertaken to avoid erroneous merges or splits of neuron segmentation.

Sample artefacts come in two flavours:

(1) A small number of neurons exhibit a dark, almost black cytosol, which caused issues in the segmentation as well as synapse detection. This effect is often restricted to the neurons' axons. We consider these sample artefacts because it is not always consistent within cell types. For example, the cytosol in the axons of DM3 adPN is dark on the left and normal light on the right. Because the dark cytosol leads to worse synapse detection, probably due to lower contrast between the cytosol and synaptic densities, we typically excluded neurons (or neuron types) with sample artefacts from connectivity analyses. Anecdotally, this appears to happen at a much higher frequency in sensory neurons compared with in brain-intrinsic neurons.

(2) Some neurons are missing large arbours (for example, a whole axon or dendrite) because a main neurite suddenly ends and cannot be traced any further. This typically happens in commissures where many neurites co-fasculate to cross the brain's midline. In some but not all cases, we were able to bridge those gaps and find the missing branch through left–right matching. Where neurons remained incomplete, we marked them as outliers.

Whether a neuron represents a biological outlier or exhibits sample preparation/segmentation artefacts is recorded in the status column of our annotations as 'outlier_bio' and 'outlier_seg', respectively. Note that these annotations are probably less comprehensive for the optic lobes than for the central brain. Examples plus quantification are presented in Extended Data Fig. 5.

**Hierarchical annotations.** Hierarchical annotations include flow, superclass, class (plus a subclass field in certain cases) and cell type. The flow and superclass were generally assigned based on an initial semi-automated approach followed by extensive and iterative manual curation. See Supplementary Table 3 for definitions and the sections below for details on certain superclasses.

Based on the superclasses we define two useful groupings which are used throughout the main text:

Central brain neurons consist of all neurons with their somata in the central brain defined by the five superclasses: central, descending, visual centrifugal, motor and endocrine.

Central brain associated neurons further include superclasses: visual projection neurons (VPNs), ascending neurons and sensory neurons (but omit sensory neurons with cell class: visual).

Cell classes in the central brain represent salient groupings/terms that have been previously used in the literature (examples are provided in Supplementary Table 3). For sensory neurons, the class indicates their modality (where known). For optic-lobe-intrinsic neurons cell class indicates their neuropil innervation: for example, cell class 'ME' are medulla local neurons, 'LA>ME' are neurons projecting from the lamina to the medulla and 'ME>LO.LOP' are neurons projecting from the medulla to both lobula and lobula plate.

**Hemilineage annotations.** Central nervous system lineages were initially mapped for the third instar larval brain, where, for each lineage, the neuroblast of origin and its progeny are directly visible[81–84]. Genetic tools that allow stochastic clonal analysis[85] have enabled researchers to visualize individual lineages as GFP-marked 'clones'. Clones reveal the stereotyped morphological footprint of a lineage, its overall 'projection envelope'[32], as well as the cohesive fibre bundles—hemilineage-associated tracts (HATs)—formed by neurons belonging to it. Using these characteristics, lineages could be also identified in the embryo and early larva[86,87], as well as in pupae and adults[31–34,37,88]. HATs can be readily identified in the EM image data, and we used them, in

conjunction with clonal projection envelopes, to identify hemilineages in the EM dataset through a combination of the following methods:

(1) Visual comparison of HATs formed by reconstructed neurons in the EM, and the light microscopy map reconstructed from anti-Neuroglian-labelled brains[31,33,34]. In cross-section, tracts typically appear as clusters of 50–100 tightly packed, rounded contours of uniform diameter (~200 nm), surrounded by neuronal cell bodies (when sectioned in the cortex) or irregularly shaped terminal neurite branches and synapses (when sectioned in the neuropil area; Fig. 2c). The point of entry and trajectory of a HAT in the neuropil is characteristic for a hemilineage.

(2) Matching branching pattern of reconstructed neurons with the projection envelope of clones: as expected from the light microscopy map based on anti-Neuroglian-labelled brains[31], the majority of hemilineage tracts visible in the EM dataset occur in pairs or small groups (3–5). Within these groups, individual tracts are often lined by fibres of larger (and more variable) diameter, as shown in Fig. 2c. However, the boundary between closely adjacent hemilineage tracts is often difficult to draw based on the EM image alone. In these cases, visual inspection and quantitative comparison of the reconstructed neurons belonging to a hemilineage tract with the projection envelope of the corresponding clone, which can be projected into the EM dataset through Pyroglancer (Table 1), assists in properly assigning neurons to their hemilineages.

(3) Identifying homologous HATs across three different hemispheres (left and right of FlyWire, hemibrain): by comparison of morphology (NBLAST[38]), as well as connectivity (assuming that homologous neurons share synaptic partners), we were able to assign the large majority of neurons to specific HATs that matched in all three hemispheres.

In the existing literature, two systems for hemilineage nomenclature are used: Ito/Lee[33,34] and Hartenstein[31,32]. Although these systems overlap in large parts, some lineages have been described in only one but not the other nomenclature. In the main text, we provide (hemi) lineages according to the ItoLee nomenclature for simplicity. Below and in the Supplementary Information, we also provide both names as ItoLee/Hartenstein, and the mapping between the two nomenclatures is provided in Supplementary Data 3. From previous literature, we expected a total of around 119 lineages in the central brain, including the gnathal ganglia (GNG)[31–34,84]. Indeed, we were able to identify all 119 lineages based on light-level clones and tracts, as well as the HATs in FlyWire. Moreover, we found one lineage, LHp3/CP5, which could not be matched to any clone. Thus, together, we have identified 120 lineages.

By comprehensively inspecting the hemilineage tracts originally in CATMAID and then in FlyWire, we can now reconcile previous reports. Specifically, new to refs. 33,34 (ItoLee nomenclature) are: CREl1/DALv3, LHp3/CP5, DILP/DILP, LALa1/BAlp2, SMPpm1/DPMm2 and VLPl5/BLVa3_ or_4—we gave these neurons lineage names according to the naming scheme in refs. 33,34. New to ref. 31 (Hartenstein nomenclature) are: SLPal5/BLAd5, SLPav3/BLVa2a, LHl3/BLVa2b, SLPpl3/BLVa2c, PBp1/ CM6, SLPpl2/CP6, SMPpd2/DPLc6, PSp1/DPMl2 and LHp3/CP5—we named these units according to the Hartenstein nomenclature naming scheme. We did not take the following clones from ref. 33 into account for the total count of lineages/hemilineages, because they originate in the optic lobe and their neuroblast of origin has not been clearly demonstrated in the larva: VPNd2, VPNd3, VPNd4, VPNp2, VPNp3, VPNp4, VPNv1, VPNv2 and VPNv3.

Notably, although light-level clones from refs. 33,34 match very well the great majority of the time, sometimes clones with the same name only match partially. For example, the AOTUv1_ventral/DALcm2_ventral hemilineage seems to be missing in the AOTUv1/DALcm2 clone in the Ito collection[33]. There appears to be a similar situation for the DM4/ CM4, EBa1/DALv2 and LHl3/BLVa2b lineages. When there is a conflict, we have preferred clones as described in ref. 34.

For calculating the total number of hemilineages, to keep the inclusion criteria consistent with the lineages, we included the type II lineages (DL1-2/CP2-3, DM1-6/DPMm1, DPMpm1, DPMpm2, CM4, CM1, CM3) by counting the number of cell body fibre tracts, acknowledging that they may or may not be hemilineages. Neuroblasts of type II lineages, instead of generating ganglion mother cells that each divide once, amplify their number, generating multiple intermediate progenitors that in turn continue dividing like neuroblasts[28,89,90]. It has not been established how the tracts visible in type II clones (and included in Extended Data Fig. 3 and Supplementary Data 3 and 4) relate to the (large number of) type II hemilineages.

There are also 3 type I lineages (VPNl&d1/BLAl2, VLPl2/BLAv2 and VLPp&l1/DPLpv) with more than two tracts in the clone; we included these additional tracts in the hemilineages provided in the text. Without taking these type I and type II tracts into account, we identified 141 hemilineages.

A minority of neurons in the central brain could not reliably be assigned to a lineage. These mainly include the (putative) primary neurons (3,780). Primary neurons, born in the embryo and already differentiated in the larva, form small tracts with which the secondary neurons become closely associated[91]. In the adult brain, morphological criteria that unambiguously differentiate between primary and secondary neurons have not yet been established. In cases in which experimental evidence exists[27], primary neurons have significantly larger cell bodies and cell body fibres. Loosely taking these criteria into account we surmise that a fraction of primary neurons forms part of the HATs defined as described above. However, aside from the HATs, we see multiple small bundles, typically close to but not contiguous with the HATs, which we assume to consist of primary neurons. Overall, these small bundles contained 3,780 neurons, designated as primary or putative primary neurons.

**Hemilineage annotations in hemibrain.** Hemilineage annotations in hemibrain were generated using the hemilineage annotations in FlyWire as the ground truth. For each hemilineage, we first obtained potential hemibrain matches to FlyWire neurons using a combination of NBLAST[38] scores and cell body fibre/cell type annotations. We then clustered neurons in all three hemispheres (FlyWire left, FlyWire right, hemibrain potential candidates) by morphology, and went through the clusters, to make sure that the hemilineage annotations correspond across brains at the finest level possible. To ensure that no neurons within a hemilineage were missed, we examined the cell body fibre bundles of each hemilineage in the hemibrain at the EM level. To further guarantee the completeness of hemilineage annotations, we inventoried all right hemisphere neurons in hemibrain with a cell type annotation, to ensure all neurons with a type annotation were assigned a hemilineage annotation where possible.

**Morphological groups.** Within a hemilineage, subgroups of neurons often share distinctive morphological characteristics. These morphological groups were identified for all hemilineages as follows. Neurons from FlyWire and hemibrain were transformed into the same hemisphere and pairwise NBLAST scores were generated for all neurons within a hemilineage. Intrahemilineage NBLAST scores were then clustered using HDBSCAN[92], an adaptive algorithm that does not require a uniform threshold across all clusters, and that does not assume spherical distribution of data points in a cluster, compared to other clustering algorithms such as k-means clustering.

To test the robustness of the morphological groups, we reran the above analysis across one, two or three hemispheres. This treatment sometimes gave slightly different results. However, some groups of neurons consistently co-clustered across the different hemispheres; we termed these 'persistent clusters'. Early-born neurons, which are often morphologically unique, frequently failed to participate in persistent clusters, and were omitted from further analysis. We linked these persistent clusters across hemispheres using two- and three-hemisphere clustering: for example, when clustering FlyWire left and FlyWire right together for hemilineage AOTUv3_dorsal, the TuBu neurons from both the left and right hemispheres would fall into one cluster, which we termed a

morphological group. Morphological groups are therefore defined by consistent across-hemisphere clustering. When neurons of a given hemilineage were sufficiently contained by the hemibrain volume, all three hemispheres (two from FlyWire and one from hemibrain) were used; otherwise, the two hemispheres from FlyWire were used. As we prioritized consistency across 1, 2 and 3 hemisphere clustering, a minority of neurons with a hemilineage annotation do not have a morphological group. For example, if neuron type A clusters with type B in one-hemisphere clustering, but clusters with type C (and not B) in two-hemisphere clustering, then type A will not have a morphological group annotation.

After generating the morphological groups, we cross-checked these annotations against existing cross-identified hemibrain types and (FlyWire only) cell types. In a minority of cases, neurons of one hemibrain/cell type were annotated with multiple morphological groups. This occasionally reflected errors in assigning types, which were corrected; and others where individual neurons from a type were singled out due to additional branches/reconstruction issues. We therefore manually corrected some morphological group annotations to make them correspond maximally with the hemibrain/cell type annotations.

Overall, we divide hemilineages in each hemisphere into 528 morphological groups, with hemilineages typically having 1–6 morphological groups (10/90 quantile) and with each morphological group containing 2–52 neurons in each hemisphere (10/90 quantile).

### Cell typing
Using methods described in detail in the sections below, we defined cell types for 96.4% of all neurons in the brain—98% and 92% for the central brain and optic lobes, respectively. The remaining 3.6% of neurons were largely (1) optic lobe local neurons for which we could not find a prior in existing literature or (2) neurons without clear contralateral pairings, including a number of neurons on the midline.

About 21% of our cell type annotations are principally derived from the hemibrain cell type matching effort (see the section below). The remainder was generated either by comparing to existing literature (for example, in case of optic lobe cell types or sensory neurons) and/or by finding left/right balanced clusters through a combination of NBLAST and connectivity clustering (Fig. 6 and Extended Data Figs. 8 and 9). New types were given a simple numerical cross-brain identifier (for example, CB0001) or, in the case of ascending neurons (ANs)/descending neurons (DNs), a more descriptive identifier (see the section below) as a provisional cell type label. A flow chart summary is provided in Extended Data Fig. 12.

For provenance, we provide two columns of cell types in our Supplementary Data:

hemibrain_type always refers to one or more hemibrain cell types; in rare occasions where a matched hemibrain neuron did not have a type, we recorded body IDs instead.

cell_type contains types that are either not derived from the hemibrain or that represent refinements (for example, a split or retyping) of hemibrain types.

Neurons can have both a cell_type and a hemibrain_type entry, in which case, the cell_type represents a refinement or correction and should take precedence. This generates the reported total count of 8,453 terminal cell types and includes 3,643 hemibrain-derived cell types (Fig. 3h (right side of the flow chart)) and 4,581 proposals for new types. New types consist of 3,504 CBXXXX types, 65 new visual centrifugal neuron types ('c' prefix, for example, cL08), 173 new VPN types ('e' suffix, for example, LTe07), 602 new AN types ('AN_' or 'SA_' prefix, for example, AN_SMP_1) and 237 new DN types ('e' suffix, for example, DNge094). The remaining 229 types are cell types known from other literature, for example, columnar cell types of the optic lobes.

**Hemibrain cell type matching.** We first used NBLAST[38] to match FlyWire neurons to hemibrain cell types (see 'Morphological comparisons' section). From the NBLAST scores, we extracted, for each

FlyWire neuron, a list of potential cell type hits using all hits in the 90th percentile. Individual FlyWire neurons were co-visualized with their potential hits in neuroglancer (see the 'Data availability' and 'Code availability' sections) and the correct hit (if found) was recorded. In difficult cases, we would also inspect the subtree of the NBLAST dendrograms containing the neurons in questions to include local cluster structure in the decision making (Extended Data Fig. 4e). In cases in which two or more hemibrain cell types could not be cleanly delineated in FlyWire (that is, there were no corresponding separable clusters) we recorded composite (many:1) type matches (Fig. 3i and Extended Data Figs. 4g and 12).

When a matched type was either missing large parts of its arbours due to truncation in the hemibrain or the comparison with the FlyWire matches suggested closer inspection was required, we used cross-brain connectivity comparisons (see the section below) to decide whether to adjust (split or merge) the type. A merge of two or more hemibrain types was recorded as, for example, SIP078,SIP080, while a split would be recorded as PS090a and PS090b (that is, with a lower-case letter as a suffix). In rare cases in which we were able to find a match for an untyped hemibrain neuron, we would record the hemibrain body ID as hemibrain type and assign a CBXXXX identifier as cell type.

Finally, the hemibrain introduced the concept of morphology types and 'connectivity types'[2]. The latter represent refinements of the former and differ only in their connectivity. For example, morphology type SAD051 splits into two connectivity types: SAD051_a and SAD051_b, for which the _{letter} indicates that these are connectivity types. Throughout our FlyWire↔hemibrain matching efforts we found connectivity types hard to reproduce and our default approach was to match only up to the morphology type. In some cases, for example, antennal lobe local neuron types like lLN2P_a and lLN2P_b, we were able to find the corresponding neurons in FlyWire.

Note that, in numerous cases that we reviewed but remain unmatched, we encountered what we call ambiguous 'daisy-chains': imagine four fairly similar cell types, A, B, C and D. Often these adjacent cell types represent a spectrum of morphologies where A is similar to B, B is similar to C and C is similar to D. The problem now is in unambiguously telling A from B, B from C and C from D. But, at the same time, A and D (on the opposite ends of the spectrum) are so dissimilar that we would not expect to assign them the same cell type (Fig. 3k and Extended Data Fig. 4h). These kinds of graded or continuous variation have been observed in a number of locations in the mammalian nervous system and represent one of the classic complications of cell typing[18]. Absent other compelling information that can clearly separate these groups, the only reasonable option would seem to be to lump them together. As this would erase numerous proposed hemibrain cell types, the de facto standard for the fly brain, we have been conservative about making these changes pending analysis of additional connectome data[2].

**Hemibrain cell type matching with connectivity.** In our hemibrain type matching efforts, about 12% of cell types could not be matched 1:1. In these cases, we used across-dataset connectivity clustering (for example, to confirm the split of a hemibrain type or a merger of multiple cell types). To generate distances, we first produced separate adjacency matrices for each of the three hemispheres (FlyWire left, right and hemibrain). In these matrices, each row is a query neuron and each column is an up- or downstream cell type; the values are the connection weights (that is, number of synapses). We then combine the three matrices along the first axis (rows) and retain only the cell types (columns) that have been cross-identified in all hemispheres. From the resulting observation vector, we calculate a pairwise cosine distance. It is important to note that this connectivity clustering depends absolutely on the existence of a corpus of shared labels between the two datasets—without such shared labels, which were initially defined by

morphological matching as described above, connectivity matching cannot function.

This pipeline is implemented in the coconatfly package (Table 1), which provides a streamlined interface to carry out such clustering. For example the following command can be used to see if the types given to a selection of neurons in the Lateral Accessory Lobe (LAL) are robust: cf_cosine_plot(cf_ids('/type:LAL0(08|09|10|42)', datasets=c("flywire", "hemibrain"))).

An optional interactive mode allows for efficient exploration within a web browser. For further details and examples, see https://natverse.org/coconatfly/.

**Defining robust cross-brain cell types.** In Fig. 6, we used two kinds of distance metrics—one calculated from connectivity alone (used for FC1–3; Fig. 6e–g) and a second combining morphology + connectivity (used for FB1–9; Fig. 6h and Extended Data Fig. 8b–f) to help define robust cross-brain cell types. The connectivity distance is as described in the 'Hemibrain cell type matching with connectivity' section above). We note that the central complex retyping used FlyWire connectivity from the 630 release. The combined morphology + connectivity distances were generated by taking the sum of the connectivity and NBLAST distances. Connectivity-only works well in the case of cell types that do not overlap in space but instead tile a neuropil. For cell types that are expected to overlap in space, we find that adding NBLAST distances is a useful constraint to avoid mixing of otherwise clearly different types. From the distances, we generated a dendrogram representation using the Ward algorithm and then extracted the smallest possible clusters that satisfy two criteria: (1) each cluster must contain neurons from all three hemispheres (hemibrain, FlyWire right and FlyWire left); (2) within each cluster, the number of neurons from each hemisphere must be approximately equal.

We call such clusters 'balanced'. The resulting groups were then manually reviewed.

**Defining new provisional cell types.** After the hemibrain type matching effort, around 40% of central brain neurons remained untyped. This included both neurons mostly or entirely outside the hemibrain volume (for example, from the GNG) but also neurons for which the potential hemibrain type matches were too ambiguous. To provide provisional cell types for these neurons, we ran the same cell typing pipeline described in the 'Defining robust cross-brain cell types' section above on the two hemispheres of FlyWire alone. In brief, we produced a morphology + connectivity co-clustering for each individual hemilineage (neurons without a hemilineage such as putative primary neurons were clustered separately) and extracted 'balanced' clusters, which were manually reviewed (Fig. 6i,j and Extended Data Fig. 9). Reviewed clusters were then used to add new or refine existing cell and hemibrain types:

- Clusters consisting entirely of previously untyped neurons were given a provisional CBXXXX cell type.
- Clusters containing a mix of hemibrain-typed and untyped neurons typically meant that, after further investigation, the untyped neurons were given the same hemibrain type.
- Hemibrain types split across multiple clusters were double checked (for example, by running a triple-hemisphere connectivity clustering), which often led to a split of the hemibrain type; for example, SMP408 was split into SMP408a–d.
- In rare cases, clusters contained a mix of two or more hemibrain types; these were double checked and the hemibrain types corrected (for example, by merging two or more hemibrain types, or by removing hemibrain type labels).

To validate a subset of the new, provisional cell types, we re-ran the clustering using three hemispheres (FlyWire + hemibrain) on 25 cross-identified hemilineages that are not truncated in the hemibrain

(Extended Data Fig. 9). The procedure was otherwise the same as for the double-clustering.

**Optic lobe cell typing.** We provide cell type annotations for >92% of neurons in both optic lobes. The vast majority of these types are based on previous literature[42,93–99]. We started the typing effort by annotating well-known large tangential cells (for example, Am1 or LPi12), VPNs (for example, LT1s) as well as photoreceptor neurons. From there, we followed two general strategies, sometimes in combination: (1) for neurons with known connectivity fingerprints, we specifically hunted upstream or downstream of neurons of interest (for example, looking for T4a neurons upstream of LPi12). (2) We ran connectivity clustering as described above on both optic lobes combined. Clusters were manually reviewed and matched against literature. This was done iteratively; with each round adding new or refining existing cell types to inform the next round of clustering. Clusters that we could not confidently match against a previously described cell type were assigned a provisional (CBXXXX) type.

This effort was carried out independently of other FlyWire optic lobe intrinsic neuron typing, including ref. 23; the sole exception was the Mi1 cell type, which was initially based on annotations reported previously[100] and then reviewed. For this reason ref. 100 should be cited for the Mi1 annotations. Note that our typing focuses on previously reported cell types rather than defining new ones, but covers both optic lobes to enable accurate typing of visual project neurons (by defining their key inputs). For the 38,461 neurons of the right optic lobe (for which a comparison is possible), we report 156 cell types for 35,567 neurons compared with 229 cell types for 37,345 neurons in ref. 23.

**VPNs and VCNs.** Similar to cell typing in the central brain, a significant proportion of VPN (61%) and visual centrifugal neuron (VCN) (60%) types are derived from the hemibrain (see the 'Hemibrain cell type matching' section). These annotations are listed in the hemibrain_type column in the Supplementary Data.

To assign cell types to the remaining neurons and in some cases also to refine existing hemibrain types, we ran a double-hemisphere (Fly-Wire left–right) co-clustering. For VCNs, this was done as part of the per-hemilineage morphology-connectivity clustering described in the 'Defining new provisional cell types' section above. For VPNs of which the dendrites typically tile the optic neuropils, we generated and reviewed a separate connectivity-only clustering on all VPNs together. Groups extracted from this clustering were also cross-referenced with new literature from parallel typing efforts[100,101] and those new cell type names were preferred for the convenience of the research community. In cases in which literature references could not be found, systematic names were generated de novo using the schemata below.

For VPNs the nomenclature follows the format [neuropil][C/T][e][XX], where neuropil refers to regions innervated by VPN dendrites; C/T denotes columnar versus tangential organization; e indicates identification through EM; and XX represents a zero padded two digit number.

For example: 'MTe47' for 'medulla-tangential 47'.

For VCNs, the nomenclature follows the format [c][neuropil][XX], where c denotes centrifugal; neuropil refers to regions innervated by VCN axons; and XX represents a zero padded two digit number.

For example, 'cM12' for 'centrifugal medulla-targeting 12'.

Note that new names were also given to non-canonical, generic hemibrain types, such as IB006. All new names are recorded in the cell_type column in the Supplementary Data.

The majority of VPNs (99.6%) and VCNs (98.3%) were assigned to specific types. Only 29 VPNs and 9 VCNs could not be confidently assigned a cell type and were therefore left untyped.

**Sensory and motor neurons.** We identified all non-visual sensory and motor neurons entering/exiting the brain through the antennal, eye, occipital and labial nerves by screening all axon profiles in a given nerve.

Sensory neurons were further cross-referenced to existing literature to assign modalities (through the class field) and, where applicable, a cell type. Previous studies have identified almost all head mechanosensory bristle and taste peg mechanosensory neurons[102] in the left hemisphere (at the time of publication: right hemisphere). Gustatory sensory neurons were previously identified in ref. 103 and Johnston's organ neurons in refs. 104,105 in a version of the FAFB that used manual reconstruction (https://fafb.catmaid.virtualflybrain.org). Those neurons were identified in the FlyWire instance by transformation and overlay onto FlyWire space as described previously[102].

Johnston's organ neurons in the right hemisphere were characterized based on innervation of the major AMMC zones (A, B, C, D, E and F), but not further classified into subzone innervation as shown previously[104]. Other sensory neurons (mechanosensory bristle neurons, taste peg mechanosensory neurons and gustatory sensory neurons) in the right hemisphere were identified through NBLAST-based matching of their mirrored morphology to the left hemisphere and expert review. Olfactory, thermosensory and hygrosensory neurons of the antennal lobes were identified through their connectivity to cognate uniglomerular projection neurons and NBLAST-based matching to previously identified hemibrain neurons[40,106].

Visual sensory neurons (R1–6, R7–8 and ocellar photoreceptor neurons) were identified by manually screening neurons with pre-synapse in either the lamina, the medulla and/or the ocellar ganglia[93].

**ANs and DNs.** We seeded all profiles in a cross-section in the ventral posterior GNG through the cervical connective to identify all neurons entering and exiting the brain at the neck. We identified all DNs based on the following criteria: (1) soma located within the brain dataset; and (2) main axon branch leaving the brain through the cervical connective.

We next classified the DNs based on their soma location according to a previous report[107]. In brief, the soma of DNa, DNb, DNc and DNd is located in the anterior half (a, anterior dorsal; b, anterior ventral; c, in the pars intercerebralis; d, outside cell cluster on the surface) and DNp in the posterior half of the central brain. DNg somas are located in the GNG.

To identify DNs described in ref. 107 in the EM dataset, we transformed the volume renderings of DN GAL4 lines into FlyWire space. Displaying EM and LM neurons in the same space enabled accurate matching of closely morphologically related neurons. For DNs without available volume renderings, we identified candidate EM matches by eye, transformed them into JRC2018U space and overlaid them onto the GAL4 or Split GAL4 line stacks (named in ref. 107 for that type) in FIJI for verification. Using these methods, we identified all but two (DNd01 and DNg25) in FAFB/FlyWire and annotated their cell type with the published nomenclature. All other unmatched DNs received a systematic cell type consisting of their soma location, an 'e' for EM type and a three digit number (for example, DNae001). A detailed account and analysis of DNs has been published[108] separately.

ANs were identified based on the following criteria: (1) no soma in the brain; and (2) main branch entering through the neck connective (note that some ANs make a dendrite after entry through the neck connective and then an axon).

To distinguish sensory ascending (SA) neurons from ANs, we analysed SA neuron morphology in the male VNC dataset MANC[109,110]. First, we identified which longitudinal tract they travel to ascend to the brain[111] and then found GAL4 lines matching their VNC morphology. We next identified putative matching axons in the brain dataset by morphology and tract membership. A detailed description of this process and the lines used has been published separately[108].

### FAFB laterality
In the fly brain, the asymmetric body is reproducibly around 4 times larger on the right hemisphere than on the left[112–114], except in rare cases of situs inversus[114,115]. However, completion of the FlyWire whole-brain connectome and associated cell typing showed the asymmetric body to be larger on the apparent left side of the brain rather than the right, suggesting an inversion of the left–right axis during initial acquisition of EM images comprising the FAFB dataset[17]. This hypothesis was confirmed by comparing of FAFB sample grids imaged using differential interference contrast microscopy to low-magnification views of corresponding EM image mosaics using CATMAID or neuroglancer. Grids were chosen with particularly obvious staining and sample preparation artefacts visible both in the differential interference contrast and low-magnification EM images (Extended Data Fig. 1), confirming that a left–right axis inversion had taken place during image acquisition.

Owing to the extensive post-processing of the FAFB dataset and derived datasets (for example, transformation fields, image mosaicing and stack registrations to produce aligned volumes, segmentation supervoxels, proofread neuron segmentations, skeletons, meshes and myriad 3D visualizations), which had been undertaken at the time at which this error was discovered, we deemed it impractical to correct this error at the raw data level. Instead, we break a convention of presentation: usually, frontal views of the fly brain place the fly's right on the viewer's left. Instead, in this paper, frontal views of the fly brain place the fly's right on the viewer's right—similar to the view one has of oneself while looking in a mirror. This maintains consistency with past publications. However, note that all labels of left and right in the figures in this paper, our companion papers, the supplemental annotations and associated digital repositories (for example, https://codex.flywire.ai, FAFB/FlyWire CATMAID) have been corrected to reflect the error during data acquisition. In these resources, a neuron labelled as being on the left is indeed on the left of the fly's brain.

For consistency with visualizations and datasets obeying the standard convention (fly's right on viewer's left), FlyWire data can be mirrored. To facilitate this, we provide tools to digitally mirror FAFB-FlyWire data using the Python flybrains (https://github.com/navis-org/navis-flybrains) or natverse nat.jrcbrains (https://github.com/natverse/nat.jrcbrains) packages (Extended Data Fig. 1c), through the navis.mirror_brain() and nat.jrcbrains::mirror_fafb() function calls, respectively. See the fafbseg-py documentation for a tutorial on mirroring.

We also provide a neuroglancer scene in which both FlyWire and hemibrain data are displayed in the correct orientation: https://tinyurl.com/flywirehbflip783. In this scene, a frontal view has both FAFB and hemibrain RHS to the left of the screen, obeying the standard convention. The scene displays the SA1 and SA2 neurons, which target the right asymmetric body for both FlyWire and the hemibrain, confirming that the RHS for both datasets has been superimposed (compare with Extended Data Fig. 1a).

### Morphological comparisons
Throughout our analyses, NBLAST[38] was used to generate morphological similarity scores between neurons—for example, for matching neurons between the FlyWire and the hemibrain datasets, or for the morphological clustering of the hemilineages. In brief, NBLAST treats neurons as point clouds with associated tangent vectors describing directionality, so called dotprops. For a given query→target neuron pair, we perform a $k$-nearest neighbours search between the two point clouds and score each nearest-neighbour pair by their distance and the dot product of their vector. These are then summed up to compute the final query→target NBLAST score. It is important to note that direction of the NBLAST matters, that is, NBLASTing neurons A→B≠B→A. Unless otherwise noted, we use the minimum between the forward and reverse NBLAST scores.

The NBLAST algorithm is implemented in both navis and the natverse (Table 1). However, we modified the navis implementation for more efficient parallel computation in order to scale to pools of more than 100,000 neurons. For example, the all-by-all NBLAST matrix for the

full 139,000 FlyWire neurons alone occupies over 500 GB of memory (32 bit floats). Most of the large NBLASTs were run on a single cluster node with 112 CPUs and 1 TB RAM provided by the MRC LMB Scientific Computing group, and took between 1 and 2 days (wall time) to complete.

Below, we provide recipes for the different NBLAST analyses used in this paper:

**FlyWire all-by-all NBLAST.** For this NBLAST, we first generated skeletons using the L2 cache. In brief, underlying the FlyWire segmentation is an octree data structure where level 0 represents supervoxels, which are then agglomerated over higher levels[116]. The second layer (L2) in this octree represents neurons as chunks of roughly $4 \times 4 \times 10$ μm in size, which is sufficiently detailed for NBLAST. The L2 cache holds precomputed information for each L2 chunk, including a representative $x/y/z$ coordinate in space. We used the $x/y/z$ coordinates and connectivity between chunks to generate skeletons for all FlyWire neurons (implemented in fafbseg; Table 1). Skeletons were then pruned to remove side branches smaller than 5 μm. From those skeletons, we generated the dotprops for NBLAST using navis.

Before the NBLAST, we additionally transformed dotprops to the same side by mirroring those from neurons with side right onto the left. The NBLAST was then run only in forward direction (query→target) but, because the resulting matrix was symmetrical, we could generate minimum NBLAST scores using the transposed matrix: $\min(A + A^T)$.

This NBLAST was used to find left–right neuron pairs, define (hemi) lineages and run the morphology group clustering.

**FlyWire—hemibrain NBLAST.** For FlyWire, we re-used the dotprops generated for the all-by-all NBLAST (see the previous section). To account for the truncation of neurons in the hemibrain volume, we removed points that fell outside the hemibrain bounding box.

For the hemibrain, we downloaded skeletons for all neurons from neuPrint (https://neuprint.janelia.org) using neuprint-python and navis (Table 1). In addition to the approximately 23,000 typed neurons, we also included all untyped neurons (often just fragments) for a total of 98,000 skeletons. These skeletons were pruned to remove twigs smaller than 5 μm and then transformed from hemibrain into FlyWire (FAFB14.1) space using a combination of non-rigid transforms[116,117] (implemented through navis, navis-flybrain and fafbseg; Table 1). Once in FlyWire space, they were resampled to 0.5 nodes per μm of cable to approximately match the resolution of the FlyWire L2 skeletons, and then turned into dotprops. The NBLAST was then run both in forward (FlyWire to hemibrain) and reverse (hemibrain to FlyWire) direction and the minimum between both were used.

This NBLAST allowed us to match FlyWire left against the hemibrain neurons. To also allow matching FlyWire right against the hemibrain, we performed a second run after mirroring the FlyWire dotprops to the opposite side.

In Fig. 3c,d, we manually reviewed NBLAST matches. For this, we sorted hemibrain neurons based on their highest NBLAST score to a FlyWire neuron into bins with a width of 0.1. From each bin, we picked 30 random hemibrain neurons (except for bin 0–0.1 which contained only 27 neurons in total) and scored their top five FlyWire matches as to whether a plausible match was among them. In total, this sample contained 237 neurons.

**Cross-brain co-clustering.** The pipeline for the morphology-based across brain co-clustering used in Fig. 6 and Extended Data Fig. 9 was essentially the same as for the FlyWire–hemibrain NBLAST with two exceptions: (1) we used high-resolution FlyWire skeletons instead of the coarser L2 skeletons (see below); and (2) both FlyWire and hemibrain skeletons were resampled to 1 node per μm before generating dotprops.

## High-resolution skeletonization

In addition to the coarse L2 skeletons, we also generated high-resolution skeletons that were, for example, used to calculate the total length of neuronal cable reported in our companion paper[1] (149.2 m). In brief, we downloaded neuron meshes (LOD 1) from the flat 783 segmentation (available at gs://flywire_v141_m783) and skeletonized them using the wavefront method implemented in skeletor (https://github.com/navis-org/skeletor). Skeletons were then rerooted to their soma (if applicable), smoothed (by removing small artifactual bristles on the backbone), healed (segmentation issues can cause breaks in the meshes) and slightly downsampled. A modified version of this pipeline is implemented in fafbseg. Skeletons are available for download (see the 'Data availability' and 'Code availability' sections).

## Connectivity normalization

Throughout this paper, the basic measure of connection strength is the number of unitary synapses between two or more neurons[79]; connections between adult fly neurons can reach thousands of such unitary synapses[2]. Previous work in larval *Drosophila* has indicated that synaptic counts approximate contact area[118], which is most commonly used in mammalian species when a high-resolution measure of anatomical connection strength is required. Connectomics studies also routinely use connection strength normalized to the target cell's total inputs[71,79]. For example, if neurons $i$ and $j$ are connected by 10 synapses and neuron $j$ receives 200 inputs in total, the normalized connection weight $i$ to $j$ would be 5%. A previous study[119] showed that while absolute number of synapses for a given connection changes drastically over the course of larval stages, the proportional (that is, normalized) input to the downstream neuron remains relatively constant[119]. Importantly, we have some evidence (Fig. 4g) that normalized connection weights are robust against technical noise (differences in reconstruction status, synapse detection). Note that, for analyses of mushroom body circuits, we use an approach based on the fraction of the input or output synaptic budget associated with different KC cell types; this differs slightly from the above definition and will be detailed in a separate section below.

## Connectivity stereotypy analyses

For analyses on connectivity stereotypy (Fig. 4 and Extended Data Fig. 6) we excluded a number of cell types:
- KCs, due to the high variability in numbers and synapse densities in the mushroom body lobes between FlyWire and the hemibrain (Fig. 5 and Extended Data Fig. 7).
- Cell types that exist only on the left but not the right hemisphere of the hemibrain because our comparison was principally against the right hemisphere.
- Antennal lobe receptor neurons, because truncation/fragmentation in the hemibrain causes some ambiguity with respect to their side annotation.
- Cell types with members that have been marked as being affected by sample or imaging artefacts (that is, status 'outlier_seg').
- VPNs, as they are heavily truncated in the hemibrain.

Among the remaining types, we used only the 1:1 and 1:many but not the many:1 matches. Taken together, we used 2,954 (hemibrain) types for the connectivity stereotypy analyses.

## Availability through CATMAID Spaces

To increase the accessibility and reach of the annotated FlyWire connectome, meshes of proofread FlyWire neurons and synapses were skeletonized and imported into CATMAID, a widely used web-based tool for collaborative tracing, annotation and analysis of large-scale neuronal anatomy datasets[79,120] (https://catmaid.org; Extended Data Fig. 10). Spatial annotations like skeletons are modelled using PostGIS

data types, a PostgreSQL extension that is popular in the geographic information system community. This enables us to reuse many existing tools to work with large spatial datasets, for example, indexes, spatial queries and mesh representation.

A publicly available version of the FlyWire CATMAID project is available online (https://fafb-flywire.catmaid.org). This project uses a new extension, called CATMAID Spaces (https://catmaid.org/en/latest/spaces.html), which allows users to create and administer their own tracing and annotation environments on top of publicly available neuronal image volumes and connectomic datasets. Moreover, users can now login through the public authentication service ORCiD (https://www.orcid.org), so that everyone can log-in on public CATMAID projects. Users can also now create personal copies (Spaces) of public projects. The user then becomes an administrator, and can invite other users, along with the management of their permissions in this new project. Invitations are managed through project tokens, which the administrator can generate and send to invitees for access to the project. Both CATMAID platforms can talk to each other and it is possible to load data from the dedicated FAFB-FlyWire server in the more general Spaces environment.

Metadata annotations for each neuron (root id, cell type, hemilineage, neurotransmitter) were imported for FlyWire project release 783. Skeletons for all 139,255 proofread neurons were generated from the volumetric meshes (see the 'High-resolution skeletonization' section) and imported into CATMAID, resulting in 726,831,877 treenodes. To reduce the import time, skeletons were imported into CATMAID directly as database inserts through SQL, rather than through public RESTful APIs. FlyWire root IDs are available as metadata for each neuron, facilitating interchange with related resources such as FlyWire Codex[1]. Synapses attached to reconstructed neurons were imported as CATMAID connector objects and attached to neuron skeletons by doing a PostgreSQL query to find the nearest node on each of the partner skeletons. Connector objects were linked to postsynaptic partners only if the downstream neuron was in the proofread data release (180,016,288 connections from the 130,054,535 synapses with at least one partner in the proofread set).

## Synapse counts

Insect synapses are polyadic, that is, each presynaptic site can be associated with multiple postsynaptic sites. In contrast to the Janelia hemibrain dataset, the synapse predictions used in FlyWire do not have a concept of a unitary presynaptic site associated with a T-bar[46]. Thus, pre-synapse counts used in this paper do not represent the number of presynaptic sites but rather the number of outgoing connections.

In *Drosophila* connectomes, reported counts of the inputs (post-synapses) onto a given neuron are typically lower than the true number. This is because fine-calibre dendritic fragments frequently cannot be joined onto the rest of the neuron, instead remaining as free-floating fragments in the dataset.

## Technical noise model

To model the impact of technical noise such as proofreading status and synapse detection on connectivity, we first generated a fictive '100%' ground-truth connectivity. We took the connectivity between cell-typed left FlyWire neurons and scaled each edge weight (the number of synapses) by the postsynaptic completion rates in the respective neuropil. For example, all edge weights in the left mushroom body calyx (CA), which has a postsynaptic completion rate of 52.5%, were scaled by a factor of $100/52.5 = 1.9$.

In the second step, we simulated the proofreading process by randomly drawing (without replacement) individual synaptic connections from the fictive ground-truth until reaching a target completion rate. We further simulate the impact of false positives and false negatives by randomly adding and removing synapses to/from the draw according to

the precision (0.72) and recall (0.77) rates reported previously[46]. In each round, we made two draws: (1) A draw using the original per-neuropil postsynaptic completion rates; and (2) a draw where we flip the completion rates for left and right neuropils, that is, use the left CA completion rate for the right CA and vice versa.

In each of the 500 rounds that we ran, we drew two weights for each edge. Both stem from the same fictive 100% ground-truth connectivity but have been drawn according to the differences in left versus right hemisphere completion rates. Combining these values, we calculated the mean difference and quantiles as function of the weight for the FlyWire left (that is, the draw that was not flipped) (Fig. 4i). We focussed this analysis on edge weights between 1 and 30 synapses because the frequency of edges stronger than that is comparatively low, leaving gaps in the data.

## KC analyses

**Connection weight normalization and synaptic budget analysis.** When normalizing connection weights, we typically convert them to the percentage of total input onto a given target cell (or cell type). However, in the case of the mushroom body, the situation is complicated by what we think is a technical bias in the synapse detection methods used for the two connectomes that causes certain kinds of unusual connections to be very different in frequency between the two datasets. We find that the total number of post-synapses as well as the post-synapse density in the mushroom body lobes are more than doubled in the hemibrain compared with in FlyWire (Extended Data Fig. 7b,c). This appears to be explained by certain connections (especially KC to KC connections, which are predominantly arranged with an unusual rosette configuration along axons and of which the functional significance is poorly understood[121]) being much more prevalent in the hemibrain than in FlyWire (Extended Data Fig. 7d). Some other neurons, including the APL giant interneuron, also make about twice as many synapses onto KCs in the hemibrain compared with in FlyWire (Extended Data Fig. 7a). As a consequence of this large number of inputs onto KC axons in the hemibrain, input percentages from all other cells are reduced in comparison with FlyWire.

To avoid this bias, and because our main goal in the KC analysis was to compare different populations of KCs, we instead expressed connectivity as a fraction of the total synaptic budget for upstream or downstream cell types. For example, we examined the fraction of the APL output that is spent on each of the different KC types. Similarly, we quantified connectivity for individual KCs as a fraction of the budget for the whole KC population.

**Calculating K from observed connectivity.** Calculation of K, that is, the number of unique odour channels that each KC receives input from, was principally based on their synaptic connectivity. For this, we looked at their inputs from uniglomerular ALPNs and examined from how many of the 58 antennal lobe glomeruli does a KC receive input from. K as reported in Fig. 6 is based on non-thresholded connectivity. Filtering out weak connections does lower K but, importantly, our observations (for example, that KCg-m cells have a lower K in FlyWire than in the hemibrain) are stable across thresholds (Extended Data Fig. 7g).

**KC model.** A simple rate model of neural networks[122] was used to generate the theoretical predictions of K, the number of ALPN inputs that each KC receives (Fig. 5k). KC activity is modelled by

$$\mathbf{h} = \mathbf{W} \cdot \mathbf{r}_{PN},$$

where $\mathbf{h}$ is a vector of length $M$ representing KC activity, $\mathbf{W}$ is an $M \times N$ matrix representing the synaptic weights between the KCs and PNs, $\mathbf{r}_{PN}$ is a vector of length $N$ representing PN activity. The number of KCs and ALPNs is denoted by $M$ and $N$, respectively. In this model, the PN

activity is assumed to have zero mean, $\bar{\mathbf{r}}_{PN} = 0$, and be uncorrelated, $\overline{\mathbf{r}_{PN} \cdot \mathbf{r}_{PN}} = \mathbf{I}_N$. Here, $\mathbf{I}_N$ is an $N \times N$ identity matrix and $\bar{\mathbf{r}}_{PN}$ denotes the average taken over independent realizations of $\mathbf{r}_{PN}$. Then, the $ij$th element of the covariance matrix of $\mathbf{h}$ is

$$[\mathbf{C}]_{ij} = \overline{[\mathbf{h}]_i [\mathbf{h}]_j} = \sum_{k=0}^{N} [\mathbf{W}]_{ik} [\mathbf{W}]_{jk}.$$

More detailed calculations can be found in a previous report[122]. Randomized and homogeneous weights were used to populate $\mathbf{W}$, such that each row in $\mathbf{W}$ has $K$ elements that are $1 - \alpha$ and $N - K$ elements that are $-\alpha$. The parameter $\alpha$ represents a homogeneous inhibition corresponding to the biological, global inhibition by APL. The value inhibition was set to be $\alpha = A/M$, where $A = 100$ is an arbitrary constant and $M$ is the number of KCs in each of the three datasets. The primary quantity of interest is the dimension of the KC activities defined by[122]:

$$\dim(\mathbf{h}) = \frac{(\mathrm{Tr}[\mathbf{C}])^2}{\mathrm{Tr}[\mathbf{C}^2]}$$

and how it changes with respect to $K$, the number of input connections. In other words, what are the numbers of input connections $K$ onto individual KCs that maximize the dimensionality of their responses, $\mathbf{h}$, given $M$ KCs, $N$ ALPNs and a global inhibition $\alpha$?

From Fig. 5k, the theoretical values of $K$ that maximize $\dim(\mathbf{h})$ in this simple model demonstrate the consistent shift towards lower values of $K$ found in the FlyWire left and FlyWire right datasets when compared with the hemibrain.

The limitations of the model are as follows:

(1) The values in the connectivity matrix $\mathbf{W}$ take only two discrete values, either 0 and 1 or $1 - \alpha$ and $\alpha$. In a way, this helps when calculating analytical results for the dimensionality of the KC activities. However, it is unrealistic as the connectomics data give the number of synaptic connections between the ALPNs and the KCs.

(2) The global inhibition provided by APL to all of the mixing layer neurons is assumed to take a single value for all neurons. In reality, the level of inhibition would be different depending on the number of synapses between APL and the mixing layer neurons.

(3) It is unclear whether the simple linear rate model presented in the original paper represents the behaviour of the biological neural circuit well. Furthermore, it remains unproven that the ALPN-KC neural circuit is attempting to maximize the dimensionality of the KC activities, albeit the theory is biologically well motivated (but see refs. 49,50).

(4) The number of input connections to each mixing layer neuron is kept at a constant $K$ for all neurons. It is definitely a simplification that can be corrected by introducing a distribution P(K) but this requires further detailed modelling.

## Statistical analyses

Unless otherwise stated, statistical analyses (such as Pearson $R$ or cosine distance) were performed using the implementations in the scipy[123] Python package. To determine statistical significance, we used either $t$-tests for normally distributed samples, or Kolmogorov–Smirnov tests otherwise.

Cohen's $d$[124] was calculated as follows:

$$d = \frac{\overline{x_1} - \overline{x_2}}{s}$$

where pooled s.d. $s$ is defined as:

$$s = \sqrt{\frac{(n_1 - 1)s_1^2 + (n_2 - 1)s_2^2}{n_1 + n_2 - 2}}$$

where the variance for one of the groups is defined as:

$$s_1^2 = \frac{1}{n_1 - 1} \sum_{i=1}^{n_1} (x_{1,i} - \overline{x_1})^2$$

and similar for the other group.

Enhanced box plots—also called letter-value plots[125]—in Fig. 5h and Extended Data Fig. 7f are a variation of box plots better suited to represent large samples. They replace the whiskers with a variable number of letter values where the number of letters is based on the uncertainty associated with each estimate, and therefore on the number of observations. The 'fattest' letters are the (approximate) 25th and 75th quantiles, respectively, the second fattest letters the (approximate) 12.5th and 87.5th quantiles and so on. Note that the width of the letters is not related to the underlying data.

## Mapping to the VirtualFlyBrain database

The VirtualFlyBrain (VFB) database[22] curates and extracts information from all publications relating to *Drosophila* neurobiology, especially neuroanatomy. The majority of published neuron reconstructions, including those from the hemibrain, can be examined in the VFB. Each individual neuron (that is, one neuron from one brain) has a persistent ID (of the form VFB_xxxxxxxx). Where cell types have been defined, they have an ontology ID (for example, FBbt_00047573, the ID for the DNa02 DN cell type). Importantly, VFB cross-references neuronal cell types across publications even if different terms were used. It also identifies driver lines to label many neurons. In this paper, we generate an initial mapping providing FBbt IDs for the closest and fine-grained ontology term that already exists in their database. For example, a FlyWire neuron with a confirmed hemibrain cell type will receive a FBbt ID that maps to that exact cell type, while a DN that has been given a new cell type might only map to the coarser term 'adult descending neuron'. Work is already underway with the VFB to assign both ontology IDs (FBbt) to all FlyWire cell types as well as persistent VFB_ids to all individual FlyWire neurons.

## Reporting summary

Further information on research design is available in the Nature Portfolio Reporting Summary linked to this article.

## Data availability

Data artefacts from this paper are available at GitHub (https://github.com/flyconnectome/flywire_annotations). This includes neuron annotations and other metadata; high-quality skeletons for all proofread FlyWire neurons; NBLAST scores for FlyWire versus hemibrain; all-by-all NBLAST scores for FlyWire. The repository may be periodically updated to improve annotations, but older versions will always remain available through GitHub's versioning system. Moreover, neuron annotations and other metadata are also provided in the Supplementary Information. NBLAST scores and skeletons have been deposited in a Zenodo repository (https://doi.org/10.5281/zenodo.10877326)[126]. Connectivity data (for example, synapses table and edge list) are available (https://doi.org/10.5281/zenodo.10676866)[127]. We provide a neuroglancer scene preconfigured for display and query of our annotations alongside the FlyWire neuron meshes and segmentation at http://tinyurl.com/flywire783. Users can add the annotations to arbitrary neuroglancer scenes themselves by adding a data subsource (Extended Data Fig. 11). There are two options: (1) "precomputed://https://flyem.mrc-lmb.cam.ac.uk/flyconnectome/ann/flytable-info-783" containing super class, cell type and side labels; (2) "precomputed://https://flyem.mrc-lmb.cam.ac.uk/flyconnectome/ann/flytable-info-783-all" additionally contains hemi-lineage information. We also provide programmatic access to the annotations through our fafbseg R and Python packages (examples are provided in Table 1 and the online documentation). Annotations

have also been shared with Codex (https://codex.flywire.ai/), the connectome annotation versioning engine (CAVE), which can be queried through the CAVEclient (https://github.com/seung-lab/CAVEclient) and the FAFB-FlyWire CATMAID spaces (https://fafb-flywire.catmaid.org). At the time of writing, access to Codex and CAVE requires signing up using a Google account. To aid a number of analyses, hemibrain neuron meshes were mapped into FlyWire (FAFB14.1) space. These can be co-visualized with FlyWire neurons within neuroglancer (https://tinyurl.com/flywire783; this scene also includes a second copy of the hemibrain data (layer "hemibrain_meshes_mirr"), which have been non-rigidly mapped onto the opposite side of FAFB).

## Code availability

Analyses were performed using open-source packages using both the R natverse[128] and Python navis infrastructures (a summary including links is provided in Table 1). The fafbseg R and Python packages have extensive functionality dedicated to working with FlyWire data, including querying annotations, fetching connectivity and working with the segmentation. Unless otherwise stated, all analyses were performed against the 783 release version (that is, the second public data release for FlyWire).

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

**Acknowledgements** We thank A. Champion and the members of the MRC LMB Scientific Computing group for assistance with compute and web infrastructure; A. McLachlan, R. Court, C. Pilgrim, D. Goutte-Gattat and D. Osumi-Sutherland from the Virtual Fly Brain for helping mapping annotations into their ontology; F. Collman and C. Schneider-Mizell for developing and maintaining the CAVE engine and associated tools; B. Pedigo for discussions as well as help with matching and typing of some FlyWire neurons; we thank the members of R. Wilson's laboratory (A.S.B. with Q. Vanderbeck, A. Li, I. Haber and P. Gibb), who reconstructed the asymmetric body neurons; P. Kandimalla, S. Noselli and the members of R. Wilson's laboratory for pointing out the left/right inversion of FAFB, and P. Kandimalla and S. Noselli for sharing their observations that situs inversus is extremely rare in wild-type Drosophila; L. Luo and J. Macke for comments on an early version of this manuscript; reviewers for suggestions and criticisms on the submitted version; and I. Tastekin and the members of the Ribeiro laboratory for their input on gustatory sensory neuron typing. A.S.B. thanks R. I. Wilson for her support and interest as he finished this project after having moved to the Wilson laboratory. D.S.H. thanks A. Cardona for support and mentoring while in his group. This work was supported by an NIH BRAIN Initiative grant 1RF1MH120679-01 to D.D.B. with G.S.X.E.J.; a Neuronex2 award to G.S.X.E.J. and D.D.B. (NSF 2014862, MRC MC_EX_MR/T046279/1); Wellcome Trust Collaborative Awards (203261/Z/16/Z, 220343/Z/20/Z and 221300/Z/20/Z) and core support from the MRC (MC-U105188491) to G.S.X.E.J.; DFG Walter-Benjamin-Fellowship (STU 793/2-1) to T.S.; EMBO fellowship (ALTF 1258-2020) and a Sir Henry Wellcome Postdoctoral Fellowship (222782/Z/21/Z) to A.S.B.

**Author contributions** P.S., Y.Y., A.S.B., K.E., P.B., M.G., M.d.S., E.J.M., G.B., L.S.C., V.A.S., A.M.C.F., L.K., M.W.P., I.F.M.T., C.R.D., I.S., A.J. and S.F. and the members of the FlyWire consortium contributed proofreading. Y.Y., K.E., P.B., P.S., A.S.B., M.C. and G.S.X.E.J. led the targeted

proofreading effort in Cambridge. S.D., M.M. and H.S.S. led the overall effort. S.D., P.S. and G.S.X.E.J. maintained the proofreading and annotation management platforms. A.R.S., S.-c.Y. and C.E.M. managed the FlyWire community and onboarded new members. P.S., A.S.B., S.D., K.E., P.B., M.G., M.d.S., E.J.M., G.B., L.S.C., V.A.S., A.M.C.F., L.K., M.W.P., M.C., V.H. and G.S.X.E.J. contributed annotations. K.E., P.B. and T.S. provided the cell types for ANs and DNs. A.M. ingested annotations into Codex. D.D.B., E.P. and T.K. developed and hosted CATMAID spaces, the FlyWire supervoxel lookup and FlyWire ⇌ FAFB transform services. S.R.J. developed pyroglancer. P.S., A.S.B. and G.S.X.E.J. developed the R and Python packages to work with the FlyWire and hemibrain datasets. P.S., Y.Y., D.S.H. and G.S.X.E.J. analysed the data and generated the figures. P.S., Y.Y., D.S.H., A.S.B., L.S.C., G.S.X.E.J. and D.D.B. wrote the manuscript with feedback from all of the authors.

**Competing interests** H.S.S. declares a financial interest in Zetta AI. The other authors declare no competing interests.

**Additional information**
**Correspondence and requests for materials** should be addressed to Davi D. Bock or Gregory S. X. E. Jefferis.

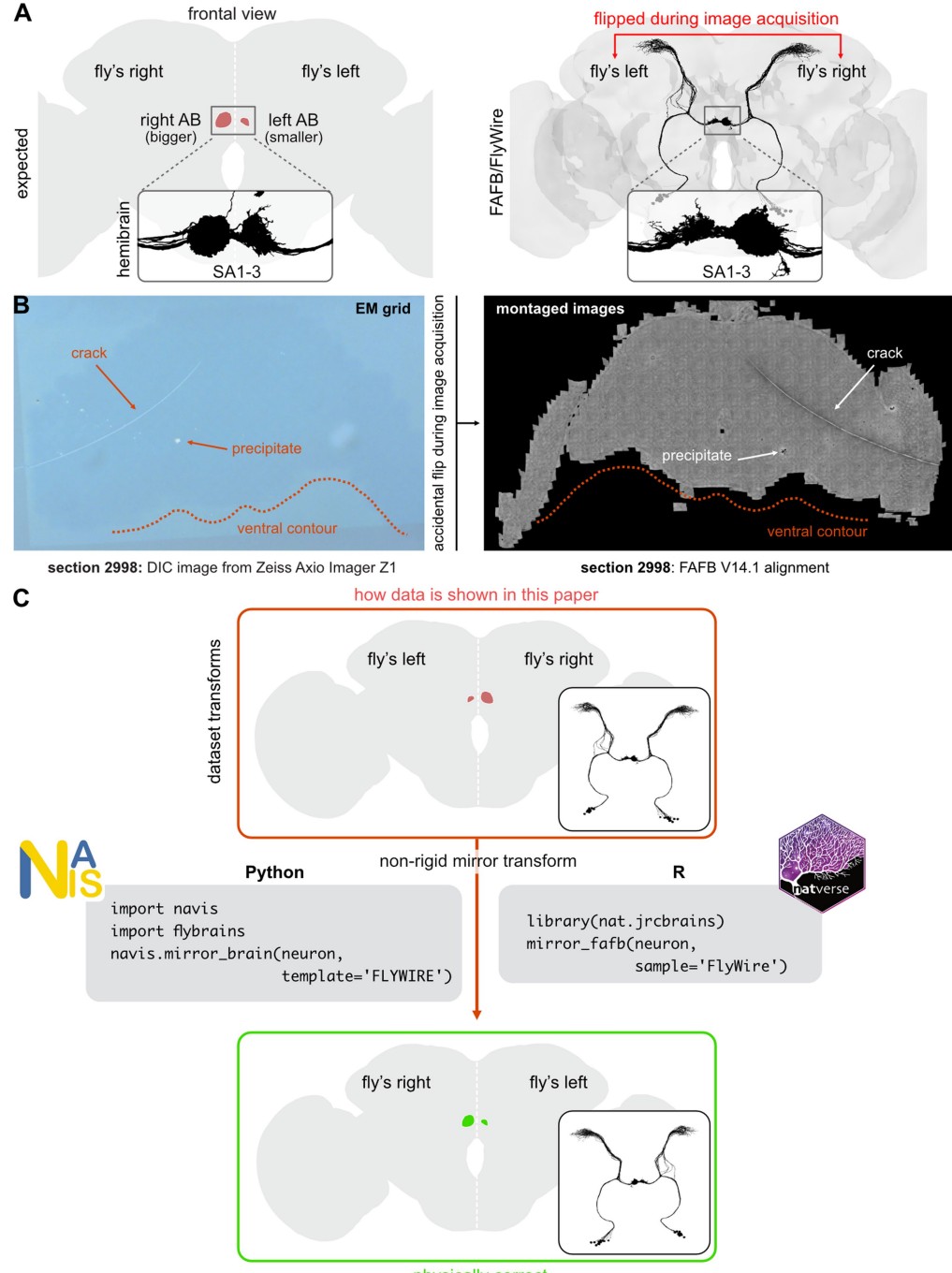

**Extended Data Fig. 1 | Completion of the FlyWire whole-brain connectome and cell typing reveal a left-right inversion of EM image data during acquisition of the underlying FAFB EM dataset. A** Frontal views of the adult fly brain are by convention shown in 2D projection, placing the fly's right on the left of the page. In this view, the asymmetric body (AB), which is nearly always larger on the fly's right[112–114], therefore appears on the left of the page (left panel). During acquisition of the FAFB dataset, image mosaics were acquired and inadvertently stored to disk with the left-right axis inverted. Therefore in frontal view, the right side of the FAFB/FlyWire brain, and the larger AB, appear on the viewer's right (right panel). Insets show axons of SA1-3 neurons, which form the major input to the AB. **B** Direct examination of an original EM-imaged grid using differential interference contrast (DIC) microscopy and an acquired EM mosaic in neuroglancer/catmaid confirms a left-right inversion during image acquisition. A grid with a crack in the support film and staining artefact precipitate was selected in order to provide fiducials easily visible by light microscopy (left panel). These same artefacts can be seen in the EM mosaic (right panel). **C** Showcase of how to programmatically correct the inversion of FAFB/FlyWire data. Due to the large size of the original and derived datasets, it was not technically practical to correct the left-right inversion once it was detected. Therefore this must be corrected post hoc. Code samples show how this can be done for e.g. mesh or skeleton data using Python or R (Methods, "FAFB Laterality").

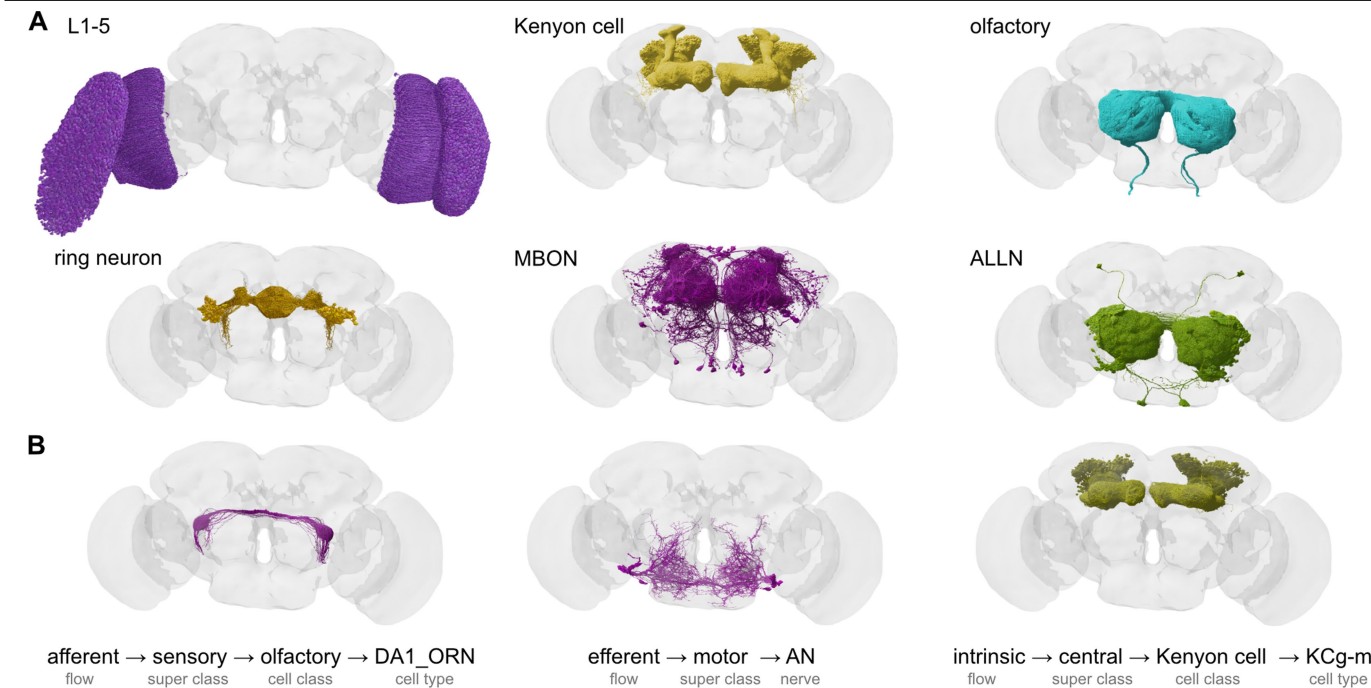

**A** L1-5 · Kenyon cell · olfactory · ring neuron · MBON · ALLN

**B**

afferent → sensory → olfactory → DA1_ORN
flow · super class · cell class · cell type

efferent → motor → AN
flow · super class · nerve

intrinsic → central → Kenyon cell → KCg-m
flow · super class · cell class · cell type

**Extended Data Fig. 2 | Hierarchical annotation examples. A** Examples for cell class annotations. **B** Examples for labels derived from the hierarchical annotations. Abbreviations: ALRN, antennal lobe receptor neuron; MBON, mushroom body output neuron; ALLN, antennal lobe local neuron; ORN, olfactory receptor neuron; AN, antennal nerve.

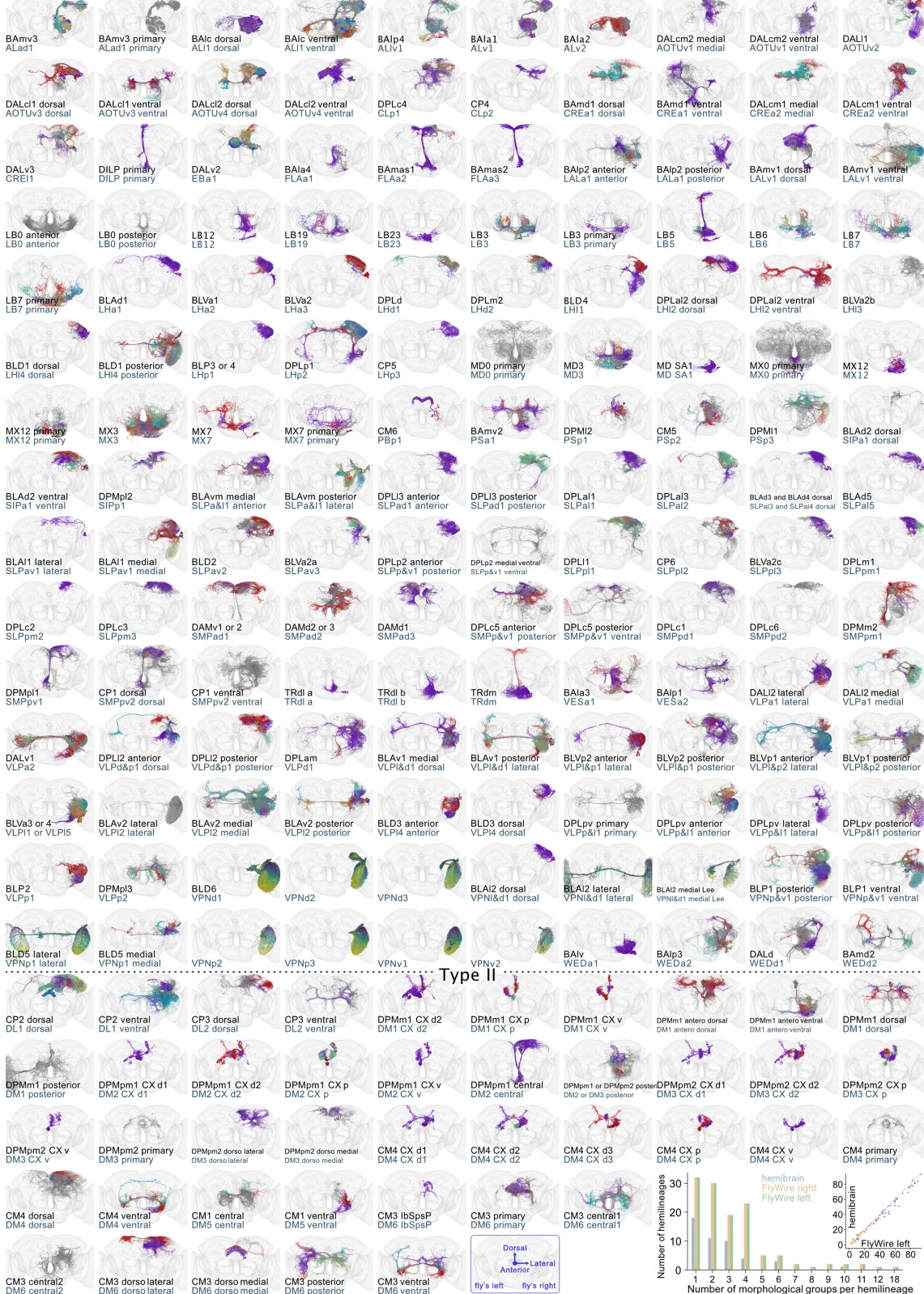

**Extended Data Fig. 3** | See next page for caption.

**Extended Data Fig. 3 | Hemilineage atlas.** Anterior views of neurons within a hemilineage (based on[37,129]), or neurons whose cell bodies form a cluster in a lineage clone (also referred to as "hemilineages" hereafter), based on the light-level data from[31–34,130]. The names of the hemilineages are at the bottom of each panel (top: Hartenstein nomenclature; bottom: ItoLee nomenclature). The snapshots only include neurons with cell bodies on the right hemisphere, and the central unpaired lineages. Except for the hemilineages that tile the optic lobe, the neurons are coloured by morphological groups (see Methods, Hemilineage annotations section). The neurons that form cohesive tracts with their cell body fibres in the Type II lineages (see Methods) are at the lower part of the panels. The last panel of the "Type II" section is for orientation purposes. The bottom right panel is a histogram of the number of morphological groups per hemilineage (blue: hemibrain; orange: FlyWire right; green: FlyWire left). Inset is the number of neurons per hemisphere in each morphological group, with points coloured by their density (yellow: denser). Corresponding group names, together with FlyWire and neuroglancer links are available in Supplementary Files 2 and 3.

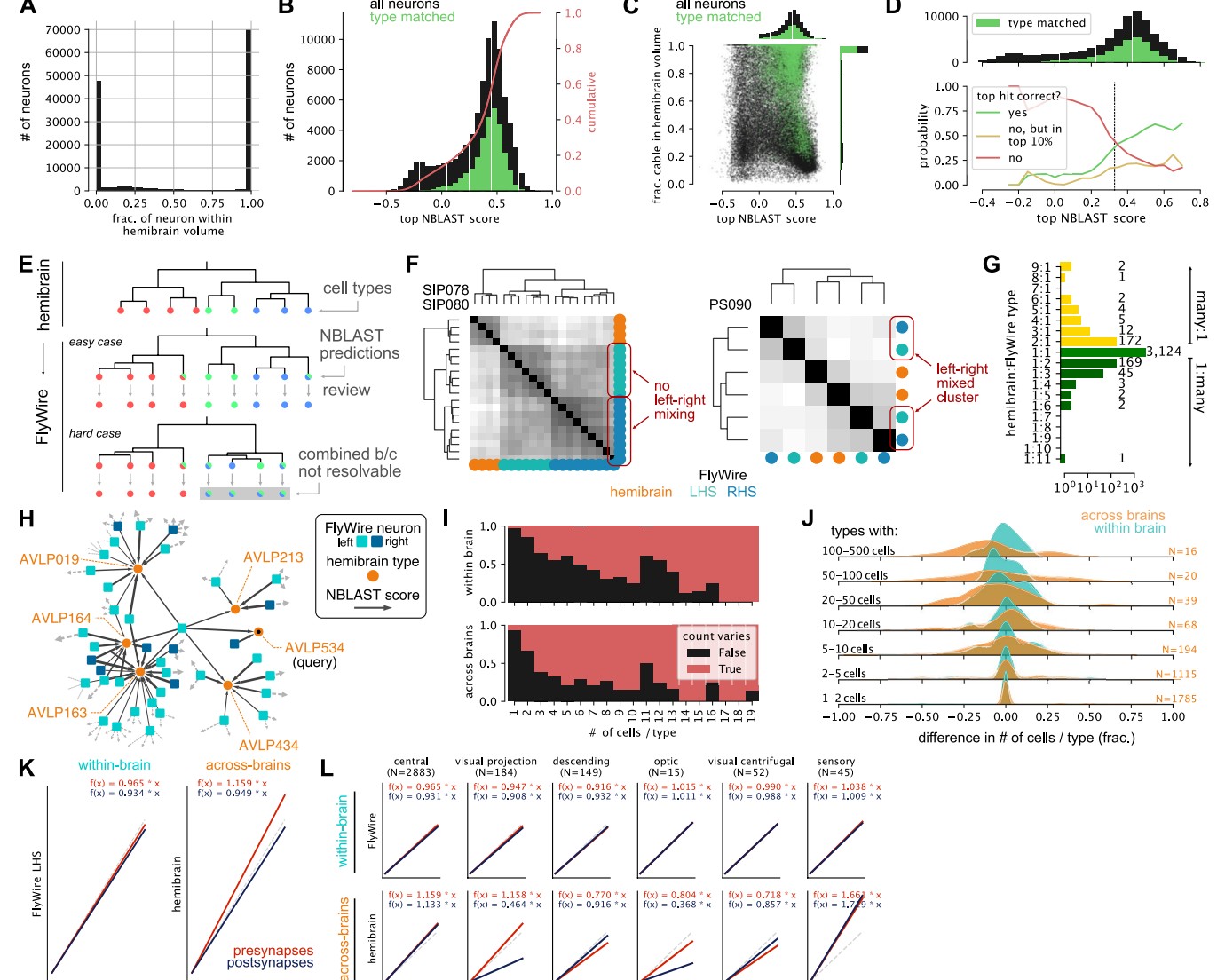

**Extended Data Fig. 4 | Across-brain neuron matching. A** Distribution of the fraction of each FlyWire neuron's cable that is contained within the hemibrain volume: 1 = fully contained; 0 = entirely outside the volume. Note that where necessary FlyWire neurons were transformed onto the opposite side of the brain to better overlap with the hemibrain. **B** Distribution of top FlyWire → hemibrain NBLAST scores. **C** Top NBLAST score vs fraction of neuron contained within hemibrain volume. In a fraction of cases, even heavily truncated neurons can produce good scores and be successfully matched. **D** Top: distribution of top NBLAST scores and fraction which was type matched. Bottom: probability that the correct hit was the top NBLAST hit (green) or at least among (yellow) the top 10% as a function of the top NBLAST score. **E** When some FlyWire neurons had good NBLAST matches against multiple hemibrain cell types, we cross-compared within-dataset morphological clustering (dendrograms). We tried to assign hemibrain types to those ambiguous FlyWire neurons to exactly match clusters in the two dendrograms ("easy case"). When this failed because a cluster in the dendrogram contained clear matches to >1 hemibrain types, we merged types ("hard case"). **F** Cross-brain NBLAST co-clustering for example cell types in Fig. 3: SIP078/SIP080 (left) and PS090 (right). All hemibrain neurons are truncated. The FlyWire PS090

neurons (2 per hemisphere, none truncated) split into two well-separated clusters each containing one left and one right neuron, suggesting that the hemibrain cell type should be split. This is not the case for SIP078/SIP080 where the dendrogram cannot be split into subclusters containing neurons from each hemisphere. **G** Counts for 1:many and many:1 type matches. These also include types derived from previously untyped hemibrain neurons. **H** Extended version of NBLAST hit graph from Fig. 3k. Here, grey dotted arrows indicate matches to types outside of the displayed subgraph. **I** Fraction of cell types showing a difference in cell counts within (left/right, top) and across (bottom) brains. **J** Distribution of cell count differences. **K** Robust linear regression (Huber w/ intercept at 0) for within- and across-dataset pre/postsynapse counts from Fig. 3h. **L** Same data as in K but separated by *superclass*. Slopes are generally close to 1: 1.021 (pre-) and 1.035 (postsynapses, i.e. inputs) between the left and right hemisphere of FlyWire, and 1.176 (presynapses, i.e. outputs) 0.983 (post) between FlyWire and the hemibrain. Note that correlation and slope are noticeably worse for cell types known to be truncated such as visual projection neurons which suggests that we did not fully compensate for the hemibrain's truncation and that the actual across-brain correlation might be even better.

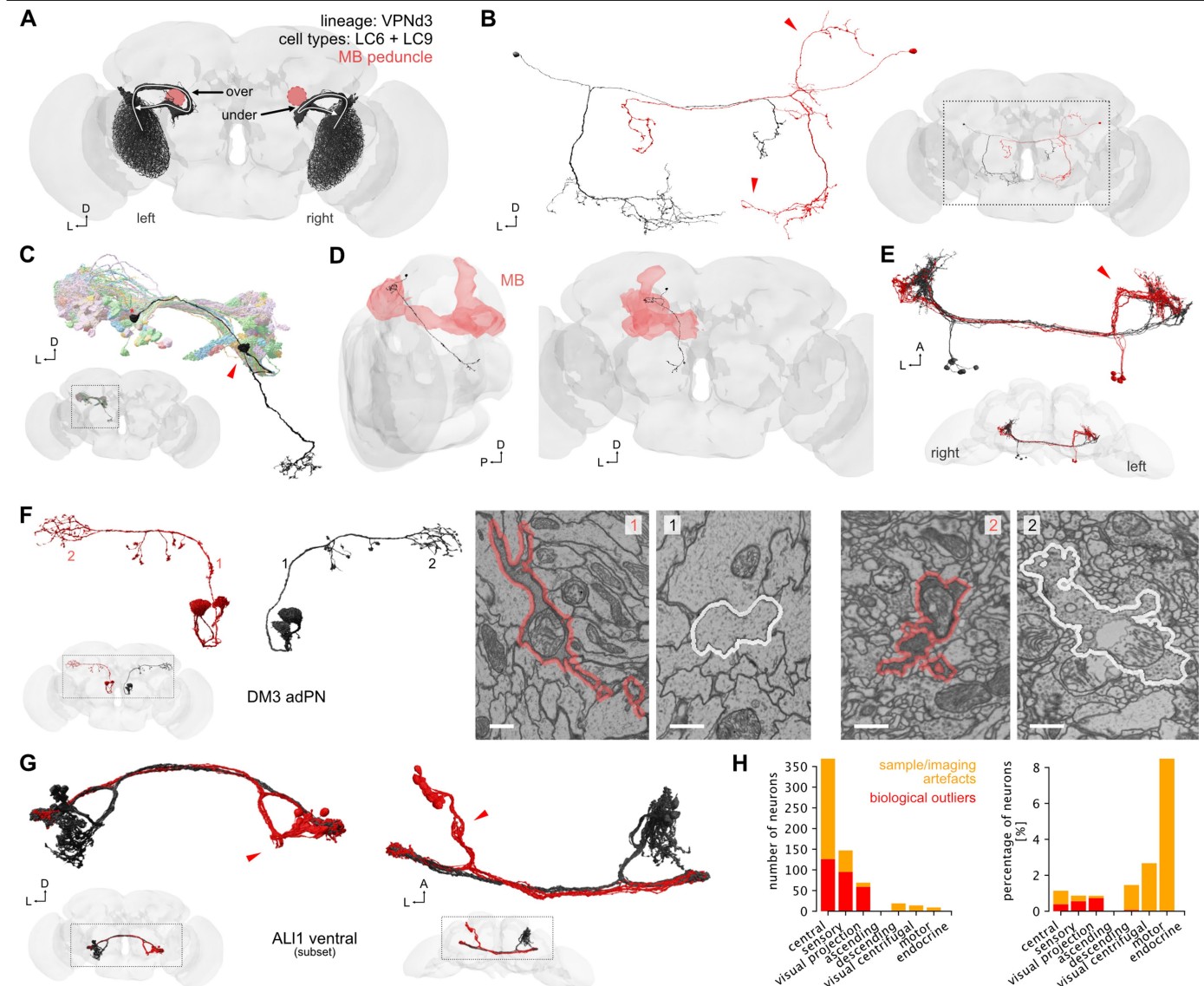

**Extended Data Fig. 5 | Examples of biological outliers and sample artefacts.**
**A** LC6 and LC9 neurons (lineage VPNd3) of the right and left hemispheres take different routes in FlyWire to equivalent destinations (previously reported in[43]). Mushroom body (MB) peduncle is shown in pink. **B** Example of a left/right neuron pair where one side has extra dorsal and smaller ventral dendrites (red arrowheads). **C** A TuBu neuron (black) with correctly placed axon but misplaced ventral dendrites. Regular TuBu neurons shown in background for reference. **D** A single Kenyon Cell whose axon projects outside of the mushroom body, descending through the medial antennal lobe tract. **E** Cell type (CB1029, DM6 ventral hemilineage) where the left neurons' dendrites

(red) take a different tract. **F** Example of sample artefact: the axon of the left DM3 adPN has very dark cytosol which affects both the neuron segmentation as well as synapse detection. Insets compare two locations along the axons between the left and right neurons. **G** A subset of neurons from the ALl1 ventral hemilineage where the right neurons are missing their entire dendrites (red arrow). The exact reason for this is unknown but it is not due to insufficient proofreading. **H** Quantification of recorded outliers and sampling artefacts broken down by super class. Total number of neurons (left) as well as fraction (right) are shown. The number of biological outlier neurons is ~0.4% of the total number of neurons in the brain.

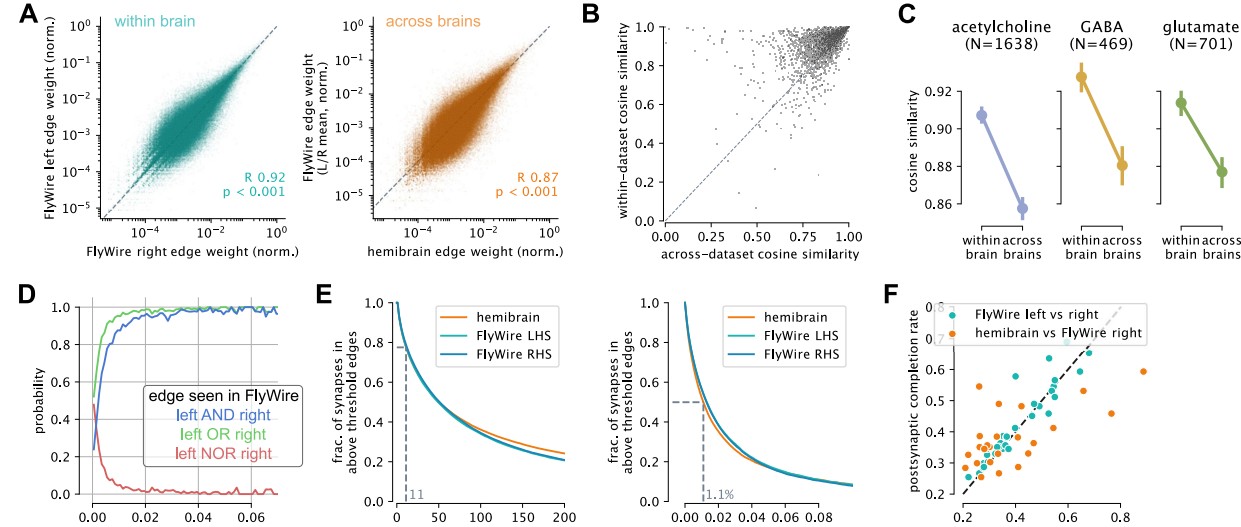

**Extended Data Fig. 6 | Across-brain connectivity. A** Comparison of normalized edge weights within (left) and across (right) brains. **B** Connectivity cosine connectivity similarity within and across brains. Each datapoint is a cell type identified across the three hemispheres. Size correlates with the number of cells per type. **C** Connectivity cosine similarity separated by neurotransmitter. Error bars represent the 95% CI. **D** Probability that an edge present in the hemibrain is found in one, both or neither of the FlyWire hemispheres. **E** Fraction of synapses contained in edges above given absolute (left) and normalized (right) weight. Horizontal lines mark the thresholds for a 90% chance that an edge is found in another hemisphere. **F** Postsynaptic completion rates. Each datapoint is a neuropil.

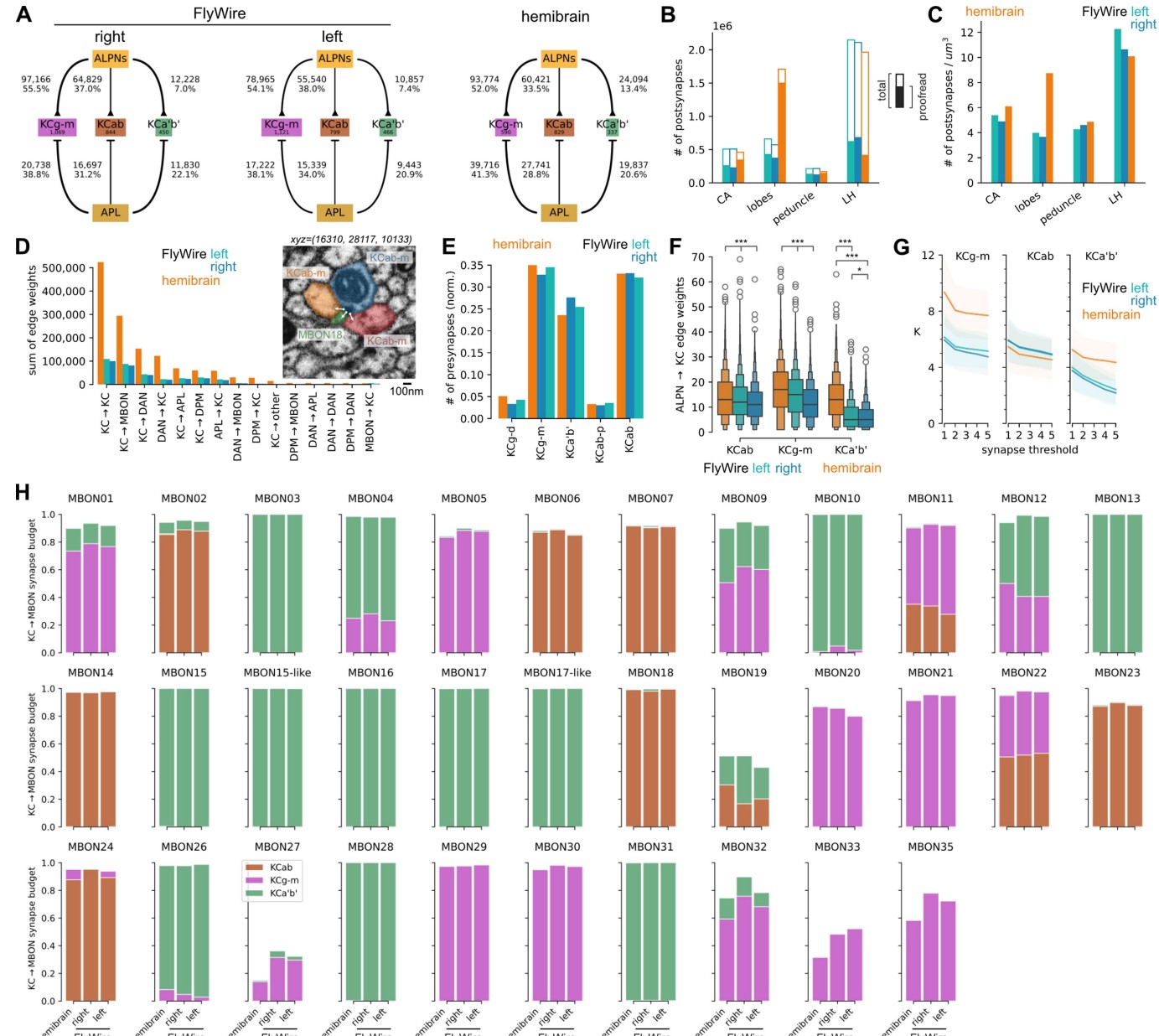

**Extended Data Fig. 7 | Across-brain mushroom body comparison. A** Graph showing ALPN/APL → KC connectivity across the three datasets. Edge labels provide weights both as total synapse counts and normalized to the total output budget of the source. In FlyWire, the mushroom bodies (MB) have 57.2% (left) and 60.7% (right) postsynaptic completion rate while the hemibrain MB has been proofread to 81.3% (see also B). To compensate for this we typically used normalized synapse counts and edge weights. Note that KCab act as an internal control as their numbers are consistent across all hemispheres and we don't expect to see any changes in their connectivity. **B** Total versus proofread postsynapse counts across MB compartments. Lateral horn (LH) shown for comparison. **C** Postsynapse density across MB compartments. **D** Connectivity between different MB cell classes. Inset shows an example of KC → KC and KC → MBON synapse in the hemibrain. **E** Presynapse counts per KC type normalized to the total number of KC synapses per dataset. **F** ALPN → KC edge weights. See Methods for details on enhanced box plots. **G** *K* (# of ALPN types providing input to a single KC) under different synapse thresholds. **H** Fraction of MBON input budget coming from individual KCab, KCg-m and KCa'b'. Abbreviations: CA, calyx; DAN, dopaminergic neuron; ALPN, antennal lobe projection neuron; KC, Kenyon Cell; MBON, mushroom body output neurons. Kolmogorov-Smirnov test (F): *, p < =0.05; **, p < =0.01; ***, p < =0.001.

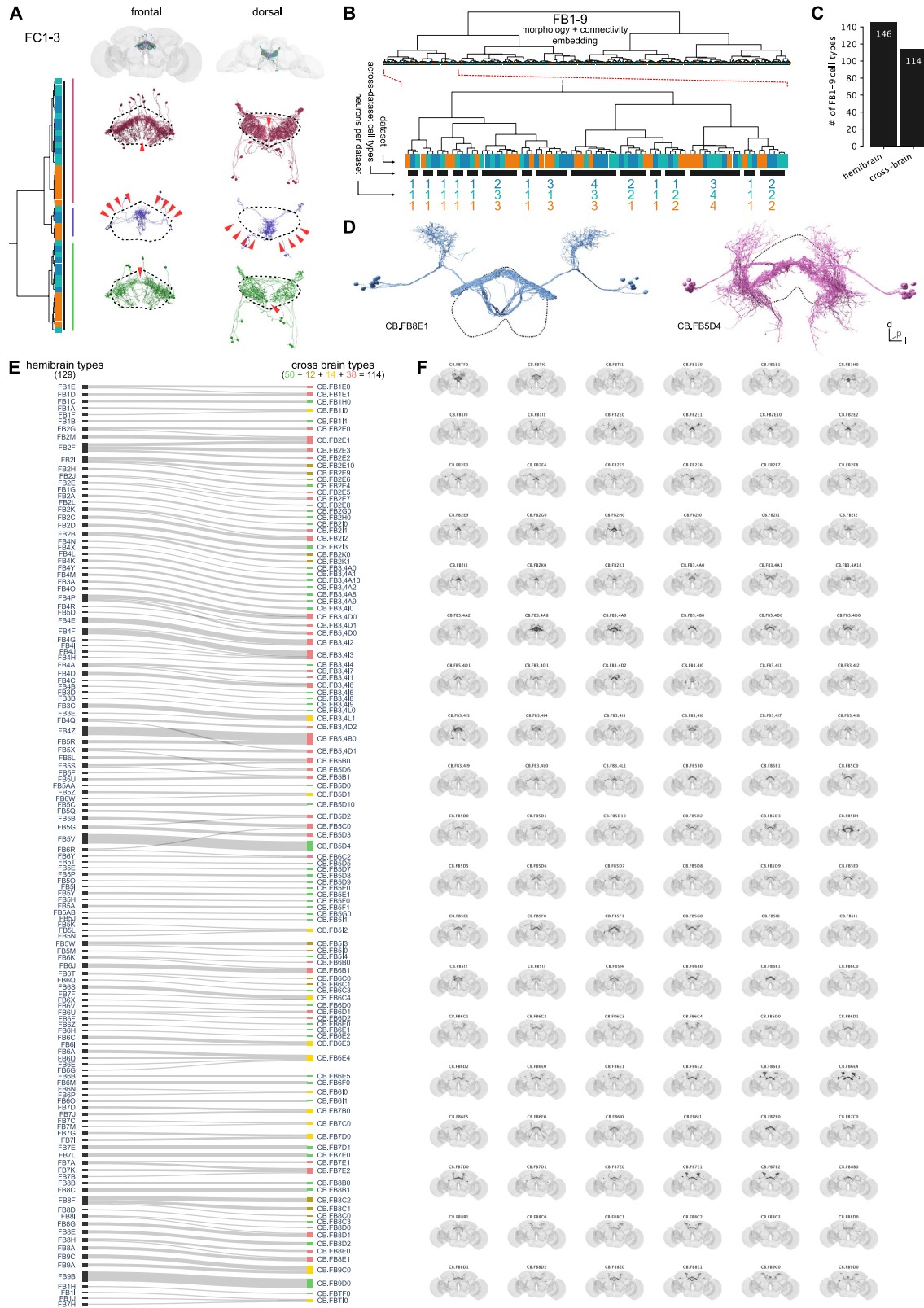

**Extended Data Fig. 8 | Across-brain co-clustering. A** FC1-3 across-brain cluster from Fig. 6d (asterisk) that was manually adjusted. This group consists of three sub-clusters that technically fulfil our definition of cell type. They were merged, however, because they individually omit columns of the fan-shaped body (arrowheads) and are complementary to each other. **B** Hierarchical clustering from combined morphology + connectivity embedding for FB1-9. Zoom-in shows cross-brain cell type clusters. **C** Number of hemibrain vs cross-brain FB1-9 cell types. **D** Examples from the FB1-9 cross-brain cell typing. Labels are composed from CB.FB{*layer*}{*hemilineage-id*}{*subtype-id*}; fan-shaped body outlined. **E** Flow chart comparing FB1-9 hemibrain and cross-brain cell types. Colours correspond to 1:1, 1:many, many:1 and many:many mappings between hemibrain and cross-brain cell types. **F** Renderings of all FB1-9 cross-brain cell types.

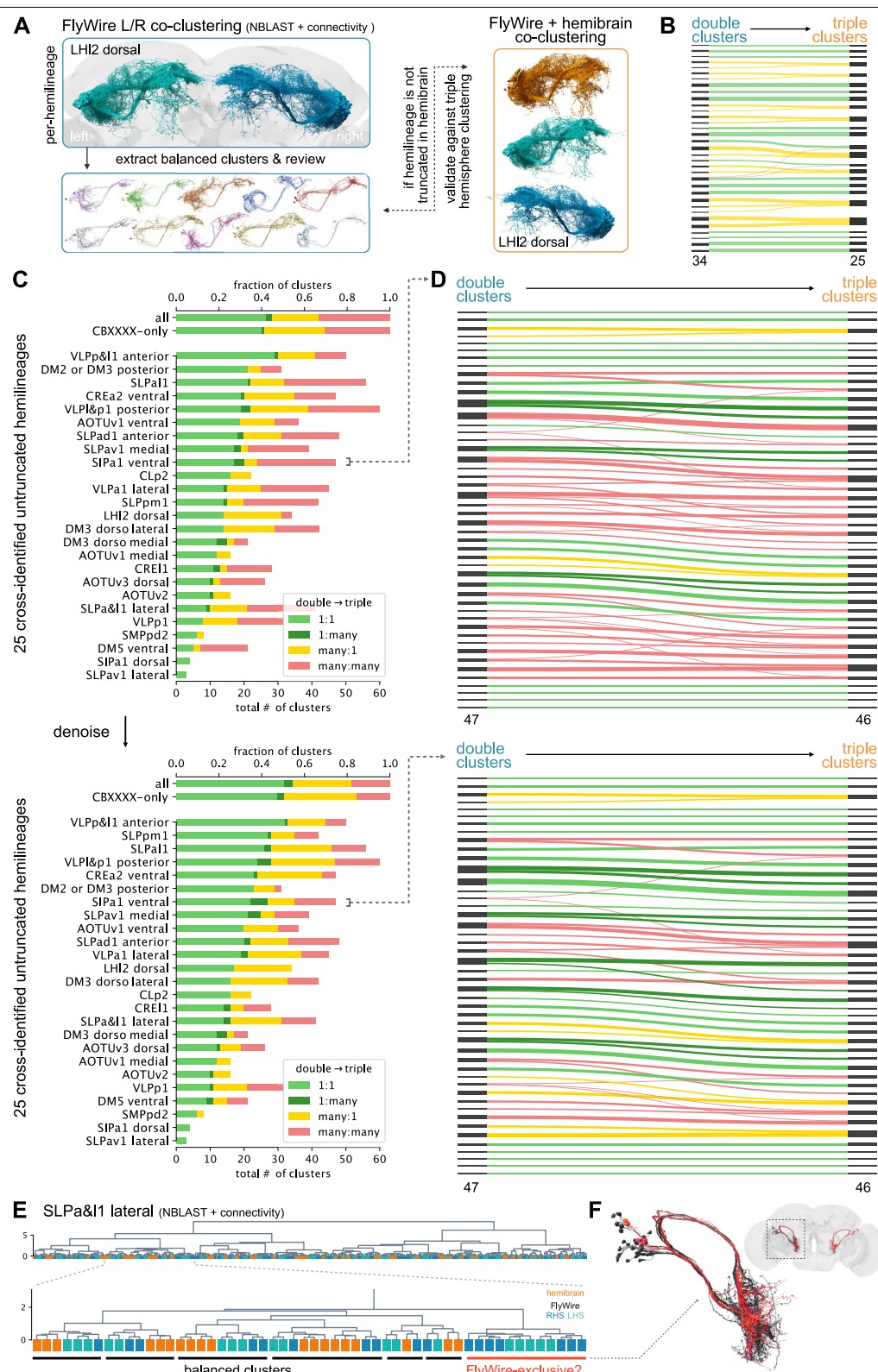

**Extended Data Fig. 9 | Double vs triple co-clustering analyses. A** Pipeline for comparing putative cell types from double (FlyWire left/right) and triple (FlyWire + hemibrain) hemisphere co-clustering. **B** Flow chart for hemilineage LHl2 dorsal illustrating how individual FlyWire neurons move between double and triple clusters. Black bars represent clusters; thickness is proportional to the number of neurons in each cluster. **C** Summary over all 25 hemilineages that were cross-identified and are untruncated in the hemibrain connectome. Top bar chart shows unfiltered results; bottom chart shows results after denoising (removal of single neurons that cause many:many mapping because they swap clusters). **D** Flow chart for example hemilineage SIPa1 ventral. Unfiltered (top) and denoised (bottom). **E-F** Example of a cluster (red in panel F) from hemilineage SLPa&l1 lateral that only seems to exist in FlyWire although similar balanced clusters (black) are present in both datasets.

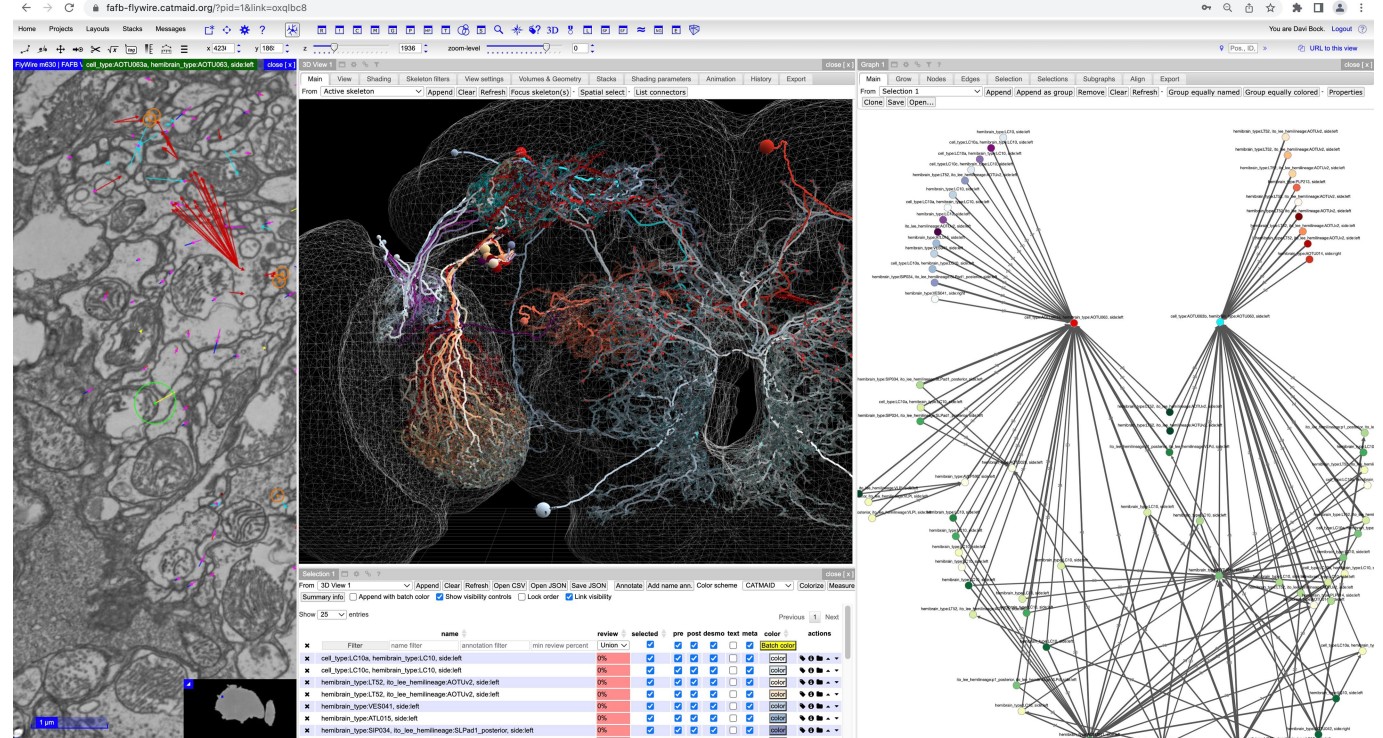

**Extended Data Fig. 10 | CATMAID spaces.** Screenshot demonstrating the use of CATMAID Spaces (https://fafb-flywire.catmaid.org/) to interrogate the FlyWire connectome. Differential inputs to AOTU63a and b are visualized (red and cyan, respectively). The Graph widget was used to show all neurons making 20 or more synapses onto AOTU63a and b, and to show only >=20 synapse connections between these neurons. Neurons whose *only* >=20 synapse connection was to either AOTU63a or b (but not both) were differentially coloured (blue-purples and greens, respectively).

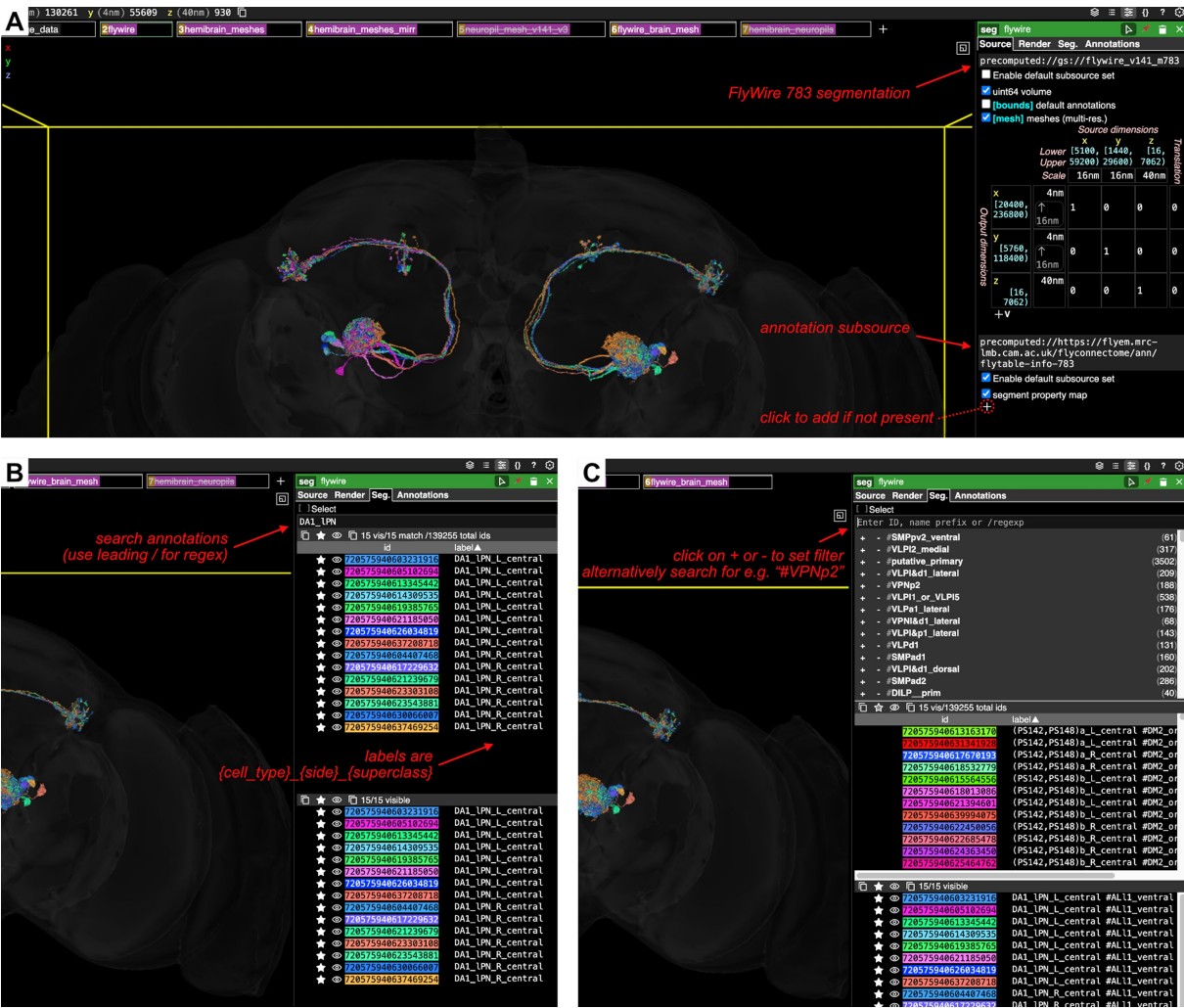

**Extended Data Fig. 11 | Annotations in Neuroglancer. A** Screenshot of neuroglancer with FlyWire 783 segmentation layer with "flytable-info-783" annotation layer subsource (scene pre-configured at http://tinyurl.com/ flywire783). **B** Example for querying annotation. **C** Example for subsource "flytable-info-783-all" which includes hemilineage annotations.

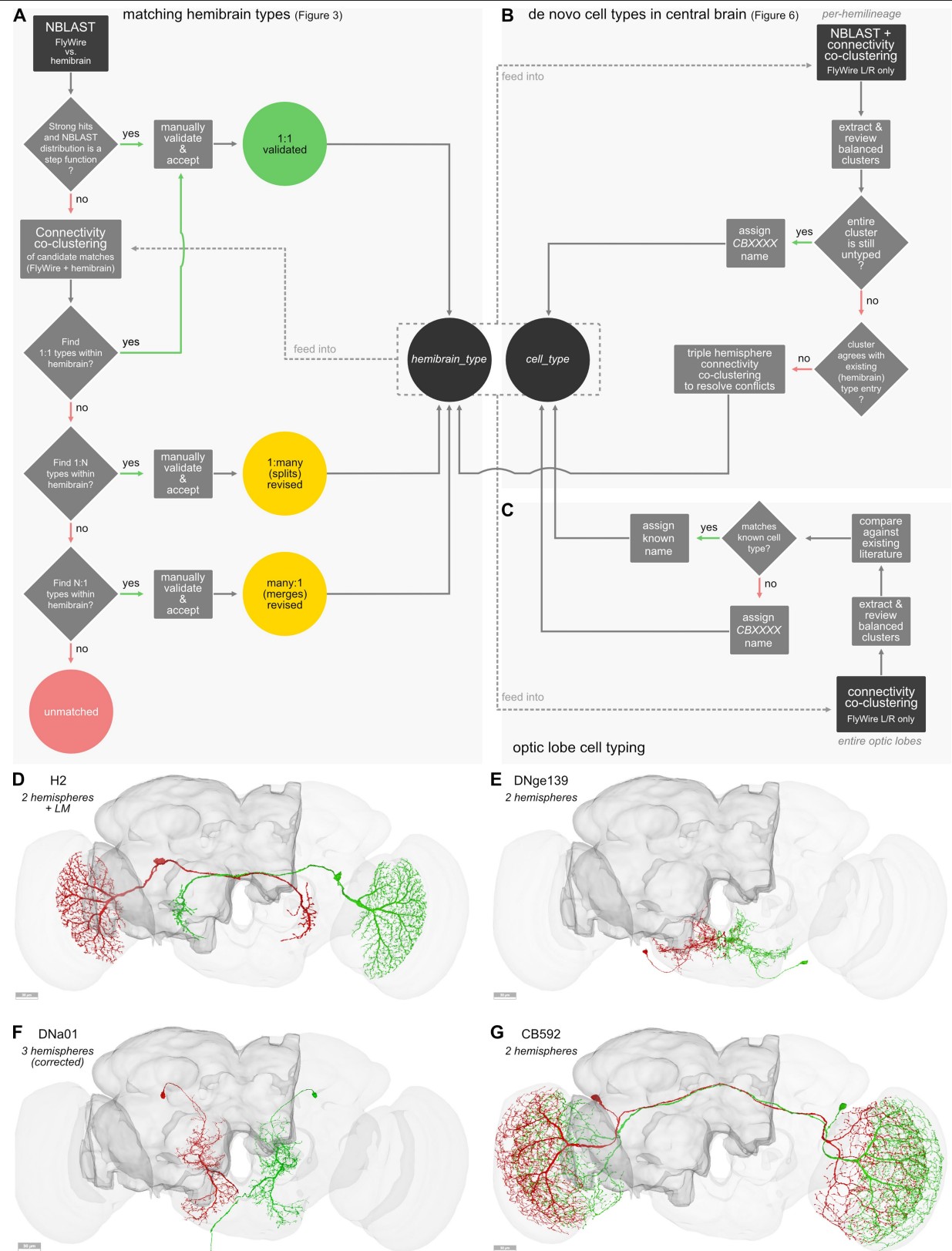

**Extended Data Fig. 12 | Matching workflow. A** Workflow for matching hemibrain types to FlyWire neurons. **B** Workflow for generation of *de-novo* cell types used to fill the gaps left from the hemibrain type matching. **C** Workflow for cell typing in the optic lobes. **D-G** Examples of cell types. H2 is based on left vs right FlyWire clustering plus existing LM data; DNge139 and CB592 are based solely on left vs right FlyWire clustering; DNa01 is based on three hemispheres worth of data but was misidentified as "VES006" in the hemibrain.

# Reporting Summary

## Statistics

For all statistical analyses, confirm that the following items are present in the figure legend, table legend, main text, or Methods section.

| n/a | Confirmed | |
|---|---|---|
| ☐ | ☒ | The exact sample size (*n*) for each experimental group/condition, given as a discrete number and unit of measurement |
| ☒ | ☐ | A statement on whether measurements were taken from distinct samples or whether the same sample was measured repeatedly |
| ☐ | ☒ | The statistical test(s) used AND whether they are one- or two-sided *Only common tests should be described solely by name; describe more complex techniques in the Methods section.* |
| ☐ | ☒ | A description of all covariates tested |
| ☒ | ☐ | A description of any assumptions or corrections, such as tests of normality and adjustment for multiple comparisons |
| ☐ | ☒ | A full description of the statistical parameters including central tendency (e.g. means) or other basic estimates (e.g. regression coefficient) AND variation (e.g. standard deviation) or associated estimates of uncertainty (e.g. confidence intervals) |
| ☐ | ☒ | For null hypothesis testing, the test statistic (e.g. *F*, *t*, *r*) with confidence intervals, effect sizes, degrees of freedom and *P* value noted *Give P values as exact values whenever suitable.* |
| ☒ | ☐ | For Bayesian analysis, information on the choice of priors and Markov chain Monte Carlo settings |
| ☒ | ☐ | For hierarchical and complex designs, identification of the appropriate level for tests and full reporting of outcomes |
| ☐ | ☒ | Estimates of effect sizes (e.g. Cohen's *d*, Pearson's *r*), indicating how they were calculated |

*Our web collection on statistics for biologists contains articles on many of the points above.*

## Software and code

Policy information about availability of computer code

| Data collection | Data collection is described in our companion paper by Dorkenwald et al. and is cited at appropriate locations throughout our manuscript. |
|---|---|
| Data analysis | For analysis we developed open-source software packages. These tools are detailed in the Methods, which also includes download locations, all of which are on Github. The key software packages are: - navis: https://github.com/navis-org/navis v1.5.0 - fafbseg-py: https://github.com/navis-org/fafbseg-py v3.0.5 - flybrains: https://github.com/navis-org/navis-flybrains v0.2.9 - skeletor: https://github.com/navis-org/skeletor v1.2.3 - fafbseg: https://github.com/natverse/fafbseg v0.14.0 - coconatfly: https://github.com/natverse/coconatfly v0.1.0 |

For manuscripts utilizing custom algorithms or software that are central to the research but not yet described in published literature, software must be made available to editors and reviewers. We strongly encourage code deposition in a community repository (e.g. GitHub). See the Nature Portfolio guidelines for submitting code & software for further information.

## Data

Policy information about availability of data

All manuscripts must include a data availability statement. This statement should provide the following information, where applicable:

- Accession codes, unique identifiers, or web links for publicly available datasets
- A description of any restrictions on data availability
- For clinical datasets or third party data, please ensure that the statement adheres to our policy

Data artefacts from this paper are available at https://github.com/flyconnectome/flywire_annotations.

This includes:
- neuron annotations + other metadata
- high quality skeletons for all proofread FlyWire neurons
- NBLAST scores for FlyWire vs. hembrain
- all-by-all NBLAST scores for FlyWire
The repository may periodically be updated to improve annotations but older versions will always remain available via Github's versioning system.

In addition, neuron annotations + other meta data are also available for download in the supplementary materials; NBLAST scores and skeletons have been deposited in a Zenodo repository: https://zenodo.org/records/10877326 (doi: 10.5281/zenodo.10877326).

We provide a neuroglancer scene preconfigured for display and query of our annotations alongside the FlyWire neuron meshes and segmentation at http://tinyurl.com/flywire783. Users can add the annotations to arbitrary neuroglancer scenes themselves by adding a data subsource (see Extended Data Figure 11). There are two options:
"precomputed://https://flyem.mrc-lmb.cam.ac.uk/flyconnectome/ann/flytable-info-783" containing super class, cell type and side labels
"precomputed://https://flyem.mrc-lmb.cam.ac.uk/flyconnectome/ann/flytable-info-783-all" additionally contains hemi-lineage information

We also provide programmatic access to the annotations through our fafbseg R and Python packages (see Table 1 and the online documentation for examples).

Annotations have also been shared with Codex (https://codex.flywire.ai/), the connectome annotation versioning engine (CAVE) which can be queried through e.g. the CAVEclient (https://github.com/seung-lab/CAVEclient), and the FAFB-FlyWire CATMAID spaces (https://fafb-flywire.catmaid.org). At the time of writing access to Codex and CAVE requires signing up using a Google account.

To aid a number of analyses, hembrain neuron meshes were mapped into FlyWire (FAFB14.1) space. These can be co-visualised with FlyWire neurons within neuroglancer (e.g. https://tinyurl.com/flywire783; this scene also includes a second copy of the hembrain data (layer hembrain_meshes_mirr) which has been non-rigidly mapped onto the opposite side of FAFB).

## Research involving human participants, their data, or biological material

Policy information about studies with human participants or human data. See also policy information about sex, gender (identity/presentation), and sexual orientation and race, ethnicity and racism.

| | |
|---|---|
| Reporting on sex and gender | Does not apply. |
| Reporting on race, ethnicity, or other socially relevant groupings | Does not apply. |
| Population characteristics | Does not apply. |
| Recruitment | Does not apply. |
| Ethics oversight | Does not apply. |

Note that full information on the approval of the study protocol must also be provided in the manuscript.

## Field-specific reporting

Please select the one below that is the best fit for your research. If you are not sure, read the appropriate sections before making your selection.

☒ Life sciences   ☐ Behavioural & social sciences   ☐ Ecological, evolutionary & environmental sciences

For a reference copy of the document with all sections, see nature.com/documents/nr-reporting-summary-flat.pdf

## Life sciences study design

All studies must disclose on these points even when the disclosure is negative.

| | |
|---|---|
| Sample size | This study analyses and compares the connectomes of two Drosophila brains. |

| Data exclusions | No neurons were excluded from overall annotation or analyses. For the analysis of across brain stereotypy we used a subset of the available matches; the specific exclusion criteria and rationale are clearly detailed in the methods. |
| Replication | Does not apply. |
| Randomization | Does not apply. |
| Blinding | Does not apply. |

# Reporting for specific materials, systems and methods

We require information from authors about some types of materials, experimental systems and methods used in many studies. Here, indicate whether each material, system or method listed is relevant to your study. If you are not sure if a list item applies to your research, read the appropriate section before selecting a response.

### Materials & experimental systems

| n/a | Involved in the study |
|---|---|
| ☒ ☐ | Antibodies |
| ☒ ☐ | Eukaryotic cell lines |
| ☒ ☐ | Palaeontology and archaeology |
| ☒ ☐ | Animals and other organisms |
| ☒ ☐ | Clinical data |
| ☒ ☐ | Dual use research of concern |
| ☒ ☐ | Plants |

### Methods

| n/a | Involved in the study |
|---|---|
| ☒ ☐ | ChIP-seq |
| ☒ ☐ | Flow cytometry |
| ☒ ☐ | MRI-based neuroimaging |

## Plants

| Seed stocks | Does not apply. |
| Novel plant genotypes | Does not apply. |
| Authentication | Does not apply. |

