## [Peer Review File · Nature]

Manuscript Title: Whole-brain annotation and multi-connectome cell typing of *Drosophila*

Reviewer Comments & Author Rebuttals

Reviewer Reports on the Initial Version:

Referees' comments:

Referee #1 (Remarks to the Author):

In this paper, Schlegel and colleagues describe methods they developed and used to annotate neurons in a new whole brain connectome of adult *Drosophila melanogaster*. Importantly, they also perform several analyses that compare the connectivity and morphology of neurons across the two halves of the whole brain dataset and with another partial brain dataset already annotated. In particular, through these comparisons, the authors: develop heuristics for judging the importance of edges (these will be specific to fly connectomes); develop new definitions for cell types (which will likely be important beyond fly connectomes); and analyze surprising differences between the two extant *Drosophila* adult brain connectomes (unclear how these sorts of stereotypy differences will affect analyses of connectomes in other animals). The methods developed here are central to making the flywire dataset useful to the field, and some, like how to define cell types across brains, will likely be of broad interest as larger and larger connectomic datasets become available in other animals.

Major

1) Database issue: Overall in flywire, it seems confusing that each neuron can have multiple community annotations with no formatting consistency, and there does not seem to be a process outlined for going from community annotations to a single, unique cell type field with a standard format. As an example, Tlp6 is labeled multiply (the difference is using the word “with”) as:

- a. #temp_NewType Translobula-plate 6; Tlp6 (flattened LOP arbors in layer 3 only, and small, sparse, arbors in the LO)
- b. #temp_NewType Translobula-plate 6; Tlp6 (with flattened LOP arbors in layer 3 only, and small, sparse, arbors in the LO)

These sorts of unique annotations are searchable, but require a lot of parsing, and it seems like it would be important to have single, concise, unique cell type signifiers for new cells that do not have corresponding cell types in the hemibrain database.

2) This issue of the left-right flip in the original data is really unfortunate and I understand that the authors are doing their best to rectify the situation, which appears to be more complicated than this reviewer understands. The entire discussion has still left me confused in quite a few situations about what is truly the left side. One case in point is Figure 3A, in which it looks like I’m seeing the brain from the front of the fly, but on which the reader’s left is labeled “left”, which I would expect to be the right side of the fly’s brain. Does this mean that the brain is being presented as mistakenly reversed, but then correctly labeled? (Or am I mistaken about the brain’s orientation?) Overall, I

found that the discussion of what was done about this remained confusing, and I was also confused about where in this paper the neurons would be drawn correctly vs. only labeled correctly. This issue also extends to what future users should plan to do; do the authors advocate this paper's procedure as the clearest way to deal with this issue moving forward?

3) I think I may have been more taken aback than the authors by the numbers of unmatched neurons and neuron types between flywire and the hemibrain.

a. From Figure 3C, it looks as though 16k neurons in flywire that share the hemibrain volume were untyped and 23k more that partially share the volume ("truncated") were untyped. Conversely, almost half of the neuron types identified in the hemibrain volume do not show up at all in flywire. I think these differences are rather extraordinary and deserve a bit more attention. The authors spin the shared neuron types positively, but I think this result could equally be spun negatively. Readers should learn more about these untyped flywire neurons and unmatched hemibrain neuron types — the paper currently focuses on the successes, but should analyze what happened in these unmatched cases so we can understand the discrepancies between these two vast datasets.

i. For instance, are these untyped cells more likely to come from specific hemilineages? What do these untyped neurons have in common that they didn't appear in the hemibrain dataset? Is this variability in neuron existence or just neuron morphology? Can one count cell bodies to ask about whether cell numbers are drastically different?

ii. Do any of the unmatched neuron types in the hemibrain have split Gal4 lines, or matching morphologies among the single cell clones of the generation 1 gal4 collection? If so, is their reported expression variable (all or nothing?) among flies? If some unmatched neuron types appear to exist in Gal4 collections but not flywire, how should we approach this discrepancy?

iii. How can we better think about this variability beyond just different growth conditions? Are there other experiments the authors could suggest that could help us better understand what's going on but that don't require decade-long EM volume analyses?

iv. Will any of the differences be attributable to the anisotropic voxels in the flywire dataset compared to the isotropic ones in the hemibrain?

b. If the hemibrain made a policy of splitting when in doubt, it's unclear to me why Figure 3D has so few neurons in the many:1 category.

4) I think adding more analyses on the cross-hemisphere similarities and differences would be a valuable contribution. In particular, I was left wondering what features/neuron morphologies/connectivities were most conserved across the hemispheres of the brain, and which were the most variable? For instance, the authors highlight LC6 and LC9 morphology differences across this brain, but there must be a spectrum of such things, and I would expect some regions/cell types to show more variability than others, and some features to show more variability than others (for instance, gross neural morphology vs. connectivity vs. synapse placement). Knowing what regions or cells are most and least variable seems likely to suggest specific hypotheses about the importance of stereotypy across the brain for different circuits.

Minor

1) On page 2, the authors cite the rich behavior of adult flies compared to larvae. (Reminder that reviewers appreciate line numbers.) I agree, but the citations here do not really seem to be

representing rich behavior per se, but rather papers that often mix circuit dissection and behavioral results. Instead, it might be nice to use the opportunity to focus on papers that are really elucidating interesting flight, motor, or walking behaviors: grooming, navigation, courtship, escape, learning, walking coordination, etc. There are lots of papers to choose from here – but perhaps highlight ethological and behavioral studies here to better support the authors' point?

2) I may have missed this here or in other papers, but has the comparison been made of the randomness of the KC inputs between the two hemispheres? It seems important to understand if the randomness is brain to brain or between hemispheres, since the two options imply different mechanisms and functional outcomes...

3) Line XXX: "Connectivity-based typing is typically used recursively" – what does "recursively" mean in this case? Iteratively? This seems to be getting quite into the weeds on methods, for a general audience.

4) Figure 1 has a *lot* of overlap with Dorkenwald et al. Figure 2. I'm not sure what to do about that, but they look and feel awfully similar.

5) Figure 2I – needs more description. How did the authors find the elbow? If a log transform is applied to the y-axis, is the elbow still in the same place? How much does it matter where this cut is made?

Referee #2 (Remarks to the Author):

As a companion to the paper by Dorkenwald et al., which reports the generation of the first adult whole-brain connectome, this paper by Schlegel et al. describes the annotation of the connectome and analysis of stereotypy within and across brains. The study addresses broad and important questions regarding how we should interpret not only the newly completed *Drosophila* connectome, but connectomes in general – most notably, how cell types should be defined, to what extent a connectome is stereotyped across animals, and how technical versus biological noise contribute to connectome variability. This work provides the foundation for researchers to make use of present and future connectomes, fulfilling a critical need as connectomics has already begun to revolutionize the study of neural circuits. Without question, I would recommend that this study is published in Nature alongside the companion paper.

Technically, this study is rigorous and comprehensive. Every time I wanted to suggest an additional analysis or plot, I found that it was already included somewhere. The authors have been thoughtful and meticulous in addressing difficult questions, particularly the question of how to define a cell type. The text does a nice job of putting the new cell type atlas into perspective and describing the advances it represents over previous datasets. Some comments and suggestions are below.

1. Do the authors have any ideas for why they were unable to find two of the known hemilineages?
2. This is a comment on Codex (flywire.ai), not the manuscript itself: I think it should be more clear

how the images in the database are inverted but the side annotations are not.

3. The authors report that the “connectivity types” in the hemibrain were generally not reproducible in Flywire. How should we interpret this? E.g. does this suggest that “connectivity types” represent an over-clustering of the data, as alluded to in the Discussion, or that connectivity patterns within a morphology class are real but not stereotyped across brains?

4. Could the authors say a bit more about the potential reasons for the difference in mushroom body postsynaptic counts between the hemibrain and Flywire? If it’s a technical bias in automated synapse detection, shouldn’t this difference be lower for proofread synapses where errors are corrected? It could be helpful to provide an image of the “unusual” synapses being referenced.

5. In Figure 5, I think statistics should be presented to support the authors’ conclusions.

6. The cell-typing approach in Figure 6 seems closely related to Figure 3, and seems a bit out of place at the end after the discussion of technical vs. biological variability and the mushroom body analysis. I would consider moving it earlier.

Minor comments:

p. 11: May want to clarify that any number of synapses (even 1) can define an edge, because this is different from the companion paper by Dorkenwald et al. where connections are always defined as at least 5 synapses.

p. 12 and Fig. 4D: The numbers for within-brain edge comparisons don’t match between the text and figure (57/59% vs. 58/60%).

p. 12: “edges of >10 synapses can be reproducibly found (>90%)” If this is referring to Fig. 4E I would suggest being more specific that edges of >10 synapses in the hemibrain are found in at least one Flywire hemisphere; otherwise it is unclear what “reproducibly found” means. Similar comments in a couple other places, e.g. does “chance of persisting” mean it must be found in more than one hemisphere or all three hemispheres?

p. 21: typo “have be”

p. 22: “Our criterion for strong (highly reliable) edges retains between 7-16% of edges and 50-70% of synapses.” I think this statement (mainly the word “retains”) is confusing because the numbers could be interpreted to be the percent of edges/synapses that are conserved across brains, not the percent that meet the “strong” threshold.

p. 39: Please define ECM

p. 40: Examples of cell classes are shown Table S3, not S2

Fig. S3E-F: These figures were hard to interpret and I am not sure I interpreted them accurately. It would be helpful to explain more clearly in the legend how the depicted clusters relate to the idea that SIP078 and SIP080 needed to be merged and PS090 needed to be split.

Fig. S4E: I am unclear about why the dotted lines were drawn at these values. Is an edge weight of 17 important for some reason?

Referee #3 (Remarks to the Author):

Schlegel et al reports on efforts to annotate the complete connectome of the flywire full adult fly brain (FAFB) reported Dorkenwald et al 2023, and to compare this new complete connectome to previously published partial connectome of the central brain (hemibrain). This is a landmark study. The hierarchical clustering of neurons into cell types and the assignment to developmental hemilineages is an incredible addition to the raw connectome and enables a deeper understanding of this unprecedented whole brain connectome dataset. Further, the pioneering comparative connectomic analyses between the two hemispheres of the flywire connectome and the hemibrain connectome provide insight into the degree of stereotypy exhibited by this insect brain. Finally, on a practical note, the paper develops a quantitative framework for two fundamental criteria: (1) what constitutes a cell type, and (2) what constitutes a statistically significant connection. Both will be practically useful in this new connectomic era.

Major comment 1

Cell types are defined as groups of cells that are quantitatively more similar to cells in a different brain than to any other cell in the same brain. I like this definition. But operationalizing this definition requires defining a quantitative notion of similarity between neurons and different choices will of course give rise to different cell types. Much of the analysis in the paper relies on the chosen definition of similarity being a good one.

My understanding is that morphological match (NBLAST score) is the main contributor to the similarity score used between neurons. We can imagine other criteria: (1) connectivity, (2) gene expression, (3) co-labeling in the same (split-)GAL4 lines. Of course, we do not at present have any such co-registered datasets for instance with gene expression and connectivity. Morphology is indeed currently the main bridge between measurements.

Given the possibility that neurons with different morphologies could yet have the same connectivity — there are anecdotal examples of neurons which (accidentally?) seem to go way out of their way just to loop back and make connections — it is hard to know how strongly we can take the Schlegel et al's claims of variability. I'm not sure what to do about this. I just want to get this off my chest. We could develop and use methods for clustering based on connectivity alone (for instance, see stochastic block models and Jonas & Kording Elife 2015).

At the very least, it would be helpful to see a limited manual validation that the morphology match is working reasonably well. A substantial fraction of central brain neurons (page5, 43%) could not be

assigned a terminal cell type. It would be helpful to see examples of neurons that cannot be matched to a type along with closest morphological and/or connectivity score hits.

Major comment 2

Related to the above, it is a little confusing how the cell typing is actually done. The confusion stems from the different ways in which this is done: by morphological matching to hemibrain types, by de novo clustering based on morphology, by morphological + connectivity matching to hemibrain. And it's a little hard to keep straight what was done for what analysis. This also raises the question of whether this work proposes a single clean quantitative and generalizable way to perform de novo cell typing in new future connectomes. It would be nice to hear the thoughts of the authors on this point.

The authors demonstrate how to refine cell type assignment by clustering across brains in Fig 6, but how robust is this clustering methodology?

The examples given in Fig6 use two different metrics, and appear to have also constrained which neurons to look at (i.e. only FC1-3, or FB1-9). The authors noted that FC1-3 is tiling/columnar, so morphological metrics were omitted from clustering. Whereas FB1-9 is tangential and can use both morphological+connectivity metrics. However this requires the user to know whether a neuron is tiling a priori. How would the clustering look if both morphology + connectivity are used for FC1-3? Is it detrimental to always include morphological scores in the clustering?

Similarly, would the clustering/dendrogram be sensible/interpretable if all central brain neurons were included in the across-brain procedure depicted in Fig6? And would brain regions or circuits appear in different parts of the dendrogram/cluster (e.g. fan-shape body vs mushroom body etc)?

Basically, is the clustering procedure here applicable to looking at small groups of neurons or can it work well globally across all neurons.

Major comment 3

The authors provide an interesting and compelling case study on variability in neuron count versus connectivity for kenyon cells in Fig 5. Neuron count variability is intrinsic to biology, as even sensory neurons (e.g. photoreceptors or # of ommatidia) can vary dramatically between individuals. How neural circuits handle dimensionality of input/output in general is beyond the scope of this paper. However the authors may provide a glimpse by examining "synapse budget" as in Fig 5 for all cell types.

Is the cell type to cell type synapse budget stable across all cell types? i.e. are the kenyon cell examples an exception or is it fairly common?

Does having a high connectivity similarity score (for cell type assignment) also mean that there's a fairly consistent "synapse budget" across brains?

Major comment 4

The authors provided an example of biological variability in asymmetric morphology in Fig 2J for lineage VPNd3 and cell types LC6+LC9. While optic lobe was not part of the detailed analysis, it is still worth quantifying whether morphological/connectivity metrics would actually group these neurons between left vs right hemisphere as the same cell type. i.e. every neuron on the left has a high morphological score hit with some neuron on the right vs some neurons just do not have "high" morphological matches from the other hemisphere.

Minor points:

1. If I get a vote, I actually prefer the old title of this paper from biorxiv v1: A consensus cell type atlas from multiple connectomes reveals principles of circuit stereotypy and variation. My main quibble with the current title: Whole-brain annotation and multi-connectome cell typing quantifies circuit stereotypy in *Drosophila*, is that it conveys the impression that whole brain has been cell typed. This is not true since the optic lobes (comprising 2/3 of the brain!) have not yet been cell typed.
2. I wonder about the strong assumption of discrete (hierarchical) clusters without considering the possibility of continuous manifolds of variation. Are there continuous variations amongst neurons of a single cell type which might accidentally lead to over clustering?
3. The shifted LC6/LC9 tract was previously reported in Morimoto et al 2020 eLife, see EM reconstruction methods section, and should be cited.

Author Rebuttals to Initial Comments:

We appreciate the high level of enthusiasm of the reviewers for this work and thank them for their specific constructive criticisms, which we believe have substantially improved the manuscript. We were particularly gratified that reviewers not only felt that this multi-connectome cell typing effort is an important contribution to *Drosophila* neurobiology, but will also be of relevance to future efforts in other species, and for these reasons should be of broad interest to the *Nature* readership.

In addition to addressing specific reviewer criticisms and suggestions, which we detail below, and making various minor clarifications and improvements to the text and figures, we also made major improvements:

1. Generation of cell type proposals for nearly all (98%) neurons in the central brain volume, including those not contained in the partial hemibrain volume. This effort was previously limited to a worked example in the central complex. These results are summarised in an updated Figure 6 and Supplemental Figure S6.2.
2. Cell types for >90% of all optic lobe neurons, where possible named based on the existing literature. Systematic names were defined for all newly identified visual projection neuron (VPN) cell types – as well as the smaller number of visual centrifugal neurons with cell bodies in the brain.

Note that accurate visual projection neuron typing was possible because we also typed the great majority (>92%) of neurons intrinsic to both optic lobes including the columnar cell types which typically provide the main driving input to VPNs; these intrinsic neurons consist of many cells but few types compared with the central brain (156 cell types, 77,489). This complements another new effort (Matsliah et al bioRxiv, 2023), which is more comprehensive in typing 97% of intrinsic neurons in a single optic lobe, defining 229 cell types (about half of which were not previously reported).

Together these changes provide comprehensive typing across the entire brain and increase the number of cell types from 4,179 to 7,846 and the number of typed neurons from 45,675 to 129,894 (93% of all neurons).

Annotations and analyses were updated and now use data from the next public release version (materialisation version 783) of the FlyWire database, which incorporates many proofreading corrections in the optic lobes. This data will be made publicly available via Codex and CATMAID shortly. In addition, the reviewers can access our data artefacts through:

1. <https://tinyurl.com/flywire783hb> points to a neuroglancer scene containing the FlyWire 783 segmentation + neuron meshes and is preconfigured to show and query our updated annotations (see also our Methods sections)
2. The “staging” branch of our Github repository contains our updated annotations and other data artefacts - see https://github.com/flyconnectome/flywire_annotations/tree/staging. That branch will be merged into the main repository when the Flywire 783 materialisation is publicly released.

Key areas of reviewer concern included:

1. How to access the curated types presented in this work within the context of the Codex resource, which hosts all community annotations at FlyWire.ai.

To address this, we 1) now provide user-friendly Neuroglancer scenes containing all annotated types as reported in this study 2) have improved programmatic access to these annotations and 3) provide additional guidance/tutorials for how to access them.

2. Why such a surprisingly large fraction ($\sim\frac{1}{3}$) of cell types proposed by the hemibrain dataset could not be located in the FlyWire dataset.

We now show that while essentially all ($\sim 99\%$) of neurons in the hemibrain dataset have clear and direct morphological matches to individual neurons in the FlyWire dataset (revised Figure 3B-D), individual neurons in the FlyWire dataset can sometimes match several of the cell types proposed by the hemibrain project equally well (revised Figure 3K). This creates ambiguity in the type assignment of FlyWire neurons to some cell types defined by the hemibrain connectome, and motivates joint definition of cell types using all available connectomes (revised Figure 6 and Supplemental Figures 6.1 and 6.2).

Specific concerns are addressed further below.

Reviewer 1

*In this paper, Schlegel and colleagues describe methods they developed and used to annotate neurons in a new whole brain connectome of adult *Drosophila melanogaster*. Importantly, they also perform several analyses that compare the connectivity and morphology of neurons across the two halves of the whole brain dataset and with another partial brain dataset already annotated. In particular, through these comparisons, the authors: develop heuristics for judging the importance of edges (these will be specific to fly connectomes); develop new definitions for cell types (which will likely be important beyond fly connectomes); and analyze surprising differences between the two extant *Drosophila* adult brain connectomes (unclear how these sorts of stereotypy differences will affect analyses of connectomes in other animals). The methods developed here are central to making the flywire dataset useful to the field, and some, like how to define cell types across brains, will likely be of broad interest as larger and larger connectomic datasets become available in other animals.*

Major comments

Major Comment 1

*1) Database issue: Overall in flywire, it seems confusing that each neuron can have multiple community annotations with no formatting consistency, and there does not seem to be a process outlined for going from community annotations to a single, unique cell type field with a standard format. As an example, *Tip6* is labeled multiply (the difference is using the word "with") as:*

a. #temp_NewType Translobula-plate 6; Tlp6 (flattened LOP arbors in layer 3 only, and small, sparse, arbors in the LO)

b. #temp_NewType Translobula-plate 6; Tlp6 (with flattened LOP arbors in layer 3 only, and small, sparse, arbors in the LO)

These sorts of unique annotations are searchable, but require a lot of parsing, and it seems like it would be important to have single, concise, unique cell type signifiers for new cells that do not have corresponding cell types in the hemibrain database.

The reviewer is correct: in Codex, which is a resource provided in the companion paper by Dorkenwald *et al.*, community annotations are intentionally designed to be free-form so as not to restrict the kind of information FlyWire users may contribute for their neurons of interest. This openness can make it difficult to easily extract coherent sets of strongly schematized sets of annotations, such as the one we provide here. This said, the search widget in Codex does allow filtering for specific fields; for example, using the query “cell_type {equal} Tlp6” will return only neurons where the schematized “cell_type” field matches the query.

To more directly and specifically extract the annotations described in this manuscript, we have made improvements to the way our annotations can be queried programmatically. Using these tools, users can now specify whether queries access the pooled annotations provided by Codex, or the hierarchical annotations described in this manuscript. Please see for the new tutorial for details:

https://fafbseg-py.readthedocs.io/en/latest/source/tutorials/flywire_annotations.html).

Major Comment 2

2) This issue of the left-right flip in the original data is really unfortunate and I understand that the authors are doing their best to rectify the situation, which appears to be more complicated than this reviewer understands. The entire discussion has still left me confused in quite a few situations about what is truly the left side. One case in point is Figure 3A, in which it looks like I'm seeing the brain from the front of the fly, but on which the reader's left is labeled "left", which I would expect to be the right side of the fly's brain. Does this mean that the brain is being presented as mistakenly reversed, but then correctly labeled? (Or am I mistaken about the brain's orientation?) Overall, I found that the discussion of what was done about this remained confusing, and I was also confused about where in this paper the neurons would be drawn correctly vs. only labeled correctly. This issue also extends to what future users should plan to do; do the authors advocate this paper's procedure as the clearest way to deal with this issue moving forward?

The reviewer is correct in that the brain was mistakenly reversed, but is correctly labelled. We have clarified the explanation for this (see “FAFB Laterality” section in the methods) and moved a briefer explanation to the front of the results section (from line 118, see also updated Supplemental Figure S1.2 + legend). We now also provide a tutorial on how to programmatically mirror neurons to produce the expected orientation for compatibility with other resources such as the Virtual Fly Brain project (see https://fafbseg-py.readthedocs.io/en/latest/source/tutorials/flywire_mirror.html)

Major Comment 3

3) I think I may have been more taken aback than the authors by the numbers of unmatched neurons and neuron types between flywire and the hemibrain.

a. From Figure 3C, it looks as though 16k neurons in flywire that share the hemibrain volume were untyped and 23k more that partially share the volume (“truncated”) were untyped. Conversely, almost half of the neuron types identified in the hemibrain volume do not show up at all in flywire. I think these differences are rather extraordinary and deserve a bit more attention. The authors spin the shared neuron types positively, but I think this result could equally be spun negatively. Readers should learn more about these untyped flywire neurons and unmatched hemibrain neuron types — the paper currently focuses on the successes, but should analyze what happened in these unmatched cases so we can understand the discrepancies between these two vast datasets.

We consider cell types to be falsifiable hypotheses - in particular with respect to hemibrain cell types in the “*terra incognita*” brain regions, which were based on only a single hemisphere and many of which are truncated by the boundaries of the hemibrain volume. With that in mind, we think the large fraction of proposed hemibrain types that could be confidently identified in FlyWire (61% in the submitted manuscript, now 68% in the revised manuscript) is a success.

However, the reviewer’s point is well taken: as we stated in both the abstract and results, it is indeed surprising that a substantial fraction of types could *not* be reidentified in FlyWire. To better explain this, we have now added analysis indicating that each individual hemibrain neuron likely has a good match in FlyWire (see Figure 3B-D). We think this will help make the point that the problem is with how hemibrain types were defined and not necessarily with finding matches for individual neurons (see also Figure 3K and Supplemental Figure S3.1H); i.e., stereotypy is great at the neuron level; but robust definition of cell type for all cells in the fly brain seems to require complete brains and multiple connectomes.

In more specific response to the reviewer’s questions:

i. For instance, are these untyped cells more likely to come from specific hemilineages? What do these untyped neurons have in common that they didn’t appear in the hemibrain dataset? Is this variability in neuron existence or just neuron morphology? Can one count cell bodies to ask about whether cell numbers are drastically different?

a-i) For this revision we have identified neurons belonging to 125 hemilineages in the hemibrain dataset. Comparison with FlyWire revealed a small trend towards fewer neurons per hemilineage in the hemibrain (Figure 2K). We also ran three-hemisphere (FlyWire + hemibrain) co-clusterings using a subset of 25 of these cross-identified lineages, and found anecdotal evidence that rarely, types may be exclusive to one or the other dataset (Supplemental Figure 6.2E-F) but we don’t believe this contributes substantially to the issue of unmatched hemibrain types.

Reviewer Response Figure 1 shows a breakdown of hemibrain types per hemilineage which demonstrates that some lineages do contain more unmatched neurons than others. For some lineages, truncation in the hemibrain is the likely explanation, for others truncation of **partner** neurons in the hemibrain may cause difficulties with connectivity clustering, for yet others determining a systematic reason is difficult. At this point we have concluded that it will be more valuable to compare our existing results with new and in-progress **complete** connectomes than to expend additional effort trying to establish the reasons for discrepancies with the truncated hemibrain.

ii. Do any of the unmatched neuron types in the hemibrain have split Gal4 lines, or matching morphologies among the single cell clones of the generation 1 gal4 collection? If so, is their reported expression variable (all or nothing?) among flies? If some unmatched neuron types appear to exist in Gal4 collections but not flywire, how should we approach this discrepancy?

a-ii) We have not performed that analysis as we feel this question, while interesting, is challenging to address comprehensively within the scope of this paper. In particular whether variability in expression patterns for a given GAL4 line is due to missing neurons or simply reflective of changes in gene expression is a major issue best addressed in separate work. This said, it is noteworthy that hemibrain cell types that precede the dataset (i.e. were principally based on light-level data) had a higher chance of being matched 1:1 than the *de novo* hemibrain types: 89% vs 50%, respectively (Reviewer Response Figure 1). This may suggest that types identified from GAL4 driver lines (i.e. genetically) are less variable than types identified from EM morphology alone; and/or, driver lines may tend to ‘lump’ types overly aggressively, whereas EM data better support oversplitting. Again, this needs to be treated systematically in a separate effort.

iii. How can we better think about this variability beyond just different growth conditions? Are there other experiments the authors could suggest that could help us

better understand what's going on but that don't require decade-long EM volume analyses?

a-iii) We now make clear that a (1) primary cause for the failure to assign FlyWire neurons to hemibrain types is that individual FlyWire neurons frequently match multiple candidate types (Figure 3K); and (2) individual neurons can almost always be strongly matched between FlyWire and hemibrain (Figure 3D). The issue may therefore lie in the initial definition of types, based on an n=1 dataset, rather than a greater variability in some types versus others. Although we argue additional connectomes will be required to address this issue, in principle it would be possible to generate or locate driver lines for neurons straddling hemibrain type boundaries and assess their variability across brains.

iv. Will any of the differences be attributable to the anisotropic voxels in the flywire dataset compared to the isotropic ones in the hemibrain?

a-iv) It is unlikely the difference in axial resolution between the two image datasets is the cause for missing type matches. Between the two datasets, the number synapses per cell type is highly correlated (Figure 4B); and for morphological comparisons (i.e. NBLAST) the relevant information is mainly in the backbone (i.e. large and medium diameter branches) which in our experience is easily recoverable from EM independent of isotropic or anisotropic resolution. The question of segmentation of isotropic vs. anisotropic EM data is also now treated further in the Discussion of the revised companion paper by Dorkenwald et al.

b. If the hemibrain made a policy of splitting when in doubt, it's unclear to me why Figure 3D has so few neurons in the many:1 category.

b) Although the hemibrain types were intended to be oversplit, there was no guarantee the splits would fall along the boundaries of types revealed by analysis of n>1 connectomes (revised Figures 3K and 6L-M). This and the strong morphological matching of individual neurons suggests that for unmatched hemibrain types, the issue lies in the type definitions, rather than greater-than-expected variability across brains.

Major Comment 4

4) I think adding more analyses on the cross-hemisphere similarities and differences would be a valuable contribution. In particular, I was left wondering what features/neuron morphologies/connectivities were most conserved across the hemispheres of the brain, and which were the most variable? For instance, the authors highlight LC6 and LC9 morphology differences across this brain, but there must be a spectrum of such things, and I would expect some regions/cell types to show more variability than others, and some features to show more variability than others (for instance, gross neural morphology vs. connectivity vs. synapse placement). Knowing what regions or cells are most and least variable seems likely to suggest specific hypotheses about the importance of stereotypy across the brain for different circuits.

Again the reviewer raises interesting questions. Throughout the proofreading and subsequent typing we recorded neurons we believe to be biological outliers which come in various flavours. With this revision we have added additional examples (see Supplemental Figure S3.2) and provide a quantification: only 0.4% of neurons in the central brain do something unexpected. These neurons can be found by filtering our annotations for “*status=outlier_bio*” (see Methods, “Biological outliers and sample artefacts”).

We also performed preliminary analyses of whether certain regions or neuron types were more variable than others. However, those analyses proved non-trivial to do rigorously (e.g. how to quantify the number of extra side branches) and any differences detected seemed small. We therefore elected not to pursue this line of work further.

Minor

1) On page 2, the authors cite the rich behavior of adult flies compared to larvae. (Reminder that reviewers appreciate line numbers.) I agree, but the citations here do not really seem to be representing rich behavior per se, but rather papers that often mix circuit dissection and behavioral results. Instead, it might be nice to use the opportunity to focus on papers that are really elucidating interesting flight, motor, or walking behaviors: grooming, navigation, courtship, escape, learning, walking coordination, etc. There are lots of papers to choose from here – but perhaps highlight ethological and behavioral studies here to better support the authors' point?

Response: We thank the reviewer for these suggestions, and agree that line numbers are helpful to everyone throughout the review and revision process. We have added line numbers to our revised manuscript. Regarding the inclusion of more ethologic descriptions of *Drosophila* behaviour, rather than ones focusing on circuit dissections of behaviour, we hope that interested readers can find entry points into the ethological literature via the papers we cite; but in the face of space limitations we have chosen to emphasise the literature on the circuit basis of behaviour since we believe this is ultimately an important use of the connectomic and cell typing efforts we present.

2) I may have missed this here or in other papers, but has the comparison been made of the randomness of the KC inputs between the two hemispheres? It seems important to understand if the randomness is brain to brain or between hemispheres, since the two options imply different mechanisms and functional outcomes...

This is a very interesting question. A key distinction is that the nonrandom input to KCs from olfactory projection neurons (PNs) was detected using neuron-level analyses. Zheng et al. 2022, which showed a clear signatures of non-random connectivity between food-responsive PN subtypes, depended on null models of PN-to-KC connectivity generated by permuting connectivity between individual PN boutons and KC claws. In contrast, in the present work, connectivity is analysed at the level of cell type, with the key theoretic value K defined as the average number of olfactory PN subtypes providing input to each KC (Figure 6F, 6K-L). This is sufficient to detect differences in K between brains; within-brain brain differences were not evident (Figure 6F). To extend the neuron-level analyses of Zheng et al. 2022 to the

contralateral hemisphere in FAFB/FlyWire, and to the hemibrain dataset, would require annotation of PN boutons and KC claws as distinct anatomical compartments, and is distinct in focus from the present work, which is on cell typing and connectivity at the level of cell types.

3) *Line XXX: “Connectivity-based typing is typically used recursively” – what does “recursively” mean in this case? Iteratively? This seems to be getting quite into the weeds on methods, for a general audience.*

We have rephrased the sentence to use “iteratively”. We also now provide line numbers; this change can be found on line 555 of the revised manuscript.

4) *Figure 1 has a *lot* of overlap with Dorkenwald et al. Figure 2. I’m not sure what to do about that, but they look and feel awfully similar.*

We agree with the reviewer that these are very similar figures. We shared our neuron renderings with *Dorkenwald et al.* for re-use in their figures as they are non trivial to produce (generating these images required a compute cluster node with a high-performance GPU, running custom scripts within a state-of-the-art 3D software package). At the same time *Dorkenwald et al.* have an understandable interest in presenting the resource which includes our annotations. Ultimately the papers need to stand on their own as well as work together, and we hope this level of moderate redundancy will be tolerable to the readership of each paper.

5) *Figure 2I – needs more description. How did the authors find the elbow? If a log transform is applied to the y-axis, is the elbow still in the same place? How much does it matter where this cut is made?*

In response to the reviewer’s critique we have replaced the elbow method. Instead, we now find a set of clusters that is consistent across the left and right hemispheres of FlyWire. Where possible, we also used hemibrain data to confirm. Please see the section on “Morphological groups” in the Methods for details.

Reviewer 2

*As a companion to the paper by Dorkenwald et al., which reports the generation of the first adult whole-brain connectome, this paper by Schlegel et al. describes the annotation of the connectome and analysis of stereotypy within and across brains. The study addresses broad and important questions regarding how we should interpret not only the newly completed *Drosophila* connectome, but connectomes in general – most notably, how cell types should be defined, to what extent a connectome is stereotyped across animals, and how technical versus biological noise contribute to connectome variability. This work provides the foundation for researchers to make use of present and future connectomes, fulfilling a critical need as connectomics has already begun to revolutionize the study of neural circuits.*

Without question, I would recommend that this study is published in Nature alongside the companion paper.

Technically, this study is rigorous and comprehensive. Every time I wanted to suggest an additional analysis or plot, I found that it was already included somewhere. The authors have been thoughtful and meticulous in addressing difficult questions, particularly the question of how to define a cell type. The text does a nice job of putting the new cell type atlas into perspective and describing the advances it represents over previous datasets. Some comments and suggestions are below.

Major comments

Major Comment 1

1. Do the authors have any ideas for why they were unable to find two of the known hemilineages?

Both of these hemilineages were reported in only a single publication (Yu *et al.*, 2013) and are missing from other collections. Given that we were also unable to find those lineages in the hemibrain dataset, we believe that they may in fact have resulted from a labelling pattern that did not target a single lineage (a known caveat of MARCM labelling as used in Yu *et al.*).

Major Comment 2

2. This is a comment on Codex (flywire.ai), not the manuscript itself: I think it should be more clear how the images in the database are inverted but the side annotations are not.

We agree with the reviewer that this is a potential source of confusion for Codex users. We have passed the reviewer's comment to the authors of the Dorkenwald *et al.* companion paper, including the maintainers of Codex. So far they have already added a note to the tooltip for the "side" annotations. We will keep working with them to add a more extensive explanation. We have also improved our explanation of the issue in the current manuscript (see response to Reviewer 1's major comment 2, above); to the extent that users of Codex read our paper, we hope any confusion will be reduced.

Major Comment 3

3. The authors report that the "connectivity types" in the hemibrain were generally not reproducible in Flywire. How should we interpret this? E.g. does this suggest that "connectivity types" represent an over-clustering of the data, as alluded to in the Discussion, or that connectivity patterns within a morphology class are real but not stereotyped across brains?

Both interpretations are potentially correct: computational analysis can identify connectivity types in the hemibrain that appear to represent overclustering when factoring in data from a second brain. This issue is now best illustrated with the "double vs triple clustering"

comparison in Figure S6 which shows that adding a third hemisphere leads to numerous cell type merges, effectively correcting the over-clustering of the data that we think happened in the hemibrain. In other words, connectivity clustering based on one brain (and especially one hemisphere) tends to over-cluster; this is corrected by comparison with a second brain.

Major Comment 4

4. Could the authors say a bit more about the potential reasons for the difference in mushroom body postsynaptic counts between the hemibrain and Flywire? If it's a technical bias in automated synapse detection, shouldn't this difference be lower for proofread synapses where errors are corrected? It could be helpful to provide an image of the "unusual" synapses being referenced.

In absolute terms, KC-to-KC connections account for most of the difference in MB synapse counts. However, other connections such as KC→MBON are also overrepresented - see Supplemental Figure S5D. We now provide an example in that figure panel of a synaptic connection from a single KC onto an MBON *and* two other adjacent Kenyon Cells to the same panel.

We unfortunately do not have a conclusive answer as to which synapse detection is better and which is worse - or indeed if the truth lies somewhere in between. The original hemibrain paper (Scheffer *et al.*, eLife, 2020) compared their synapse detection model against the one used in FAFB ("synful"; Buhmann *et al.*, Nat. Meth., 2021) and found that their model performed better (see their Appendix 1 - Figure 2). For FAFB, the Buhmann *et al.* model was validated against cutouts from various parts of the brain but unfortunately not the mushroom body lobes where we find most of the differences. Likewise, Scheffer *et al.* provide raw data for their f-scores but without associated neuropil names.

Major Comment 5

5. In Figure 5, I think statistics should be presented to support the authors' conclusions.

We thank the reviewer for this suggestion and have added statistical tests to Figures 5 and S5 where appropriate. Due to the high N's involved, most differences end up statistically highly significant despite very small effect sizes. To help the reader, we provide Cohen's *d* as a metric for the effect size.

Major Comment 6

6. The cell-typing approach in Figure 6 seems closely related to Figure 3, and seems a bit out of place at the end after the discussion of technical vs. biological variability and the mushroom body analysis. I would consider moving it earlier.

We thank the reviewer for this suggestion and in fact discussed grouping the results from Figure 3 and 6 together in the paper prior to our initial submission. However, we felt that because Figure 6 was more forward-looking, i.e. we define and apply a new method for defining cell types based on joint analysis of connectome data, it belonged at the end;

whereas in Figure 3, hemibrain types are sought (and largely found) in the FlyWire dataset, and the intervening figures are used to show interesting analyses based on this type matching. In the revised manuscript, we now also provide a complete proposed cell typing for the portions of the FlyWire dataset not contained by the hemibrain, including for both optic lobes, using the joint connectome approach across the two hemispheres of FlyWire. We also validate a selected subset of these proposed types using hemibrain data (Figure 6L-M, Supplemental Figure S6.2). We think this emphasises the prospective nature of Figure 6 and hopefully serves to further distinguish it from Figure 3.

Minor comments

p. 11: May want to clarify that any number of synapses (even 1) can define an edge, because this is different from the companion paper by Dorkenwald et al. where connections are always defined as at least 5 synapses.

Thank you, we have made this explicit in our definition of “edge” on page 12.

p. 12 and Fig. 4D: The numbers for within-brain edge comparisons don't match between the text and figure (57/59% vs. 58/60%).

We have corrected this.

p. 12: “edges of >10 synapses can be reproducibly found (>90%)” If this is referring to Fig. 4E I would suggest being more specific that edges of >10 synapses in the hemibrain are found in at least one Flywire hemisphere; otherwise it is unclear what “reproducibly found” means. Similar comments in a couple other places, e.g. does “chance of persisting” mean it must be found in more than one hemisphere or all three hemispheres?

We meant that if an edge in one hemisphere has more than 10 synapses, then 90% of the time, one can find that same edge in the other two hemispheres. We have clarified this text in the manuscript accordingly (from line 315).

p. 21: typo “have be”

Fixed, thanks!

p. 22: “Our criterion for strong (highly reliable) edges retains between 7-16% of edges and 50-70% of synapses.” I think this statement (mainly the word “retains”) is confusing because the numbers could be interpreted to be the percent of edges/synapses that are conserved across brains, not the percent that meet the “strong” threshold.

Rephrased to “applies to between 7-16% of edges and 50-60% of synapses”.

p. 39: Please define ECM

Fixed, thank you.

p. 40: Examples of cell classes are shown Table S3, not S2

Fixed, thank you.

Fig. S3E-F: These figures were hard to interpret and I am not sure I interpreted them accurately. It would be helpful to explain more clearly in the legend how the depicted clusters relate to the idea that SIP078 and SIP080 needed to be merged and PS090 needed to be split.

We have added additional explanations to the plots and the figure legend.

Fig. S4E: I am unclear about why the dotted lines were drawn at these values. Is an edge weight of 17 important for some reason?

This was an error in the original plot: the dotted line should have been at weight 11 to match our threshold of >10 synapses for reproducible edges. We have fixed that panel and added an explainer to the figure legend.

Reviewer 3

Schlegel et al reports on efforts to annotate the complete connectome of the flywire full adult fly brain (FAFB) reported Dorkenwald et al 2023, and to compare this new complete connectome to previously published partial connectome of the central brain (hemibrain). This is a landmark study. The hierarchical clustering of neurons into cell types and the assignment to developmental hemilineages is an incredible addition to the raw connectome and enables a deeper understanding of this unprecedented whole brain connectome dataset. Further, the pioneering comparative connectomic analyses between the two hemispheres of the flywire connectome and the hemibrain connectome provide insight into the degree of stereotypy exhibited by this insect brain. Finally, on a practical note, the paper develops a quantitative framework for two fundamental criteria: (1) what constitutes a cell type, and (2) what constitutes a statistically significant connection. Both will be practically useful in this new connectomic era.

Major comments

Major Comment 1

Cell types are defined as groups of cells that are quantitatively more similar to cells in a different brain than to any other cell in the same brain. I like this definition. But operationalizing this definition requires defining a quantitative notion of similarity between neurons and different choices will of course give rise to different cell types. Much of the analysis in the paper relies on the chosen definition of similarity being a good one.

My understanding is that morphological match (NBLAST score) is the main contributor to the similarity score used between neurons. We can imagine other criteria: (1) connectivity, (2) gene expression, (3) co-labeling in the same (split-)GAL4 lines. Of course, we do not at present have any such co-registered datasets for instance with gene expression and connectivity. Morphology is indeed currently the main bridge between measurements.

Given the possibility that neurons with different morphologies could yet have the same connectivity — there are anecdotal examples of neurons which (accidentally?) seem to go way out of their way just to loop back and make connections — it is hard to know how strongly we can take the Schlegel et al's claims of variability. I'm not sure what to do about this. I just want to get this off my chest. We could develop and use methods for clustering based on connectivity alone (for instance, see stochastic block models and Jonas & Kording Elife 2015).

At the very least, it would be helpful to see a limited manual validation that the morphology match is working reasonably well. A substantial fraction of central brain neurons (page5, 43%) could not be assigned a terminal cell type. It would be helpful to see examples of neurons that cannot be matched to a type along with closest morphological and/or connectivity score hits.

We agree with the reviewer that the choice of similarity metric will fundamentally affect which cell types will/can be extracted from the data. The crucial point from our perspective is that if we apply the same metric across multiple connectomes, we should ideally get the same cell types.

First, to address the question of whether matching by morphology alone works reasonably well, we manually reviewed a sample of hemibrain and their top hits in FlyWire by NBLAST score. Extrapolating from that sample to the entire population we expect that 99% of hemibrain neurons have a morphologically plausible match in the FlyWire brain (Figure 3B-E).

The main downside of using connectivity for across-brain matching of cell (types) is that it needs to be bootstrapped first - i.e. we need at least a starting set of labels shared between both datasets. By contrast, morphological matching requires a spatial (non-rigid) transform between both brain spaces which can be computed from the synapse cloud and therefore does not require having a corpus of known matches. We do agree though that connectivity can be very impactful for both cell typing as well as cross-matching of neurons. That is why we used both connectivity-alone as well as morphology+connectivity to try to resolve some morphologically ambiguous cell type matches. Details of this are provided throughout the Methods - see for example the section on "Hemibrain cell type matching with connectivity".

Major Comment 2

Related to the above, it is a little confusing how the cell typing is actually done. The confusion stems from the different ways in which this is done: by morphological matching to hemibrain types, by de novo clustering based on morphology, by morphological +

connectivity matching to hemibrain. And it's a little hard to keep straight what was done for what analysis.

For this revision we generated *de novo* cell types for almost all central brain neurons that were previously left untyped. We also identified cell types for ~90% of optic lobe neurons. With this, the cell types we provide come principally from three sources:

1. The hemibrain type matching effort which used an initial morphology-only matching followed by connectivity-matching to resolve conflicts.
2. The optic lobe cell typing for which we used a connectivity-only clustering with extensive literature search.
3. *De novo* cell types in the central brain which were generated from a systematic hemilineage-by-hemilineage morphology+connectivity co-clustering.

The above is summarised in Supplemental Figure S9. In addition, we have made improvements throughout the methods to clarify.

This also raises the question of whether this work proposes a single clean quantitative and generalizable way to perform de novo cell typing in new future connectomes. It would be nice to hear the thoughts of the authors on this point.

For *de novo* cell typing of a future connectome, we propose the following:

1. Initial morphology-only matching to existing connectomes to assign obvious matches. In the case of *Drosophila* we expect this to yield upwards of 2k cell types which can be used to bootstrap (2).
2. Run a combined morphology + connectivity co-clustering across all available connectomes, and extract balanced clusters. Use the balanced clusters to assign or refine existing cell types.

We have adjusted the “Lessons for cell typing” section of the Discussion to make this clearer.

The authors demonstrate how to refine cell type assignment by clustering across brains in Fig 6, but how robust is this clustering methodology?

This is a good question but difficult to answer conclusively given we currently only have 1.5 connectomes to work with. To at least partially address the question we compared two types of clustering on 25 cross-identified hemilineages untruncated in the hemibrain (Supplemental Figure S6.2):

1. A double co-clustering using the left and right hemisphere of FlyWire
2. A triple co-clustering using FlyWire + hemibrain

Both of these used morphology + connectivity. We then compared the balanced clusters derived from the double- versus the triple-clustering. This comparison suggests that up to 70% of the double-clusters survive addition of a third hemisphere with minor edits (1:many, many:1). That percentage increases to 84% if we account for small amounts of noise in our analyses. While we find these results very encouraging, they do emphasise the value of having more than one connectome in defining stable cell types.

The examples given in Fig6 use two different metrics, and appear to have also constrained which neurons to look at (i.e. only FC1-3, or FB1-9). The authors noted that FC1-3 is tiling/columnar, so morphological metrics were omitted from clustering. Whereas FB1-9 is

tangential and can use both morphological+connectivity metrics. However this requires the user to know whether a neuron is tiling a priori. How would the clustering look if both morphology + connectivity are used for FC1-3? Is it detrimental to always include morphological scores in the clustering?

NBLAST is - by design - sensitive to the neurons' absolute position in space which means that neurons of a cell type that tiles space (i.e. that are either non- or only partially overlapping) will be given a bad score and *vice versa*. In the case of FC1-3 NBLAST would therefore group multiple cell types innervating the same column instead of the same cell type across columns. It's possible to use morphology metrics that are independent of the absolute spatial position (e.g. Sholl analysis) but in our experience they are not as performant as NBLAST in most other scenarios. Where possible we typically combine NBLAST and morphology scores, in part because morphology serves as a useful constraint to prevent off target matches. That said, one could feasibly run a connectivity-only clustering first and only add morphology if that produces bad clusters/matches.

Similarly, would the clustering/dendrogram be sensible/interpretable if all central brain neurons were included in the across-brain procedure depicted in Fig6? And would brain regions or circuits appear in different parts of the dendrogram/cluster (e.g. fan-shape body vs mushroom body etc)?

Basically, is the clustering procedure here applicable to looking at small groups of neurons or can it work well globally across all neurons.

For the examples in Figure 6 it made sense to run the clustering only on the neurons of interest. In general, we think it is preferable to run smaller batches because it makes for faster NBLAST/connectivity calculations while also constraining the problem (less likely to get off target hits). That said, there are no technical reasons beyond CPU/memory limitations for running e.g. an all-by-all NBLAST. For this revision, for example, we ran a connectivity co-clustering of all 8k visual projection neurons. For human review of the resulting clusters, we then divided the dendrogram into more amenable chunks which worked very well.

Major Comment 3

The authors provide an interesting and compelling case study on variability in neuron count versus connectivity for kenyon cells in Fig 5. Neuron count variability is intrinsic to biology, as even sensory neurons (e.g. photoreceptors or # of ommatidia) can vary dramatically between individuals. How neural circuits handle dimensionality of input/output in general is beyond the scope of this paper. However the authors may provide a glimpse by examining "synapse budget" as in Fig 5 for all cell types.

Is the cell type to cell type synapse budget stable across all cell types? i.e. are the kenyon cell examples an exception or is it fairly common?

Does having a high connectivity similarity score (for cell type assignment) also mean that there's a fairly consistent "synapse budget" across brains?

These are very interesting issues but, as the reviewer notes, a detailed examination is beyond the scope of the current study. Nevertheless, we can provide an initial answer to the reviewer's questions using existing figures:

1. Figure 4B compares absolute synapse counts aggregated per cell. Both pre- and post-synapse counts are highly correlated within and across brains, showing that most cell types occupy a consistent fraction of the synaptic budget of the whole brain.
2. Figure 4D shows connectivity similarity within- and across-brains. This analysis uses cosine similarity which takes into account the direction of the connectivity vector but not its magnitude. Cosine similarity therefore operates on connectivity normalised to the neuron's total number of in- and outputs, i.e. the synapse budget.

Taken together, these data suggest that preservation of synapse budgets is likely not unique to Kenyon cells – although it's most obvious there given the very large differences in Kenyon cell numbers. There is additional evidence from comparing the larval *Drosophila* nerve cord circuits at different developmental stages (Gerhard *et al.*, eLife 2017) and the adult antennal lobe (Tobin *et al.*, eLife 2017) supporting this hypothesis.

Major comment 4

The authors provided an example of biological variability in asymmetric morphology in Fig 2J for lineage VPNd3 and cell types LC6+LC9. While optic lobe was not part of the detailed analysis, it is still worth quantifying whether morphological/connectivity metrics would actually group these neurons between left vs right hemisphere as the same cell type. i.e. every neuron on the left has a high morphological score hit with some neuron on the right vs some neurons just do not have "high" morphological matches from the other hemisphere.

Using the all-by-all NBLAST scores we provide as supplemental data we can address this question by asking "What are the top N contralateral matches for any given LC6/LC9 neuron?". For LC6, the top contralateral match is always another LC6 neuron. For LC9, the top contralateral match is another LC9 in 89% of all cases. This increases to 95% if we look amongst their top 5 contralateral matches. Furthermore the NBLAST score for the incorrect top contralateral matches is on average lower: 0.42 vs 0.34.

Minor comments

Minor points:

1. If I get a vote, I actually prefer the old title of this paper from biorxiv v1: A consensus cell type atlas from multiple connectomes reveals principles of circuit stereotypy and variation. My main quibble with the current title: Whole-brain annotation and multi-connectome cell typing quantifies circuit stereotypy in Drosophila, is that it conveys the impression that whole brain has been cell typed. This is not true since the optic lobes (comprising 2/3 of the brain!) have not yet been cell typed.

The reviewer raises a valid point. With this revision we have now cell typed >90% of the entire brain which we hope warrants the current title of the manuscript.

2. I wonder about the strong assumption of discrete (hierarchical) clusters without considering the possibility of continuous manifolds of variation. Are there continuous variations amongst neurons of a single cell type which might accidentally lead to over clustering?

The question of whether (some) cells represent a continuous spectrum rather than discrete types is something we considered ourselves. Many of our matching and typing efforts involved inspecting dendrograms e.g. from co-clustering of the two FlyWire hemispheres. In those dendrograms, cell types were typically well defined with small within- and larger between-cluster distances which argues for discrete types. This observation does not mean that there couldn't be a spectrum of morphologies (or connectivity) within a cell type.

3. The shifted LC6/LC9 tract was previously reported in Morimoto et al 2020 eLife, see EM reconstruction methods section, and should be cited.

Thank you very much. Although we discussed this observation with Michael Reiser's group as far back as 2017, we did not realise that they had gone on to publish it. Reference added!

Reviewer Reports on the First Revision:

Referees' comments:

Referee #1 (Remarks to the Author):

The authors have addressed all my comments from the first review. I appreciate in particular the new prose on the left-right flip and the improved usability in the database/accession tools. I find the explanations of the mismatches between the hemibrain and flywire data also now to be much clearer and more accessible.

Minor

Telling readers to cite a specific other paper in tandem with this one feels out of place in a scientific paper. I would expect to see this on the Codex and other software tools, where it feels more appropriate; this is also where I've seen this type of thing for other large software projects. (And this request already exists there.)

Line 200 – mention “see methods” for how you did morphological groups?

Figure 1cde are missing axis labels (presumably “count”?)

Referee #1 (Remarks on code availability):

My graduate student reviewed the original and revised code -- not the raw code for segmentation, but the interface designed to allow people to access the data. He reports that the changes made during review were good and made the data easier to work with.

Referee #2 (Remarks to the Author):

The revisions have significantly improved this paper, which already represented a landmark study providing a rigorous and comprehensive analysis of cell types and connectivity in this unprecedented dataset. The addition of new cell type labels that now cover >90% of neurons is impressive and represents a major advance from the previous submission. All of my previous comments have been addressed. Some very minor comments on the revision are listed below.

1. In the abstract, this sentence makes it seem like there is a large discrepancy between the hemibrain and FlyWire connectomes: “one third of the cell types proposed in the hemibrain connectome could not be reliably identified in the Flywire connectome”. I worry that readers (many of whom won't read the actual paper) may interpret this to mean that 1/3 of the hemibrain neurons don't exist in FlyWire, which is far from true.

2. Related to above: The abstract and text focus mainly on the number of cell types discovered, but I

personally think what could be emphasized more is 1) the ability to find neuron matches for 99% of hemibrain neurons, which demonstrates notable stereotypy across brains, and 2) how the new cell types cover 93% of all brain neurons and 98% of central brain neurons, which is impressive.

3. For the analysis in Fig. 3C, it would be helpful to know (approximately) how many neurons were manually reviewed since these results are being used to extrapolate numbers for the entire brain. I think the y-axis provides a clue, but it still would be good to state in the text or methods.

4. Line 575: What does the “129,894” number correspond to? I’m guessing it’s the number of FlyWire neurons assigned to a cell type, but could be helpful to clarify.

5. Minor typos:

- grammatical errors in line 189, 820
- “misguided” (line 791)
- “are neurons projection” (line 837)
- “miss-identified” (Supp Fig S9 legend)

Referee #2 (Remarks on code availability):

There is extensive code and documentation available at the link. I did not go through it myself to verify usability and reproducibility, but I believe many people in the community are already using it.

Referee #3 (Remarks to the Author):

Overall, I’m very pleased with this revision and impressed with the progress made towards completing cell typing of the optic lobes. I think the paper title is now fully deserved. :-)

I only have one major comment. I congratulate the authors for significantly expanding the number of neurons typed to 98% including 90% of the optic lobes. However, given that these new cell types are distinct from the Matsliah et al effort, and given that there will soon be a *Janelia* optic lobe connectome and cell typing effort released, I worry about how to guide the community towards reconciling these 3 disparate efforts. It would be very helpful to see some analysis and comparison of at least the optic lobe cell types of Schegel et al and Matsliah et al. I realize that these are contemporaneous efforts, but the community will be greatly helped by some guidance and analysis from the authors of this paper and Matsliah et al (many of whom are on both papers).

Author Rebuttals to First Revision:

Referees' Comments

Referee #1

Referee #1 (Remarks to the Author):

The authors have addressed all my comments from the first review. I appreciate in particular the new prose on the left-right flip and the improved usability in the database/accession tools. I find the explanations of the mismatches between the hemibrain and flywire data also now to be much clearer and more accessible.

Minor

Telling readers to cite a specific other paper in tandem with this one feels out of place in a scientific paper. I would expect to see this on the Codex and other software tools, where it feels more appropriate; this is also where I've seen this type of thing for other large software projects. (And this request already exists there.)

Although we do understand the reviewer's concern about specific citation instructions, the FlyWire resource is jointly described by both the Dorkenwald et al and Schlegel et al papers. We expect that citations for the resource will come not only from users of Codex and our other software libraries (e.g. natverse, navis), but also from authors discussing cell typing of the *Drosophila* connectome and connectomes in general. With that in mind, we do think providing guidance within this manuscript (in addition to the flywire website) is warranted. We have

rephrased the sentence from "[...] *should be co-cited* [...]" to "[...] *will preferably be co-cited* [...]" which we hope clarifies that this is not a dictate, but rather a request for consideration.

Line 200 – mention “see methods” for how you did morphological groups?

Added a reference to the methods.

Figure 1cde are missing axis labels (presumably “count”?)

Added axes labels to these panels.

Referee #1 (Remarks on code availability):

My graduate student reviewed the original and revised code -- not the raw code for segmentation, but the interface designed to allow people to access the data. He reports that the changes made during review were good and made the data easier to work with.

Thank you. We are pleased our changes improved usability.

Referee #2

Referee #2 (Remarks to the Author):

The revisions have significantly improved this paper, which already represented a landmark study providing a rigorous and comprehensive analysis of cell types and connectivity in this unprecedented dataset. The addition of new cell type labels that now cover >90% of neurons is impressive and represents a major advance from the previous submission. All of my previous comments have been addressed. Some very minor comments on the revision are listed below.

1. In the abstract, this sentence makes it seem like there is a large discrepancy between the hemibrain and FlyWire connectomes: "one third of the cell types proposed in the hemibrain connectome could not be reliably identified in the Flywire connectome". I worry that readers (many of whom won't read the actual paper) may interpret this to mean that 1/3 of the hemibrain neurons don't exist in FlyWire, which is far from true.

We rephrased that sentence in the abstract to "*Although nearly all hemibrain neurons could be matched morphologically in FlyWire, surprisingly, about one third of cell types proposed for hemibrain could not be reliably re-identified*".

2. Related to above: The abstract and text focus mainly on the number of cell types discovered, but I personally think what could be emphasized more is 1) the ability to find neuron matches for 99% of hemibrain neurons, which demonstrates notable stereotypy across brains, and 2) how the new cell types cover 93% of all brain neurons and 98% of central brain neurons, which is impressive.

We thank the reviewer for this suggestion. The abstract has been substantially condensed to meet Nature format requirements. However, we have brought this point forward, writing: “Although nearly all hemibrain neurons could be matched morphologically in FlyWire, surprisingly, about one third of cell types proposed for hemibrain could not be reliably re-identified.” Further we have edited the Discussion, stating “The cell type atlas that we provide of 8,453 cell types, *covering 96.4% of all neurons in the brain*, is to our knowledge the largest ever proposed” (emphasis added).

3. For the analysis in Fig. 3C, it would be helpful to know (approximately) how many neurons were manually reviewed since these results are being used to extrapolate numbers for the entire brain. I think the y-axis provides a clue, but it still would be good to state in the text or methods.

For this figure, we manually reviewed matches for 237 neurons spread evenly across the range of NBLAST scores. We added that number to the Methods.

4. Line 575: What does the “129,894” number correspond to? I’m guessing it’s the number of FlyWire neurons assigned to a cell type, but could be helpful to clarify.

That is correct. We have rephrased the sentence to make this clear.

5. Minor typos:

- *grammatical errors in line 189, 820*
- *“misguided” (line 791)*
- *“are neurons projection” (line 837)*
- *“miss-identified” (Supp Fig S9 legend)*

All fixed, thanks.

Referee #2 (Remarks on code availability):

There is extensive code and documentation available at the link. I did not go through it myself to verify usability and reproducibility, but I believe many people in the community are already using it.

Referee #3

Referee #3 (Remarks to the Author):

Overall, I’m very pleased with this revision and impressed with the progress made towards completing cell typing of the optic lobes. I think the paper title is now fully deserved. :-)

I only have one major comment. I congratulate the authors for significantly expanding the number of neurons typed to 98% including 90% of the optic lobes. However, given that these

new cell types are distinct from the Matsliah et al effort, and given that there will soon be a Janelia optic lobe connectome and cell typing effort released, I worry about how to guide the community towards reconciling these 3 disparate efforts. It would be very helpful to see some analysis and comparison of at least the optic lobe cell types of Schegel et al and Matsliah et al. I realize that these are contemporaneous efforts, but the community will be greatly helped by some guidance and analysis from the authors of this paper and Matsliah et al (many of whom are on both papers).

We appreciate and agree with the reviewer's comment that it will be important for the community to eventually reconcile these efforts. Given that our manuscript is likely the most general in scope, since it covers the whole brain, and also the first to be peer-reviewed, we feel a comparison and reconciliation of the optic lobe cell types would be better placed in either Matsliah *et al.*, the Janelia effort, or perhaps even ideally a subsequent stand-alone integrative review of all 3 datasets. We are of course already in touch with these other authors about achieving this in due course. Note also that we will continue to update the cell types released at https://github.com/flyconnectome/flywire_annotatons.